



Atmospheric
Measurement
Techniques

# The effect of low-level thin arctic clouds on shortwave irradiance: evaluation of estimates from spaceborne passive imagery with aircraft observations

**Hong Chen**[1,2]**, Sebastian Schmidt**[1,2]**, Michael D. King**[2]**, Galina Wind**[3]**, Anthony Bucholtz**[4]**, Elizabeth A. Reid**[4]**, Michal Segal-Rozenhaimer**[5,6,7]**, William L. Smith**[8]**, Patrick C. Taylor**[8]**, Seiji Kato**[8]**, and Peter Pilewskie**[1,2]

[1]University of Colorado, Department of Atmospheric and Oceanic Sciences, Boulder, CO, USA
[2]University of Colorado, Laboratory for Atmospheric and Space Physics, Boulder, CO, USA
[3]Science Systems and Applications, Inc., Lanham, MD, USA
[4]Naval Research Lab, Monterey, CA, USA
[5]Bay Area Environmental Research Institute Sonoma, Sonoma, CA, USA
[6]NASA Ames Research Center, Moffett Field, CA, USA
[7]Department of Geophysics, Porter School of the Environment and Earth Sciences, Tel Aviv University, Tel Aviv, Israel
[8]NASA Langley Research Center, Climate Science Branch, Hampton, VA, USA

**Correspondence:** Hong Chen (hong.chen-1@colorado.edu), and Sebastian Schmidt (sebastian.schmidt@lasp.colorado.edu)

**Abstract.** Cloud optical properties such as optical thickness along with surface albedo are important inputs for deriving the shortwave radiative effects of clouds from spaceborne remote sensing. Owing to insufficient knowledge about the snow or ice surface in the Arctic, cloud detection and the retrieval products derived from passive remote sensing, such as from the Moderate Resolution Imaging Spectroradiometer (MODIS), are difficult to obtain with adequate accuracy – especially for low-level thin clouds, which are ubiquitous in the Arctic. This study aims at evaluating the spectral and broadband irradiance calculated from MODIS-derived cloud properties in the Arctic using aircraft measurements collected during the Arctic Radiation-IceBridge Sea and Ice Experiment (ARISE), specifically using the upwelling and downwelling shortwave spectral and broadband irradiance measured by the Solar Spectral Flux Radiometer (SSFR) and the BroadBand Radiometer system (BBR). This starts with the derivation of surface albedo from SSFR and BBR, accounting for the heterogeneous surface in the marginal ice zone (MIZ) with aircraft camera imagery, followed by subsequent intercomparisons of irradiance measurements and radiative transfer calculations in the presence of thin clouds. It ends with an attribution of any biases we found to causes, based on the spectral dependence and the variations in the measured and calculated irradiance along the flight track.

The spectral surface albedo derived from the airborne radiometers is consistent with prior ground-based and airborne measurements and adequately represents the surface variability for the study region and time period. Somewhat surprisingly, the primary error in MODIS-derived irradiance fields for this study stems from undetected clouds, rather than from the retrieved cloud properties. In our case study, about 27 % of clouds remained undetected, which is attributable to clouds with an optical thickness of less than 0.5.

We conclude that passive imagery has the potential to accurately predict shortwave irradiances in the region if the detection of thin clouds is improved. Of at least equal importance, however, is the need for an operational imagery-based surface albedo product for the polar regions that adequately captures its temporal, spatial, and spectral variability to estimate cloud radiative effects from spaceborne remote sensing.

## 1 Introduction

Understanding the warming of the Arctic necessitates an understanding of the radiative impact of clouds and surface albedo, especially at the surface where the interaction with the cryosphere occurs (Curry et al., 1996; Shupe and Intrieri, 2004). Clouds cool the surface in the shortwave (SW) wavelength range by reflecting solar radiation and warm the surface in the longwave (LW) range. Low-level, liquid-bearing clouds have recently received special attention because they significantly contributed to the 2012 enhanced Greenland ice melt (Bennartz et al., 2013). When they are optically thin (liquid water path, LWP, smaller than $20 \, \mathrm{g \, m^{-2}}$), their SW cooling effect is small because they do not reflect much sunlight, especially when the surface is already bright. In the LW, on the other hand, their emissivity increases rapidly with the LWP, making them blackbodies, and they warm the surface, especially if they are at a low altitude. For larger LWPs, the SW cooling eventually dominates as the cloud becomes more reflective.

Valuable data on Arctic clouds have been collected by ground-based observations over the past few decades (Curry et al., 1996, Shupe et al., 2011), but they are limited in spatial coverage and need to be augmented by additional observations, especially from spaceborne remote sensing measurements to help gain meaningful insights into cloud radiative effects in the Arctic as a whole.

Hartmann and Ceppi (2014) used the dataset from the Clouds and the Earth's Radiant Energy System (CERES) and showed that every $10^6 \, \mathrm{km^2}$ decrease in September sea ice extent is associated with a $2.5 \, \mathrm{W \, m^{-2}}$ increase in annual-mean absorbed solar radiation averaged over the region from 75 to 90° N. Kay and L'Ecuyer (2013) used combined products from active and passive remote sensing and showed that during the 2007 summer, the cloud reduction and sea ice loss in the Arctic resulted in more than $20 \, \mathrm{W \, m^{-2}}$ anomalies in shortwave radiation at the top of the atmosphere (TOA). The radiation products used in these studies, e.g., CERES-EBAF (Clouds and Earth's Radiant Energy Systems – Energy Balanced and Filled; Loeb et al., 2012) and 2B-FLXHR-LIDAR (Level 2B radiative fluxes and heating rates calculated from radiative transfer model by utilizing radar–lidar cloud and aerosol retrievals from A-Train satellites; Henderson et al., 2013), all rely on coincident cloud observations from the Moderate Resolution Imaging Spectroradiometer (MODIS).

MODIS is a 36-band passive imager on board the Terra and Aqua satellites. It provides cloud optical parameters (COPs), e.g., cloud optical thickness (COT), cloud effective radius (CER), and cloud thermodynamic phase, from which irradiance can be derived. The COPs from MODIS have been used extensively in studies of cloud radiative effects (e.g., Wielicki et al., 1996; Platnick et al., 2003; Loeb and Manalo-Smith, 2005; Oreopoulos et al., 2016). Due to the lack of temperature and reflectance contrast between clouds and the underlying surface in the Arctic, detecting the clouds is challenging for passive remote sensing, especially when they are thin and occur at a low level. Liu et al. (2010) showed that the MODIS cloud detection algorithm performs better over the ocean than over the ice. The traditional cloud retrieval algorithm (Nakajima and King, 1990) retrieves COT and CER from the reflectance at two channels, one where clouds do not absorb (660, 860, or 1240 nm) and one where cloud drops are weakly absorbing (1630 or 2130 nm). Over snow and ice, the surface albedo is already high in the visible and near-infrared ranges (leaving little dynamic range for cloud remote sensing of optical thickness) and varies regionally and temporally (leading to uncertainties in the retrieval products). This, in combination with low-sun conditions, makes it difficult to obtain accurate cloud optical properties from passive remote sensing. To improve the reliability of MODIS cloud retrievals in the Arctic, an algorithm has been developed that uses two shortwave-infrared bands of 1630 and 2130 nm, where snow and ice are relatively dark (Platnick et al., 2001; King et al., 2004). However, the surface albedo varies with surface type even for these bands, and the operational algorithm assumes constant values obtained from a climatology based on 5 years of Terra MODIS data (Moody et al., 2007).

In addition to the COPs themselves, the snow–ice surface albedo also plays an important role in determining the cloud radiative effect and radiation energy budget in the Arctic (Curry et al., 1995; Shupe and Intrieri, 2004). The surface albedo changes significantly from the visible to the near-infrared wavelength ranges (Wiscombe and Warren, 1981; Brandt et al., 2005) with different spectral dependence depending on the surface conditions (e.g., snow and ice). Inhomogeneous surface conditions such as floes of partially snow-covered ice, varying snow depth and snow grain size, and surface topography (e.g., sastrugi) all affect the spectral shape and magnitude of the surface albedo. To improve the understanding of the inhomogeneous Arctic surface and the spectral dependence of surface albedo, spectral surface albedo measurements for snow and ice have been collected during ground-based field experiments in the polar regions (e.g., Perovich et al., 2002a; Brandt et al., 2005). In addition, Perovich et al. (2002b) showed that different surface types, e.g., ice, ponds, leads, can be identified from aerial camera images through an image-processing software. Moreover, a spectral surface albedo model has been developed for different Arctic surfaces such as white sea ice, snow, and melting ponds on sea ice (Malinka et al., 2016, 2018). However, an operational surface albedo product based on spaceborne observations is still not available for the polar regions – in contrast to the land surfaces of the lower latitudes (Strahler et al., 1999).

Finally, accurate knowledge of the water vapor is also important, even in the shortwave (as we will show in this paper). In summary, the challenges for deriving shortwave irradiance from passive remote sensing are (a) inaccurate detection of clouds and cloud optical property retrievals over snow or ice surfaces, (b) lack of accurate surface albedo as a constraint

11 September    13 September

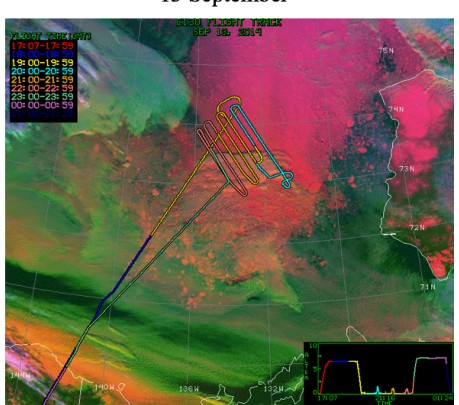

**Figure 1.** ARISE flight tracks overlaid on MODIS false color imagery (0.65 μm for red, 11 μm for blue, and 3.7–11 μm for green) from NASA Langley Research Center on 11 and 13 September 2014. The focus region of these two research flights was [72.5° N, 74.5° N, 136° W, 130° W] in the marginal ice zone.

in the radiative transfer model (RTM), and (c) insufficient knowledge about the water vapor profile.

The aim of this paper is to use aircraft radiation measurements collected during the NASA Arctic Radiation – Ice-Bridge Sea and Ice Experiment (ARISE; Smith et al., 2017) to evaluate irradiance as derived from coincident satellite imagery and to investigate the causes of any biases. In the first step, the spectral snow surface albedo was derived from upwelling and downwelling irradiance measurements, accounting for partially snow-covered scenes by the snow fraction estimated from aircraft camera imagery. In the second step, we used an RTM to calculate the upwelling and downwelling broadband and spectral irradiance at flight level, incorporating the MODIS-derived COPs and spectral surface albedo derived from the aircraft measurements as inputs.

The calculated irradiances were then compared with the measured broadband and spectral irradiance pixel by pixel for two cases – above-cloud and below-cloud. Section 2 describes the data and method used in this study. Section 3 provides the results and discussions for the measured spectral surface albedo, as well as for the comparisons between irradiance calculations and measurements. Conclusions are drawn in Sect. 4.

## 2 Data and methods

ARISE was a NASA airborne measurement campaign to study snow and ice properties in the Arctic marginal ice zone (MIZ) in conjunction with cloud microphysics and radiation (Smith et al., 2017). The NASA C-130 aircraft was instrumented with shortwave and longwave radiometers, described in this section, along with cloud microphysics probes, aerosol optical properties instruments, and snow and ice remote sensors. The experiment was based at Eielson Air Force Base near Fairbanks, Alaska, from 2 September to 2 Octo-

ber 2014, to capture the September sea ice minimum. In the Arctic, overpasses of polar-orbiting satellites are fairly common. ARISE targeted multiple overpasses of MODIS and CERES on Aqua and Terra or VIIRS on Suomi National Polar-orbiting Partnership (NPP) on almost every flight. One of the primary objectives of ARISE was to validate irradiance (or flux densities) derived from CERES–MODIS observations with aircraft radiation measurements. Figure 1 shows two science flights on 11 and 13 September that sampled above- and below-cloud conditions, respectively. These flights include so-called "lawn mower" patterns, a series of parallel flight legs laterally offset by about 20 km. They were specifically designed for ARISE to sample one or two 100 km × 100 km grid boxes per flight with a sufficient number of coincident CERES footprints (each with a 20 km diameter at nadir), as to acquire statistically significant above- or below-cloud aircraft measurements for the validation of CERES–MODIS-derived irradiance.

Comparing the aggregated data from ARISE directly with the CERES–MODIS flux products within the grid box, e.g., using histograms, is challenging because of the heterogeneity of the scenes in terms of surface albedo, cloud conditions, and changing solar zenith angle. Therefore, in this paper, we instead compare aircraft observations directly (pixel by pixel) with calculations based on MODIS cloud retrievals along the flight track. The comparison of the aggregated data with CERES–MODIS products is done in a separate publication; we do not use CERES in our analysis because its large footprint does not lend itself to a direct comparison with aircraft data in a heterogeneous environment.

The first step is to merge observations of the broadband shortwave irradiance from the BroadBand Radiometer system (BBR, details in Sect. 2.1) and of the spectral shortwave irradiance from the Solar Spectral Flux Radiometer (SSFR, details in Sect. 2.2). This merged product combines the high radiometric accuracy and high-fidelity angular re-

sponse from BBR with the spectral resolution from SSFR and is referred to as "SSFR–BBR" data. From these data, the surface albedo is derived for low-level legs under clear-sky conditions. To account for the heterogeneous surface (dark ice mixed with snow-covered ice), the surface albedo is acquired as a function of snow fraction, which is estimated from images of a downward-looking video camera (Sect. 2.3; details on the snow-cover-dependent surface albedo derivation in Sect. 3.1). Finally, atmospheric profiles and reanalysis data (Sect. 2.4) along with MODIS cloud products are used to calculate all-sky spectral and broadband irradiances along the flight track (Sect. 2.5), for subsequent comparison with the observations in Sects. 3.2 and 3.3.

## 2.1 BroadBand Radiometer system (BBR)

The BBRs deployed during ARISE are modified CM 22 precision pyranometers from Kipp & Zonen (Bucholtz et al., 2010). The BBR included downward-looking and upward-looking sensors. The radiometers were fix-mounted on the aircraft and measured upwelling and downwelling broadband irradiance (unit: $W\,m^{-2}$), that is, the spectrally integrated irradiance from 200 to 3600 nm. To account for the change of sun-sensor geometry due to aircraft attitude (pitch and roll), a software attitude correction (Long et al., 2010) was applied to the BBR data. In addition, a sunshine pyranometer (SPN1) was flown to measure diffuse and global radiative fluxes (Badosa et al., 2014; Long et al., 2010). The SPN1 radiometer was originally intended for ground-based use but is suited for airborne measurements of global and diffuse radiative fluxes because it does not have any moving parts, unlike traditional instruments such as the Multifilter Rotating Shadowband Radiometer (MFRSR). Smith et al. (2017) provide mission-specific details on both instruments. The BBR has a reported uncertainty of 3 % (Smith et al., 2017).

## 2.2 Solar Spectral Flux Radiometer (SSFR)

To attribute discrepancies between satellite-derived irradiance and airborne observations to causes such as erroneous water vapor, cloud properties, or three-dimensional radiative transfer effects, spectrally resolved measurements are needed (Schmidt and Pilewskie, 2012). SSFR is a moderate-resolution flux spectrometer built at the Laboratory for Atmospheric and Space Physics (LASP, University of Colorado Boulder). It is an updated version of the heritage spectrometer system originally developed at NASA Ames (Pilewskie et al., 2003). The SSFR radiometer system consists of two spectrometers for each viewing direction (zenith and nadir): (1) a Zeiss grating spectrometer with a silicon linear photodiode detector array covering a wavelength range from 350 to 950 nm and (2) a Zeiss grating spectrometer with an InGaAs linear photodiode detector array covering a wavelength range from 950 to 2150 nm. The spectral resolution of the silicon channels is 6 nm with a sampling of 4 nm. For the

InGaAs channels, the spectral resolution is coarser – 12 nm with 6 nm sampling. From the SSFR measurements, spectral albedo, net flux, and absorption can be derived.

SSFR is typically flown in conjunction with an active leveling platform (ALP, also built at LASP), which was developed for counteracting the changing aircraft attitude to keep the zenith light collector horizontally aligned (the nadir light collector was fix-mounted). This is particularly important in the Arctic, where low sun elevations lead to large systematic errors for fix-mounted or poorly stabilized sensors (Wendisch et al., 2001). One reason is that radiation from the lower hemisphere (for example, from clouds below or at the aircraft altitude) is registered by the zenith detector when it is tilted, which leads to systematic biases that cannot be corrected. Another reason lies in the specific design of the SSFR light collectors, which are realized as integrating spheres with a circular aperture on top. They diffuse the incoming light collected by the aperture and bundle it into a fiber optics cable that transmits it to the radiometer system inside the aircraft (Schmidt and Pilewski, 2012). The integrating sphere has an imperfect response to the incidence (polar) angle $\theta$ (Kindel, 2010), in contrast to the response of broadband radiometers such as BBR, which are closer to $\cos(\theta)$ as required for irradiance. At high sun elevations, a so-called hot-spot arises from a baffle that prevents light from being directly transmitted into the fiber optics. Since the response deviates significantly from $\cos(\theta)$, the direct and the diffuse light need to be corrected. This is done by separating the diffuse and direct components, using radiative transfer calculations in conjunction with SPN1 measurements (details are provided in Appendix A) and further assuming that the downwelling diffuse radiation is close to isotropic. This assumption is an approximation, which becomes invalid if parts of the lower hemisphere are in the light collector's field of view.

The light collector's angular response to the azimuthal angle also needs to be considered. Throughout the course of the mission, the zenith data revealed a dependence on the relative azimuth of the sun to the aircraft. This dependence was characterized at the end of the mission, by two calibration circles flown on 2 October. The non-homogeneous azimuthal response of the zenith light collector occurred for solar zenith angles greater than 66°. Generally, an azimuthally variable response could be attributed either to aircraft interference (e.g., by the tail and/or propellers of the host aircraft) or to the light collector itself. For the former, BBR and SPN1 (both fix-mounted on the C-130) would also be affected. To assess their azimuthal response, the attitude-corrected BBR data (Bannehr and Schwiesow, 1993; Bucholtz et al., 2008; Long et al., 2010) were compared with the SPN1 global irradiance data, as well as with radiative transfer calculations. This comparison revealed that in this case, aircraft interferences were minor compared to atmospheric effects (e.g., cirrus) and that only SSFR measurements, but not BBR and SPN1, had a significant azimuthal dependence, suggesting

the SSFR light collector as the source, rather than aircraft interferences. In order to determine the azimuthal dependence, the SSFR measurements were referenced to the BBR measurement[1] during the calibration circle (details in Appendix B). This azimuthal correction function (dependent on the relative azimuth angle of the aircraft and the sun) was then used for the zenith SSFR data for all research flights. After azimuthal correction, the SSFR downwelling irradiance was scaled to BBR using the method described in Appendix D. It is in this sense that the BBR and SSFR measurements are merged. By using BBR, SPN1, and SSFR in such a way, the redundancies between the instruments were used to capitalize on the strengths of the individual instruments (BBR: un-biased angular response and high radiometric accuracy; SPN1: diffuse/global separation; SSFR; spectral resolution for subrange of BBR and SPN1). The SSFR nadir signal was also referenced to the BBR data in a similar manner (see Appendix D) because BBR has the better angular response, whereas SSFR provides spectral resolution. The details about the merging method and the uncertainties of the merged irradiance product are provided in Appendix D.

The angular dependence of SSFR was verified in the laboratory. In addition, wavelength and radiometric calibrations were performed before and after the mission. The wavelength calibrations ensured spectral accuracy by referencing the SSFR measurements to several line sources. The primary radiometric calibration, performed with a NIST-traceable (National Institute of Standards and Technology) calibrated lamp, links SSFR measured digital counts to spectral irradiance. The radiometric calibration was also transferred to a so-called secondary radiometric field standard, which monitored the stability of the radiometers throughout the mission.

### 2.3 Imagery from downward-looking video camera

A downward-looking video camera (referred to as "nadir camera") is often included as a standard payload on NASA aircraft. It is a standard, commercially available video camera and typically records scenes for context only and is not radiometrically or geometrically calibrated. Despite this shortcoming, the videos recorded by the nadir camera are used for quantitative image analysis. From the video, we first extract image frames with an average rate of 2 Hz (two frames per second). The extracted image has a pixel resolution of 2592 (width) × 1944 (height). To co-register the aircraft nadir imagery with the measurements from other instruments, the times for the individual image frames are needed, but the image frames themselves did not contain a digitally stored time. They include a timestamp located at the lower left side

that contains time information, and we used optical character recognition (OCR) to retrieve the time from this information.

In the second step, the nadir camera imagery was used to quantify the fractional snow coverage. The snow fraction, which is the fraction of bright pixels of the image, was estimated. To this end, the image was converted from RGB (red, green, and blue) into grayscale by

$$\text{gray} = 0.299\text{R} + 0.587\text{G} + 0.114\text{B} \tag{1}$$

for each pixel. The weights come from standardized encoding recommendations for television (referred to as BT.601[2]). Another choice would have been to use a single-color channel, or even use the color information to distinguish surface types, but that was not necessary here. For more sophisticated imagery analysis, see Perovich et al. (2002b).

One issue of the nadir camera imagery was the darkening effect from the center to the edge of its field of view, which is known as the vignette effect. To compensate, the brightness of the image was linearly increased from edge to center through an image blending and interpolation technique by Haeberli and Voorhies (1994):

$$\text{out} = (1 - \beta) \times \text{black} + \beta \times \text{gray}, \tag{2}$$

where black is a black image with the same dimensions as gray, and $\beta$ is the image blending factor, a 2-D matrix with increasing values of 1.1–1.5 from the image center to the edge. The operator "×" denotes element-by-element multiplication. To avoid the vignetting extremes in the corners, only the imagery within a concentric sampling area was used to derive snow fraction (left panel of Fig. 2a). The key step of the snow fraction detection algorithm is the separation of dark versus bright pixels. To do this, an adaptive thresholding technique was applied. It is an approach for handling an image with unevenly distributed intensities by dividing the image into subimages and assigning different thresholds for each of the subimages (Gonzalez and Woods, 2002). The details of the adaptive thresholding are described in Appendix C. The snow fraction is then estimated by

$$\text{Frac} = \frac{N_{\text{bright}}}{N_{\text{total}}}, \tag{3}$$

where $N_{\text{bright}}$ is the number of pixels above the variable threshold, and $N_{\text{total}}$ is the total number of pixels within the sampling area. The imagery and detection results are illustrated in Fig. 2a, whereas Fig. 2b shows the simultaneously measured upwelling and downwelling spectral flux. The uncertainties associated with the estimated snow fraction are discussed in Appendix D.

---

[1] Since BBR has a near-ideal angular response, the attitude correction with respect to the polar angle can be performed by software as long as data are limited to small deviations from level. By contrast, SSFR with its non-ideal angular response requires an active leveling platform.

[2] https://www.itu.int/dms_pubrec/itu-r/rec/bt/R-REC-BT.601-7-201103-I!!PDF-E.pdf, last access: 17 October 2020

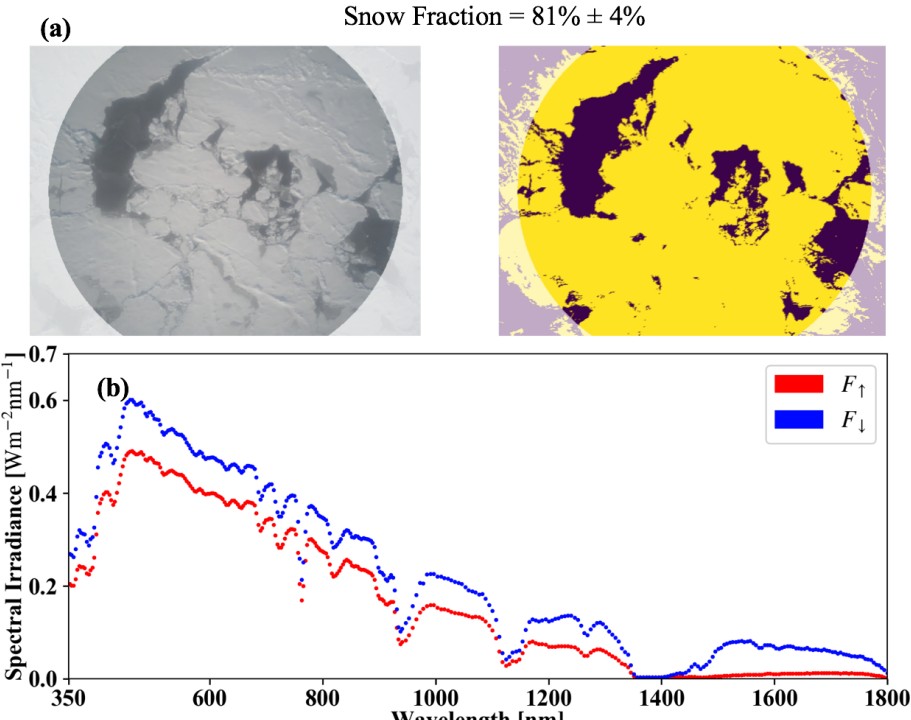

**Figure 2. (a)** An example of the snow fraction along with its uncertainty estimated from the nadir camera imagery at 20:03:32 UTC on 13 September, at 73.85° N, 132.95° W. The flight altitude was 134 m. The left panel is the nadir camera imagery. The radius of the field of view was about 380 m. The right panel uses yellow and purple to indicate bright and dark pixels as detected by the adaptive thresholding method. The snow fraction is derived from the abundance of yellow pixels. **(b)** The upwelling and downwelling irradiance from SSFR–BBR at the same time.

### 2.4 C-130 thermometer and hygrometer and Modern-Era Retrospective analysis for Research and Applications version 2 (MERRA-2)

The NASA C-130 aircraft was equipped with a thermometer and a hygrometer to measure air temperature and relative humidity, but it did not carry a dropsonde system. Figure 3b shows the profiles derived from the C-130 during a descending leg from 19:31:14 (altitude: 6.447 km) to 19:50:05 (altitude: 0.258 km) on 13 September 2014. Due to a malfunction of the hygrometer on 11 September 2014, no water vapor profile from the C-130 is available on this day. Instead (Fig. 3a), we used the temperature and water vapor content profiles from MERRA-2, which is an atmospheric reanalysis dataset from NASA (Bosilovich et al., 2015). MERRA-2 (M2I3NVASM) provides 3-hourly assimilated 3-D meteorological fields (dimensions: 576 in longitude; 361 in latitude; 72 pressure levels from 985 to 0.01 hPa). The comparison of the in situ profiles and MERRA-2 (Fig. 3b) shows good agreement, although the reanalysis does not reproduce the details of the vertical profile. A more systematic comparison of reanalysis and in situ data from ARISE is done by Rozenhaimer et al. (2018) and is not the focus of this paper. The observations reveal much drier and slightly colder conditions than captured in the subarctic climatology from An-

derson et al. (1986), referred to here as AFGL. Nevertheless, we used the climatology above 6.5 km to provide complete temperature and water vapor profiles from 0 to 120 km, after rescaling them to the observed temperature and water vapor values at 6.5 km. The constructed atmospheric profiles were then used in the RTM (described in the next subsection) to obtain irradiance calculations.

### 2.5 Radiative transfer calculations based on MODIS cloud products

The publicly available pixel-level MODIS cloud products (MOD/MYD06, collection 6.1), which are provided in 5 min granules (Platnick et al., 2017b), are used in this study. The MODIS cloud product includes COPs such as COT, CER, and cloud thermodynamic phase, which are essential parameters for calculating cloud radiative effects. As described before, the MODIS COT and CER are retrieved simultaneously using a bispectral reflectance method (Nakajima and King, 1990). To minimize the influence of the surface on cloud retrievals, the 1630 and 2130 nm bands are used since the snow and ice surfaces are relatively dark at those two bands (Platnick et al., 2001; King et al., 2004). These retrievals are included in the MOD/MYD06 files and will be referred to as the "1621" cloud product. Limited in situ observations sug-

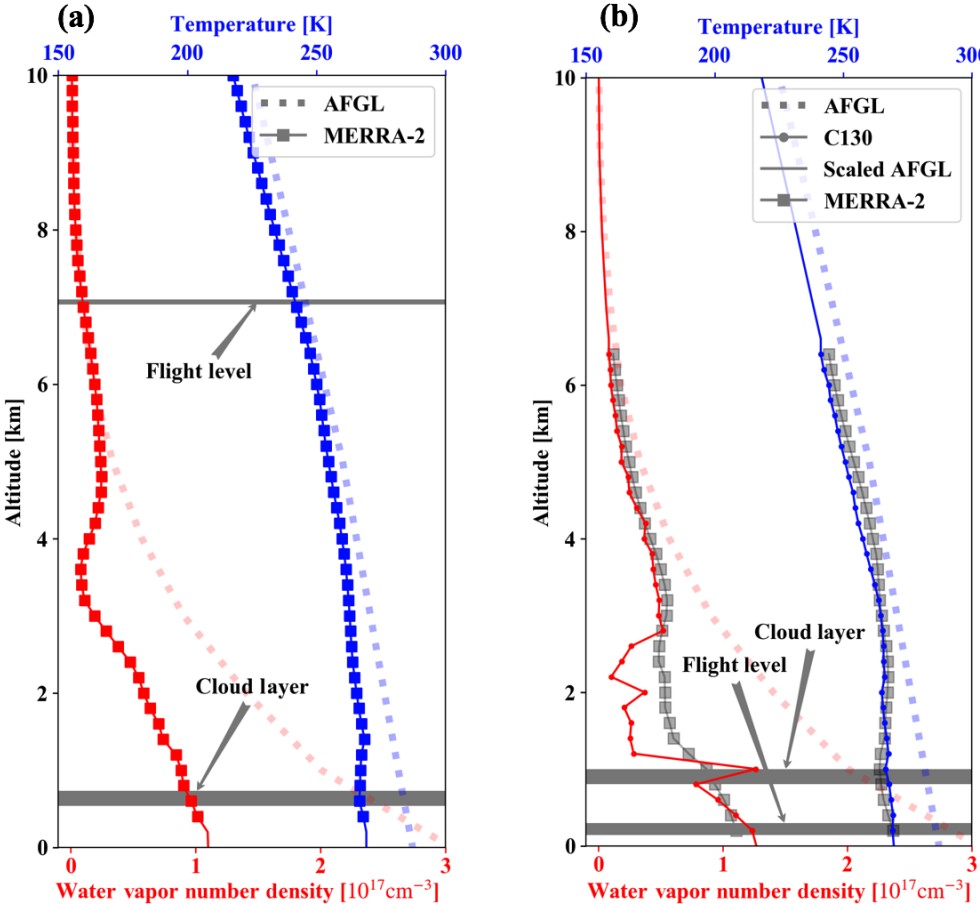

**Figure 3.** Vertical profiles of temperature and water vapor from MERRA-2 and from the climatology (AFGL) for **(a)** 11 September and **(b)** from the C-130 for 13 September 2014. On 11 September, MERRA-2 data at 21:00 UTC were averaged over the region of 72.5° N, 74° N, 135° W, 130.625° W to represent the atmospheric profile there. The vertical cloud distribution was unavailable from the in situ data. On 13 September, aircraft data from a descending leg (19:31 to 19:50 UTC at 74.1° N, 133.8° W) were used for the atmospheric profiles. Based on the water vapor profile, the cloud was likely located below 1.0 km (indicated in gray). Since hygrometer measurements were not available on 11 September, the cloud top height (1.1 km) was obtained from the MODIS L2 product) and the geometric thickness was set to 0.2 km (just like on 13 September). The flight level range is also shown. The solid lines for both days represent the temperature and water vapor profiles that went into the radiative transfer calculations.

gested that the clouds consisted primarily of liquid water, and the MODIS cloud phase product showed less than 2 % of ice clouds along the flight track. Therefore, the clouds were assumed to be liquid.

The MODIS 1621 product includes COPs for cloudy and partially cloudy conditions. The latter are denoted as "PCL" in the MODIS data variable name. The product was extracted along the flight track and then input into a radiative transfer model (RTM) to calculate spectral and broadband irradiance

at flight level. A 1-D RTM (libRadtran version 2.0.1, Emde et al., 2016) was used for the calculations. It requires the following inputs:

1.  day of the year (for accurate sun–Earth distance);

2.  atmospheric profile, here, the subarctic summer atmo-
spheric profile from Anderson et al. (1986) along with

a.  water vapor content profile from MERRA-2 for 11 September and from the C-130 hygrometer for 13 September 2014,

b.  temperature profile from MERRA-2 for 11 September and from the C-130 thermometer for 20 13 September 2014;

3.  solar zenith angle;

4.  wavelength;

5.  surface albedo at the specified wavelength (see Sect. 3.1);                                                          25

6.  slit functions (also known as instrument line shape), which describe the bandpass function of the spectrometer (here, SSFR slit functions as measured in the laboratory are used: full width at half maximum (FWHM) of

6 nm for the silicon channels and FWHM of 12 nm for InGaAs channels);

7. cloud optical thickness and cloud effective radius, here, from MODIS COPs;

8. phase functions, here, from Mie calculations distributed with libRadtran;

9. output altitude grid, here only at the aircraft flight level.

The RTM uses a solar spectrum with 1 nm resolution as solar source at TOA (Kurucz, 1992). The Discrete Ordinate Ra-
10 diative Transfer program (DISORT; Stamnes et al., 1988) is used as the radiative transfer solver. LOWTRAN 7 (Pierluissi and Peng, 1985) is used for the molecular absorption param-
eterization. The cloud layer altitude was set to 0.8 to 1.0 km for 13 September according to the water vapor profile from
15 the aircraft hygrometer. Since the hygrometer data were not available for 11 September and the cloud layer could not be identified from the temperature profile, the mean of cloud top height from MODIS and a cloud geometrical thickness of 0.2 km were used in the calculations. The RTM output in-
20 cludes downwelling (global and direct) and upwelling irradi-
ance at the specified wavelengths and output altitude (in this case, at the flight altitude). The cloud layer location and flight level altitude range were indicated in Fig. 3. The wavelength range of the calculations is set to 200 to 3600 nm, which en-
25 compasses both BBR and SSFR.

## 3 Analysis and results

This section shows the results for the spectral surface albedo derivation from the irradiance data and the aircraft cam-
era imagery, as well as the comparison of broadband and
30 spectral irradiance between aircraft measurements and radia-
tive transfer calculations. The spectral mixed-scene surface albedo parametrization (described first) is used as input to the RTM calculations in the subsequent comparisons with broad-
band and spectral irradiance observations. Finally, any biases
are attributed to different sources based on their spectral fin-
gerprint.

### 3.1 Spectral surface albedo

From the simultaneous measurements of spectral down-
welling and upwelling irradiances ($F(\lambda)^{\downarrow}$ and $F(\lambda)^{\uparrow}$), the
40 surface albedo

$$\alpha(\lambda) = \frac{F(\lambda)^{\uparrow}}{F(\lambda)^{\downarrow}} \qquad (4)$$

can be derived through atmospheric correction (Appendix E) from low near-surface legs under clear-sky conditions. Clear-
sky measurements were a rare occurrence because low-level
clouds were ubiquitous. In this study, we used clear-sky mea-
surements of SSFR–BBR from 20:00:26 to 20:10:51 UTC

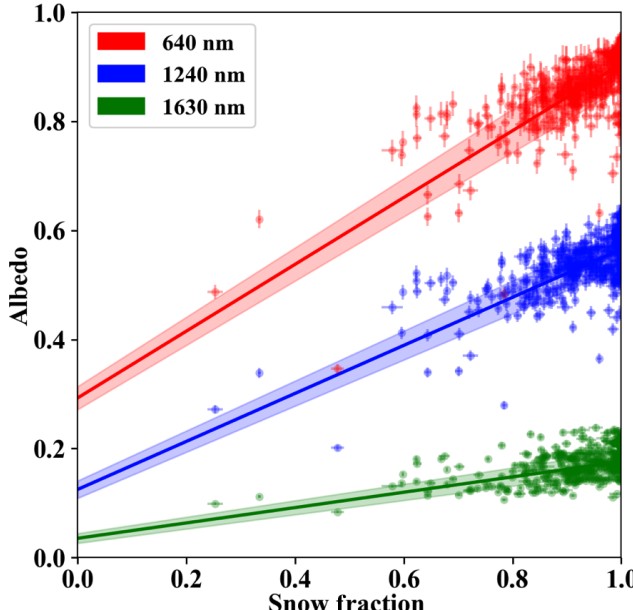

**Figure 4.** Estimated snow fraction from nadir imagery versus SSFR–BBR-measured surface albedo at 640, 1240, and 1630 nm. The surface albedo and snow fraction uncertainties are indicated as vertical and horizontal error bars. The solid lines show linear re-
gression fits, and the shaded region indicates their uncertainties.

on 13 September (referred to as "0913-clear-sky"). A time-
synchronized video of the flight is provided in the Supple-
ment (S1 "s1_flight-video_0913-clear-sky.mp4"). This video shows that the Arctic surface varied significantly – from 50
snow scenes to scenes with a large amount of dark ice. Clear-
sky scenes (no clouds above or below) were identified from the forward and nadir cameras. During the 0913-clear-sky case, the aircraft flew at an altitude at around 149 m.

To make full use of the direct measurements of the spec- 55
tral surface albedo from SSFR–BBR, we parameterized the surface albedo by snow fraction, which can be estimated from the nadir camera imagery (described in Sect. 2.3). The parameterization was done through a data aggregation technique that combines collective measurements in a par- 60
tially snow-covered environment. Figure 4 shows the surface albedo at 640, 1240, and 1630 nm plotted versus the snow fraction. The uncertainties of the surface albedo and snow fraction are indicated as vertical and horizontal error bars, respectively (details are provided in Appendix D). The data 65
showed that linear regression can be used to establish a sim-
ple relationship between snow fraction and albedo, assum-
ing that each observed spectrum is a mixture of only two so-called end-members: the spectral albedos of a dark and a bright surface. These end-members can vary depending on 70
the local conditions. For example, the dark component can be either open ocean or young ice. The bright component can be either thick ice or a snow-covered surface. The result-
ing spectral surface albedo for a mixed sampling region is

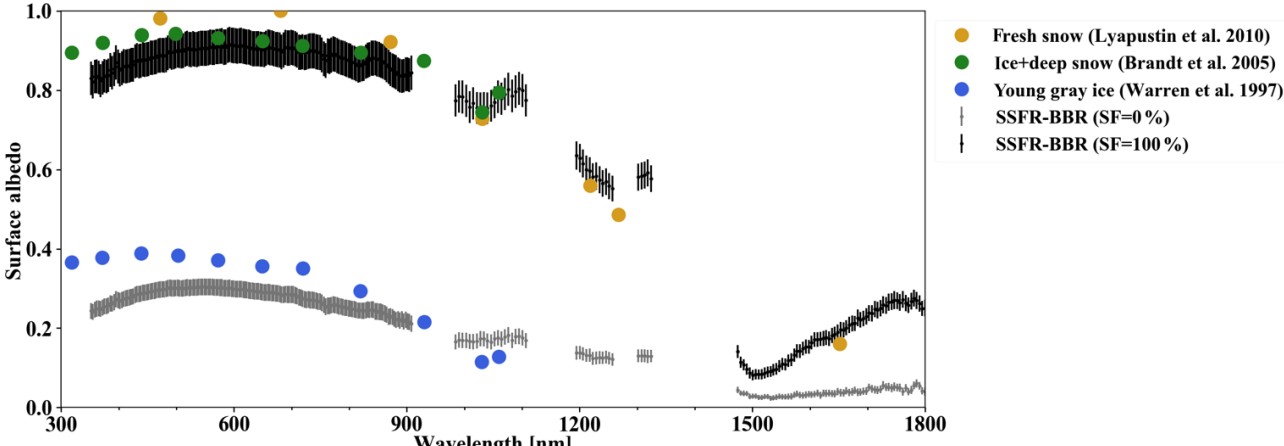

**Figure 5.** Spectral surface albedo derived from SSFR–BBR measurements for SF = 100 % (black) and SF = 0 (gray), along with their uncertainties. In addition, different albedos from the literature are shown for comparison.

established through the slopes $s_\lambda$ and intercepts $i_\lambda$ of the linear fit, with the snow fraction SF ranging from 0 to 1 as the independent variable:

$$\alpha_\lambda = i_\lambda + s_\lambda \, \text{SF}. \tag{5}$$

5 The linear regression coefficients ($i_\lambda$ and $s_\lambda$) and associated uncertainties were obtained through orthogonal distance regression (Boggs and Rogers, 1990) for all the SSFR wavelengths except for the water absorption bands and those less than 350 nm or greater than 1800 nm because of a low signal-10 to-noise ratio. We provided the coefficients in the Supplement (see S2 "s2_surface-albedo-coefficients.h5"). This simple surface albedo parameterization has obvious drawbacks; for example, the implicit linear-mixing assumption, the variability of the end-members, and data sparsity of the individ-15 ual end-members (in the example in Fig. 4, snow fractions below 0.6 rarely occur).

The snow spectral end-member (snow fraction of 1) of the mixed-scene spectral surface albedo (referred to as "2014-09-13 surface albedo") is shown in Fig. 5. The error bars 20 of the surface albedo are larger in the shortwave than in the near-infrared range. As expected, the surface albedo is high in the shortwave range from 400 to 900 nm and decreases in the near-infrared range. The SSFR–BBR-derived albedo spectra resemble the ground-based measurements of 25 thick snow over ice near Davis Station, Antarctica (Brandt et al., 2005), and they are also close to springtime aircraft measurements near Utqiaġvik – formerly known as Barrow (Alaska; Lyapustin et al., 2010). Figure 5 also shows the surface albedo with zero snow fraction. As pointed out above, 30 snow fractions below 0.6 were extremely rare during 0913-clear-sky. Nevertheless, the mixed-surface data, extrapolated to 0 snow fraction, compares surprisingly well to ground-based measurements of young gray ice, taken during the Australian National Antarctic Research Expeditions (ANARE) in 35 1996 (Warren et al., 1997). The spectra shape of the surface

albedo at 0 snow fraction (along with the nadir camera imagery from S1) suggests that during the sampled time period, the dark pixels were ice at various freezing states instead of open ocean. As mentioned above, the binary representation of surface types oversimplifies the actual mixture of ice and 40 snow but is adequate to serve as surface albedo input for the RTM to constrain the irradiance calculations over mixed surfaces, which is our primary goal here.

## 3.2 Broadband irradiance comparison

In this section, we show broadband irradiance comparisons 45 between SSFR and BBR measurements and MODIS COPs based RTM calculations at aircraft flight level for an above-cloud case (referred to as "0911-above-cloud") and a below-cloud case (referred to as "0913-below-cloud"), collected by the research flights on 11 and 13 September, respectively. 50

The RTM irradiances were calculated for wavelengths from 200 to 3600 nm. Since the SSFR–BBR-derived surface albedo described in the previous subsection was not available at wavelengths shorter than 350 nm, in gas absorption bands, and for wavelengths greater than 1800 nm due to a 55 low signal-to-noise ratio, several techniques were applied to fill in the surface albedo spectra (details in Appendix F). For both the 0911-above-cloud and 0913-below-cloud cases, the surface albedo along the flight track was calculated from SF as a driving parameter to Eq. (5). The SF was determined dif- 60 ferently for the two cases. For 0913-below-cloud, SF was obtained from the camera imagery; for 0911-above-cloud, that was not possible because the surface was not visible through the clouds, and SF was instead set to a constant value of 76.4 %, which was obtained by varying SF in Eq. (5) un- 65 til $\alpha_{1640\,\text{nm}}$ reproduced the observed clear-sky baseline for the upwelling irradiance at 1640 nm. Since SF is inferred from the albedo at a single wavelength in this case, it may reflect an "effective", rather than the actual snow fraction

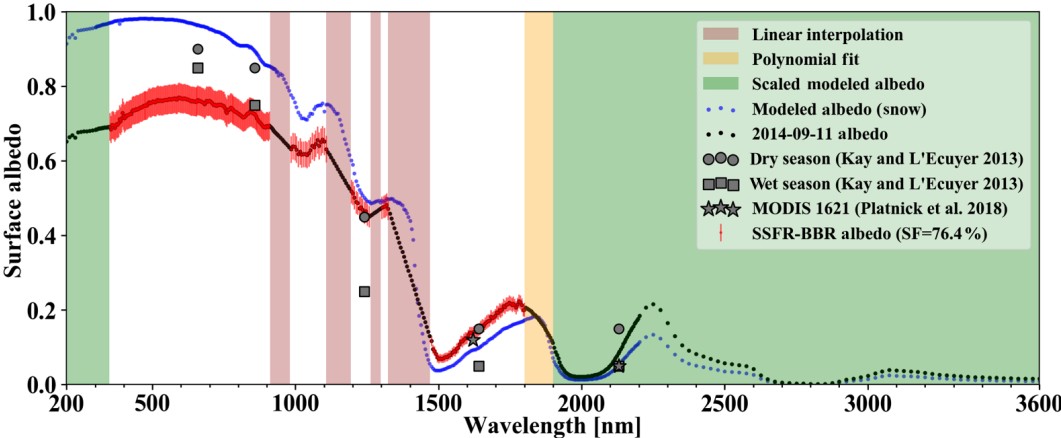

**Figure 6.** Spectral surface albedo (black) along with their uncertainties used in the RTM for the 11 September 2014 calculations. The spectral albedo uses the SSFR–BBR-derived albedo with SF = 76.4 % (red) except for the wavelength ranges marked (1) in green replaced by scaled modeled snow albedo (blue), (2) in red (gas absorption bands) linear interpolation, and (3) in yellow (1800 to 1900 nm) polynomial fit using SSFR–BBR-derived albedo from 1650 to 1800 nm.

(as obtained from the camera imagery where available), unless the spectral shape of the albedo as acquired for 0913-clear-sky matches the one of 0911-above-cloud. This is not necessarily the case. For example, changes in snow grain size between the two cases would disproportionately affect 1640 nm. It should also be noted that the observed albedo is the blue-sky albedo (direct beam and diffuse light conditions), whereas the albedo required for the cloud cases is the white-sky albedo (diffuse light only). However, MODIS-derived surface albedo in the Arctic (not shown here) shows that usually the blue-sky albedo of snow does not deviate significantly from the white-sky albedo. The difference between the two is discussed by Gardner and Sharp (2010). In addition, it is assumed that the simple parameterization as expressed in Eq. (5) holds for the whole study region. This is justified because the measurements occurred in the same general area. Figure 6 shows the surface albedo calculated for SF = 76.4 % for 0911-above-cloud. Comparing with dry- and wet-season surface albedo climatology from Kay and L'Ecuyer (2013), the wet-season climatology agrees well with SSFR–BBR-derived surface albedo in the shortwave range (wavelength less than 900 nm) except for wavelength 660 nm, where climatology has a higher surface albedo. In the shortwave near-infrared range (wavelength greater than 900 nm) however, the dry-season climatology agrees better with SSFR–BBR-derived albedo than the wet-season climatology. It is worth noting that the surface albedo assumed in MODIS 1621 cloud retrievals (Platnick et al., 2018) agrees with the surface albedo we obtained from SSFR–BBR.

Figure 7 shows the broadband irradiances from SSFR–BBR, BBR, and the calculations (Fig. 7a: downwelling; Fig. 7b: upwelling) for 0911-above-cloud, where the aircraft was flying at an altitude around 7 km. The observed variability in the downwelling signal is due to the occurrence

of cirrus above the aircraft, which is confirmed by the forward camera (Fig. 7a 1–3). In Fig. 7a, cirrus-free regions are highlighted in green. It shows that the cirrus decreases the measured downwelling irradiance by up to 10 % (40 W m$^{-2}$). However, there is no appreciable cirrus cover in the regions where low-level clouds are present. Since those are the focus of the paper, cirrus were not considered in the RTM. For the upwelling irradiance, the MODIS-derived baseline value of 230 W m$^{-2}$ corresponds to locations where MODIS did not detect any clouds. It is important to note that the value of the baseline indicates the RTM calculations under clear-sky conditions, which would change if a different surface albedo parameterization or a different snow fraction were used. For a SF = 76.4 %, the calculations agree with the measurements within 10 %. The cloud optical thickness along the flight track (included in Fig 7b) ranges from 0.5 to 15.3, with a median of 5.7, suggesting that MODIS does not retrieve clouds with an optical thickness below 0.5. In contrast to the calculations, the measurements show a continuous variation from leg to leg, suggesting that the clouds actually extended beyond the locations where MODIS detected them. Since the SSFR–BBR sensors integrate the cosine-weighted radiances hemispherically, they do not have the same field of view (FOV) as MODIS pixels. The clouds detected by SSFR–BBR but not by MODIS could therefore be caused by clouds located outside the FOV of MODIS. To take this into account, we assume a 90° (±45°) FOV for the SSFR–BBR that encompasses roughly half of the irradiance signal for an isotropic radiance distribution. When the aircraft was flying at 7 km, the FOV diameter of SSFR–BBR was 14 km (indicated as horizontal bars in Fig. 7b, translated into a time range using the aircraft speed). This is larger than the 1 km MODIS pixel-level product FOV. However, the results indicate that cloud portion missed by MODIS ex-

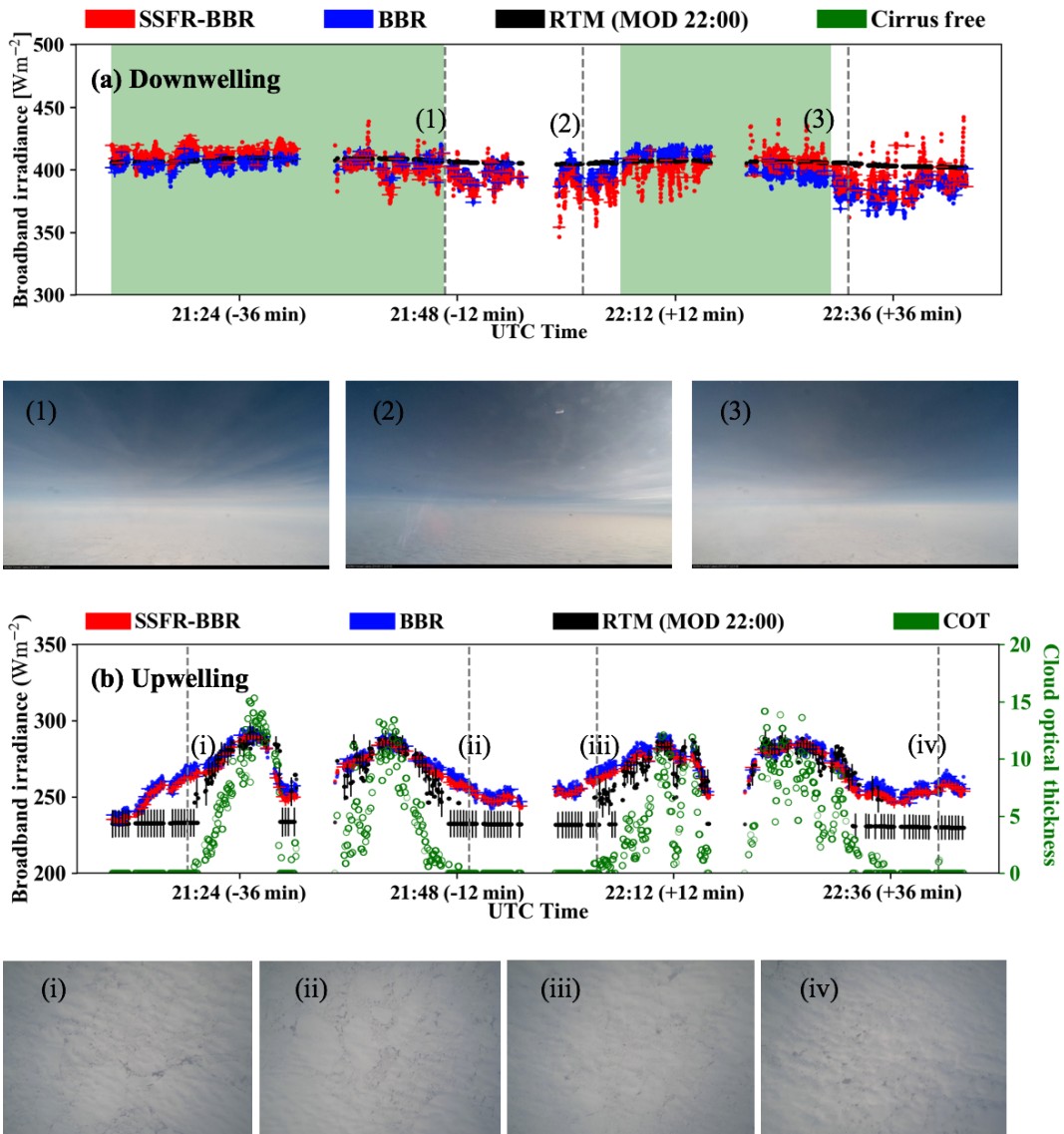

**Figure 7.** Broadband **(a)** downwelling and **(b)** upwelling irradiance from SSFR–BBR, BBR, and MODIS COPs (Terra MODIS at 22:00) RTM calculations on 11 September (above clouds) along with their uncertainties **(c)** and **(d)** the histograms. The observed irradiances include a horizontal error bar (indicating the size of the SSFR–BBR FOV) in addition to the vertical error bar (indicating the uncertainty of SSFR–BBR irradiance). The cloud optical thickness from MODIS is indicated in green. The average cloud optical thickness is 6.03. The forward camera images are provided at (1) 21:46:39, (2) 22:01:53, and (3) 22:31:05. The nadir camera images are provided at (i) 21:18:15, (ii) 21:49:22, (iii) 22:03:28, and (iv) 22:41:18 UTC. The time differences between aircraft measurements and the MODIS granule are indicated in the axis labels. The average flight altitude was 7 km and the average aircraft ground speed was $150 \, \mathrm{m \, s^{-1}}$.

ceeds the FOV of the aircraft radiometer and therefore cannot be explained by the mismatch in the observational geometry. To further corroborate that the MODIS algorithm is indeed missing clouds, a sequence of nadir camera imagery (Fig. 7bi–iv) is considered. At close inspection, the images reveal wave patterns, suggesting the existence of thin clouds in regions where MODIS does not detect any. In this case, undetected, optically thin clouds made up more than one-fifth of the points along the flight track. Figure 7b indicates that these undetected clouds lead to an underestimation of the up-

welling irradiance by $30 \, \mathrm{W \, m^{-2}}$ averaged over these pixels ($> 10\,\%$ discrepancy). Figure 8 shows the histograms of the (a) downwelling and (b) upwelling irradiance of the cirrus-free data (marked in green in Fig. 7a). Without including the data affected by the cirrus, the downwelling irradiances from RTM agree with observations within measurement uncertainty. The upwelling irradiances from the RTM show two distinct modes. The mode on the left corresponds to clear sky and the mode on the right to a range of somewhat higher reflected irradiance due to those clouds that are detected by

https://doi.org/10.5194/amt-14-1-2021

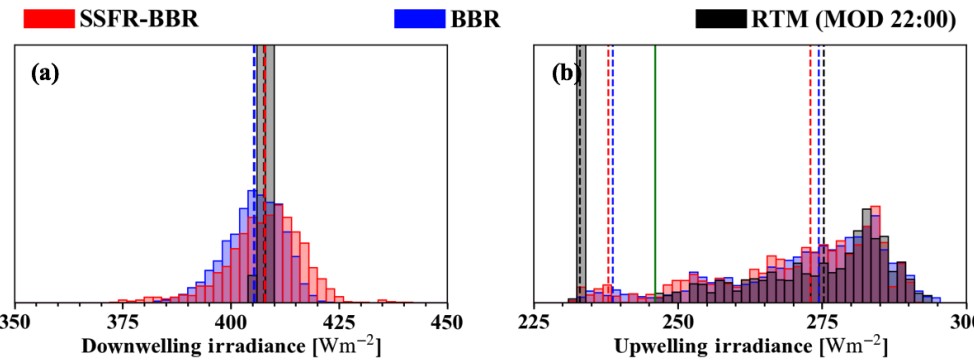

**Figure 8.** Histograms of broadband **(a)** downwelling and **(b)** upwelling irradiance from SSFR–BBR (red), BBR (blue), and MODIS COPs (black, Terra MODIS at 22:00) RTM calculations on 11 September (above clouds). Only "cirrus-free" data (marked in green in Fig. 7a) is included. For **(a)**, the mean values of BBR, SSFR–BBR, and RTM calculations are indicated by the colored dashed lines. For **(b)**, the mean is calculated for each of the two modes separated by the green line and indicated by the colored dashed lines.

MODIS. From the clear-sky mode (black dashed line on the left at $233\,\mathrm{W\,m^{-2}}$) to the thinnest detectable cloud (green line at $246\,\mathrm{W\,m^{-2}}$), there is a gap, which, according to the measurements, is actually filled with a continuum of values from thinner clouds and/or from a variable surface albedo. Because of this gap, the low bias due to undetected clouds is smaller than or equal to $13\,\mathrm{W\,m^{-2}}$. Whereas this bias caused by undetected clouds in the upwelling irradiance is almost negligible, it becomes significant for the transmitted irradiance (see below). The calculated irradiances for the locations where MODIS does detect clouds are only $10\,\mathrm{W\,m^{-2}}$ lower than the measurements (4 %), which is only slightly larger than the BBR–SSFR measurement uncertainty (3 %, see Appendix D) and can be explained by either (a) incorrect COPs (optical thickness, effective radius, or thermodynamic phase) and/or (b) inaccurate or variable surface albedo. To quantify the contributions of these effects to the total discrepancy, the spectral information from SSFR is used in Sect. 3.3.

After the investigation of the above-cloud case for MODIS-derived irradiance, we turn our attention to the below-cloud case – 0913-below-cloud, which relates to near-surface irradiance. The primary cloud layer consisted of stratocumulus cloud and was located between 0.8 and 1.2 km. The cloud optical thickness (indicated in Fig. 9a) ranges from 4.1 to 8.1, with a median of 5.8. A secondary cloud layer close to the surface, located below the aircraft's minimum flight altitude of 500 ft (approximately 150 m), frequently occurs due to a temperature inversion close to the surface, where leads and cracks in the ice provide the necessary moisture for their formation. These clouds also need to be considered to quantify the radiative surface budget, but they are excluded from the analysis here because the aircraft could not underfly them. As a result, only the data from 22:21:00 to 22:25:48 UTC (minimal occurrence of the secondary cloud layer as indicated by the forward and nadir camera imagery) were selected for comparison. A time-synced video for this flight leg is provided in the Supple-

ment (see S3 "s3_flight-video_0913-below-cloud.mp4"). As mentioned before, in contrast to the above-cloud case where the surface albedo was held constant in the RTM, the surface albedo variability on the below-cloud leg was considered here. Figure 9 shows the upwelling and downwelling broadband irradiance comparison between calculations and observations from SSFR–BBR and BBR. When incorporating the "13 September surface albedo" into the RTM, the upwelling irradiance calculations resemble the SSFR–BBR and BBR measurements (Fig. 9b). The calculations agreed well with SSFR–BBR and BBR when clouds were detected except for the time period before 22:22:48 UTC when the aircraft was entering the cloud field. The MODIS granule from Aqua was a snapshot of the cloud scene at 22:10, 10 min prior to the beginning of the flight leg. Measurement–model discrepancies for specific pixels can therefore be explained by changes of the cloud field over time. The bimodal behavior that is apparent in the time series (Fig. 9a and b) as well as in the histograms (Fig. 10) stems from time periods with and without clouds in the model input. The observations show no evidence of any cloud gap – hence only one mode appears. The "cloud gaps" apparent in the satellite but not aircraft measurements could be caused by different viewing and sun-sensor geometries between the satellite and aircraft instruments. For example, tall clouds could block the direct sun beam measured by the aircraft radiometer when flying below clouds under low-sun conditions. By evaluating the fields of cloud optical thickness and radiance at 860 nm from MODIS (Fig. 9c and d) and the flight video S3, we found that any cloud gaps are not large or frequent enough permit the direct beam to be transmitted. This leads to a smooth irradiance time series in the aircraft measurements. The gaps (circled in Fig. 9c, most likely at sub-grid scale for the 1 km product), however infrequent, do occur in the satellite retrievals. From the histograms of Fig. 10, one can estimate the pixel-level bias caused by undetected clouds. In this case, the thinnest detectable clouds are associated with $234\,\mathrm{W\,m^{-2}}$ in the cal-

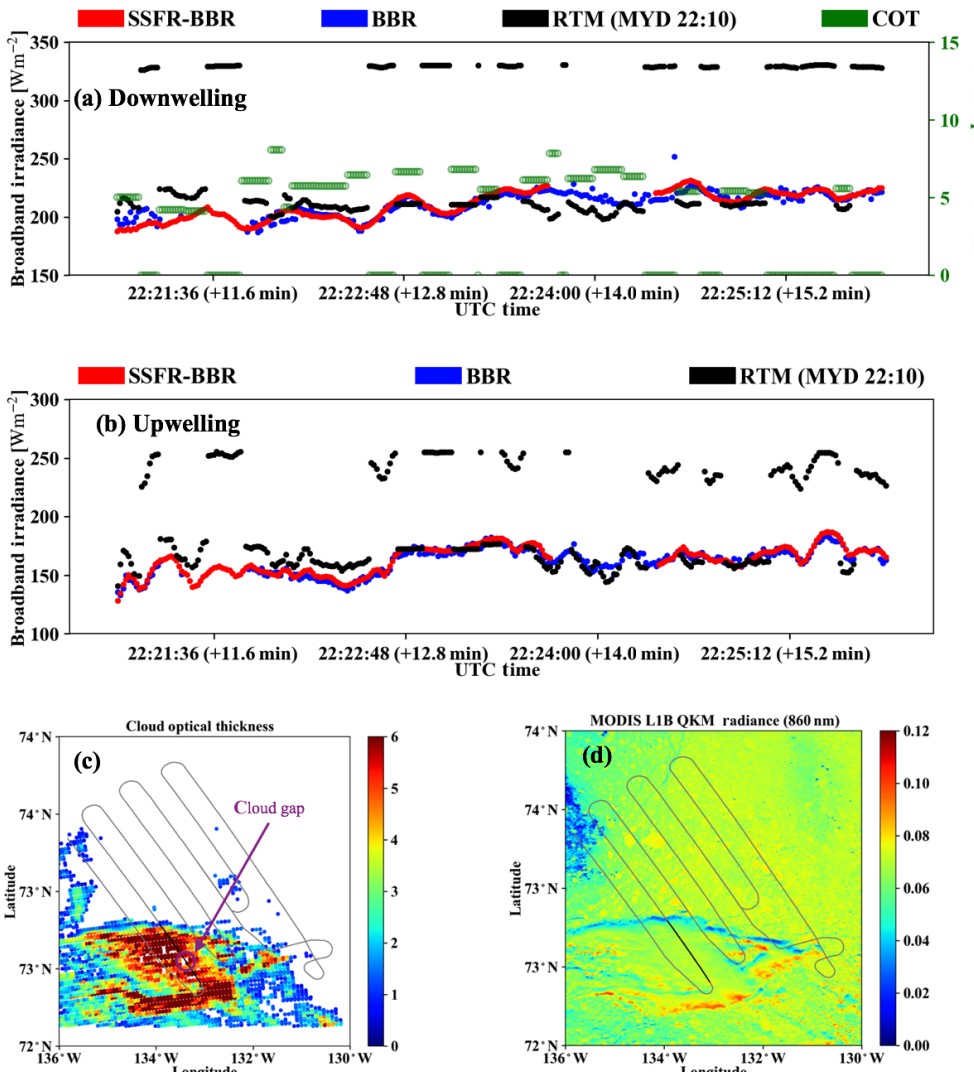

**Figure 9.** Broadband **(a)** downwelling and **(b)** upwelling irradiance from SSFR–BBR, BBR, and MODIS COPs (Aqua MODIS at 22:10) RTM calculations on 13 September (below clouds). The time difference between aircraft measurements and MODIS granule is indicated on the axis labels. In addition, the field of cloud optical thickness and radiance at 860 nm from MODIS are provided in **(c)** and **(d)**. On the map, the black line indicates the flight track studied in **(a)** and **(b)**. The average flight altitude was 235 m, and the average aircraft ground speed was $106\,\mathrm{m\,s^{-1}}$.

culations, as opposed to the clear-sky value of $330\,\mathrm{W\,m^{-2}}$ (bias of $86\,\mathrm{W\,m^{-2}}$). For the upwelling irradiance, the bias is 57 (245–188) $\mathrm{W\,m^{-2}}$, and the net irradiance high bias due to undetected clouds is therefore $29\,\mathrm{W\,m^{-2}}$.

## 3.3 Spectral irradiance comparison

Although the model–measurement biases in the broadband irradiances are negligible when clouds were detected, the time series as shown in Fig. 7b do not quite match, especially for the thin parts of the clouds near the edge of a field. To diagnose the cause, we use the spectrally resolved measurements by SSFR–BBR in this section.

For 0911-above-cloud, Fig. 11 presents the spectral upwelling irradiance comparison at 860 and 1640 nm. To put these results into context, the RTM calculations were not only performed with a surface albedo from Eq. (5) (with SF = 76.4 %), but also with climatological surface albedos of the Arctic dry and wet seasons (0.85 and 0.75) for 860 nm from Kay and L'Ecuyer (2013). As shown in Fig. 11a, the baseline of the clear-sky RTM calculations varied significantly with surface albedo. The clear-sky measurements from 21:12:25–21:15:35 UTC are slightly below the SF = 76.4 % baseline calculation for 860 nm and above for other times. It is impossible to tell whether the variability at this wavelength stems from surface albedo variability or from undetected clouds. For 1640 nm (Fig. 11b), however,

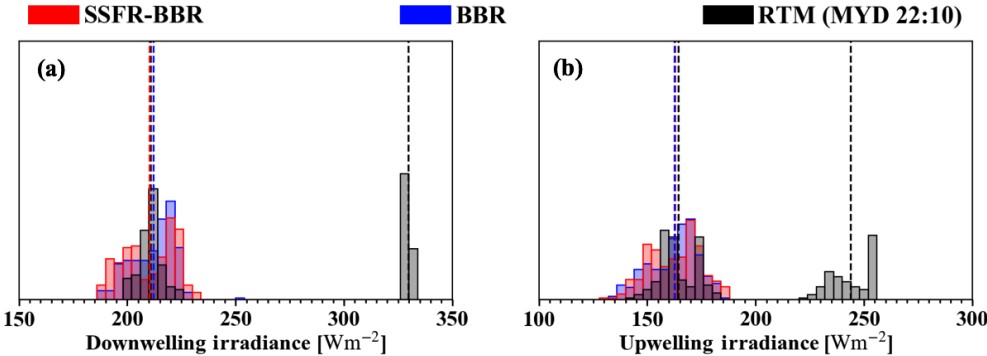

**Figure 10.** Histograms of broadband **(a)** downwelling and **(b)** upwelling irradiance from SSFR–BBR (red), BBR (blue), and MODIS COPs (black, Aqua MODIS at 22:10) RTM calculations on 13 September (below clouds). The mean value of the SSFR–BBR and BBR data is calculated and indicated by red and blue dashed lines. For the RTM calculations, the mean value is calculated for each of the two modes separated by the green solid line and indicated by the black dashed lines.

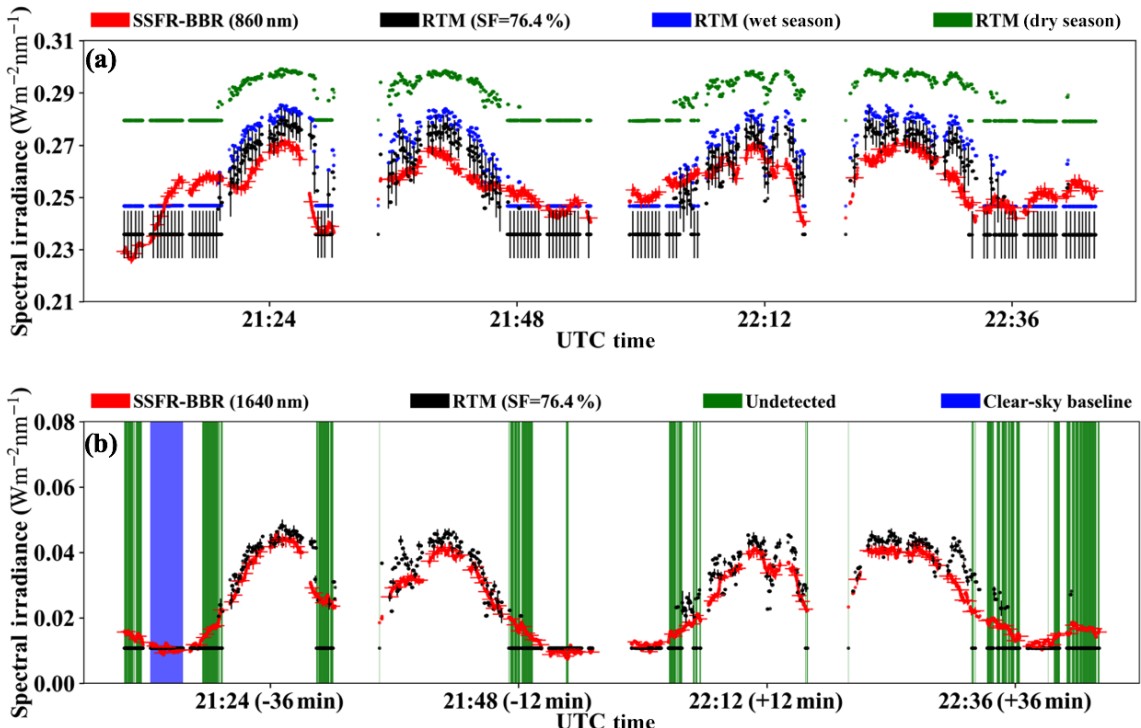

**Figure 11.** Spectral upwelling irradiance at 860 nm **(a)** and 1640 nm **(b)** from SSFR–BBR (red) and MODIS COPs RTM calculations using "13 September surface albedo" with SF = 76.4 % (black) on 11 September. In addition, calculations with climatological snow albedos are shown in **(a)** (Arctic wet season: 0.75; Arctic dry season: 0.85). The time periods where clouds were not detected are marked in green in **(b)**. The clear-sky period that was used to determine the snow fraction is highlighted in blue in **(b)**. The uncertainties of the spectral irradiances are indicated as vertical error bars, and the horizontal error bars correspond to the radiometer FOV as in Fig. 7. Both need to be considered to identify undetected clouds.

the clear-sky baseline is much more defined and less variable than other wavelengths, which is why we determined the SF value based on that wavelength.

Since any inaccuracies in the spectral surface albedo will propagate into model biases for both cloudy and clear-sky conditions, an operational surface albedo retrieval in the Arctic would be highly desirable. In this context, it is impor-

tant to note that the small broadband model–measurement discrepancy of 8 W m$^{-2}$ from Fig. 7 is only achieved when the SSFR–BBR-derived surface albedo is used in the RTM calculations; when using a climatology instead, it would be larger. In other words, in absence of an operational product, the surface albedo variability dominates the uncertainty in clear- and cloudy-sky irradiance calculations.

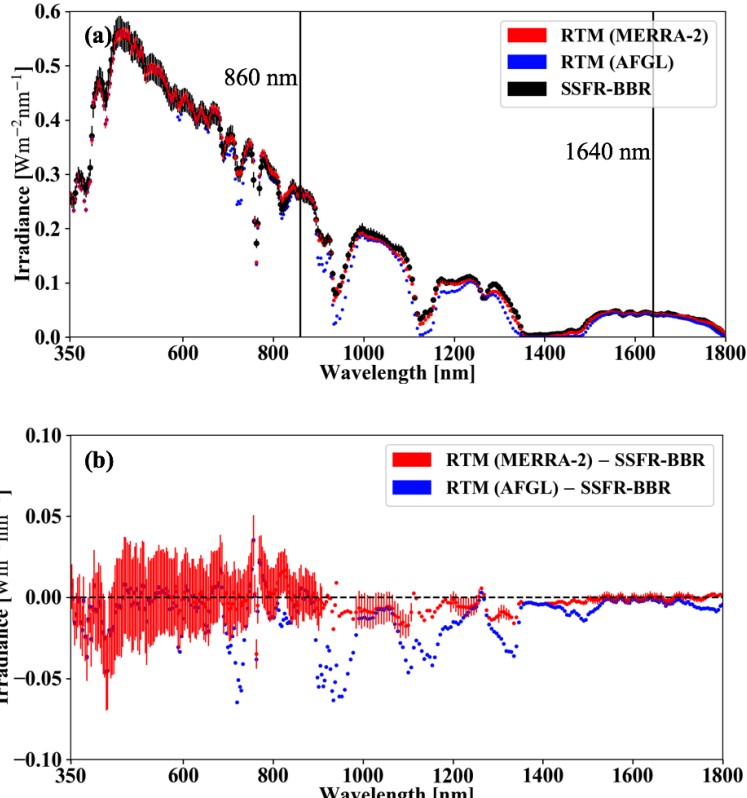

**Figure 12. (a)** Spectral upwelling irradiance from SSFR–BBR (black) and MODIS COPs RTM calculations with atmospheric profiles from MERRA-2 (red) and with AFGL subarctic summer climatology (blue) at 21:24 UTC on 11 September. **(b)** Irradiance difference between RTM and SSFR–BBR. The uncertainty of the SSFR–BBR irradiance is indicated as error bars (for one spectrum only).

At 1640 nm (Fig. 11b), there is good model–measurement agreement for the clear-sky baseline and for cloudy pixels that MODIS detects. That is because snow is dark in the shortwave infrared and because MODIS COPs in the Arc-
5 tic are primarily based on these wavelengths. Because of the obvious distinction between cloudy and clear pixels in the measurements and calculations, it is possible to estimate the fraction of partially or fully cloudy pixels that are not detected by MODIS. Among all the cloudy pixels along the
10 flight leg (i.e., pixels with clouds above or below the MODIS detection threshold), 27 % (highlighted in green) are actually cloudy even though MODIS identifies them as clear sky. One interesting finding from the broadband irradiance comparison (Fig. 7b) is that the calculations are low-biased rela-
15 tive to the observations. However, from the spectral comparison (Fig. 11), the calculations have larger (similar) values than the SSFR measurements at 860 (1640 nm). To reconcile the apparently contradictory results, we use the full spectrum from the calculations and observations at 21:24 UTC
on 11 September, when the broadband calculation indicates a 6 W m$^{-2}$ low bias.

Figure 12a and b show the spectral upwelling irradiance from the RTM calculations and from the SSFR–BBR measurements, as well as the difference between RTM and SSFR–BBR. In addition to the RTM calculations with atmo-
25 spheric profiles from MERRA-2 (referred to as RTM$_{MERRA}$), we provided the calculations with the atmospheric profile climatology (AFGL subarctic summer, Anderson et al., 1986, referred to as RTM$_{AFGL}$). The agreement between RTM$_{MERRA}$ and SSFR-BBR in the water vapor absorption
bands indicates that MERRA-2 is sufficient to prescribe the water vapor content in the calculations. The broadband irradiance difference between RTM$_{MERRA}$ and RTM$_{AFGL}$ due to water vapor is 13.5 W m$^{-2}$. Outside of the gas absorption bands, the calculations agree with the measurements
at wavelengths smaller than around 850 nm but are slightly low-biased at near-infrared wavelengths. Spectral discrepancies are caused by the use of inaccurate (1) surface albedo (2) cloud optical parameters, some of which compensate for each other in the broadband integral. Such error compensa-
tion may lead to an improved model–measurement agreement for the "wrong reasons"; therefore, validation efforts should include spectrally resolved measurements.

So far, the analysis did not reveal whether the observed model–measurement discrepancies are due to biases in the
45 COPs or in the surface albedo. Figures 13 and 14 are an attempt to disentangle both sources of uncertainty despite the limited number of observations during ARISE. Figure 13

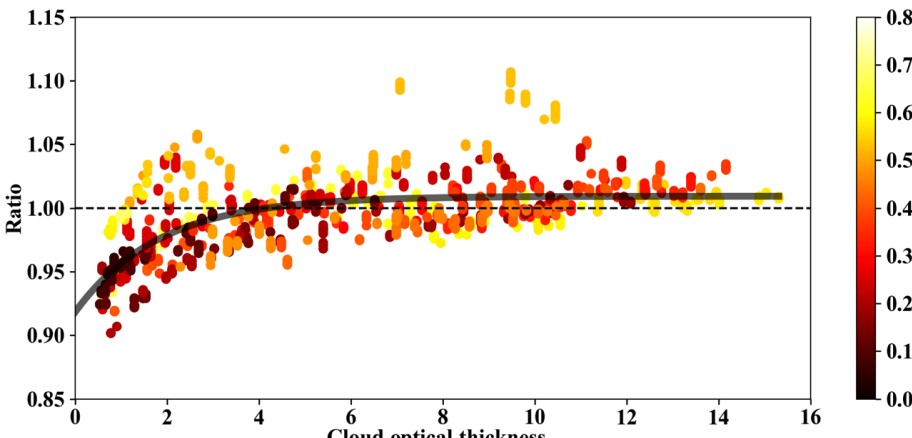

**Figure 13.** Ratio (RTM / SSFR–BBR) of upwelling broadband irradiance as a function of cloud optical thickness from MODIS 1621 cloud product on 11 September. The time differences between aircraft measurements and MODIS granule (unit: hours) are color-coded. The black curve is an exponentially fitted line using $r = a - e^{b \cdot COT + c}$, where $a = 1.0093$, $b = -0.5464$, and $c = -2.3954$.

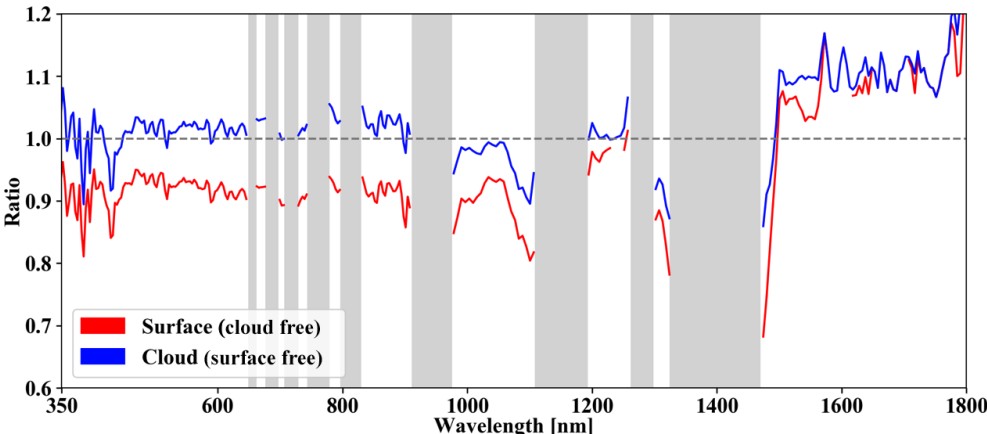

**Figure 14.** The spectrum of ratio when COT $= 0$ (red, indicating cloud-free) and when COT $= \infty$ (blue, indicating surface free) for wavelengths range from 350 to 1800 nm. The gas absorption bands are indicated in gray. Ratios at the gas absorption bands are excluded.

shows the ratio between modeled (labeled "RTM") and measured ("SSFR–BBR") upwelling broadband irradiance at flight level as a function of the retrieved COT for the collection of cloudy pixels from 11 September. At large COT, clouds dominate the upwelling irradiance, whereas the surface dominates in the limit of zero COT (as stated above, the retrieved minimum is 0.5). The ratio of RTM / SSFR–BBR can be used to indicate how biased the surface albedo is in the RTM when COT is approaching 0 and how biased the cloud optical properties are when the COT approaches large values. The data reveal a functional relationship between COT and the RTM / SSFR–BBR ratio. An exponential fitting of

$$r = a - e^{b \cdot COT + c} \qquad (6)$$

is used to parameterize the upwelling irradiance ratio as a function of COT. The black curve in Fig. 13 suggests that the surface albedo in the calculations is biased low by about 8 %, whereas almost no bias is detectable in the cloud properties ($a$ of $\sim 1.01$). Figure 14 shows the spectral fits for the wavelengths between 350 and 1800 nm. Two spectra are calculated: the spectrum of the ratio when COT $= 0$ (denoted as $r_0(\lambda)$), corresponding to cloud-free conditions, and the spectrum of the ratio at infinite COT (denoted as $r_\infty(\lambda)$), corresponding to cloudy conditions. The $r_0(\lambda)$ spectrum (red) is consistently lower than 1.0 at short wavelengths ($< 1300$ nm) and slightly greater than 1.0 for wavelengths longer than 1500 nm. This suggests that the surface albedo is underestimated for the shorter wavelengths and overestimated for the longer wavelengths. Since changing the snow fraction will only increase or decrease surface albedo for all wavelengths, simply changing the snow fraction does not improve the agreement for both long and short wavelengths. As mentioned before, the albedo we used in the RTM is the so-called blue-sky albedo, which differs from the white-sky albedo that is captured by the measured upwelling irradiance under

cloudy conditions. Instead, the discrepancies could be caused by the physical changes of the surface, different sun angles, and/or instrument performance changes. The $r_\infty(\lambda)$ spectrum (blue) oscillates around 1.0 for the shorter wavelengths and is consistently larger than 1.0 for longer wavelengths, which suggest that the retrieved effective radius is slightly biased. Unfortunately, owing to limited sampling time, the below-cloud flight (13 September) leg does not lend itself to any conclusions from a cloud transmittance perspective since it is not the same cloud field as on 11 September. In future flight campaigns, coordinated above- and below-cloud legs will furnish more information on bias analyses than possible from ARISE.

## 4 Conclusions

In this paper, we used aircraft observations to validate shortwave irradiance derived from satellite passive imagery (MODIS) of low-level cloud fields. This was done with two consecutive flights from the NASA ARISE campaign, which sampled the radiation below and above a cloud field in a similar location of the MIZ. Such validation studies are especially important in the Arctic because observations from the surface are sparse. Despite their limitations, passive imagery products are one of the essential data sources for observationally based estimates of the surface radiative flux under cloudy conditions, which necessitates a quality assessment of cloud detection and the derivation of cloud optical parameters for a variety of specific cloud and surface types as well as surface angles. In addition, accurate knowledge of the surface albedo and of the water vapor vertical distribution is required to derive the net fluxes at the surface, above the cloud layer, and at the top of the atmosphere. The two cases analyzed here only focused on one region with one specific surface and cloud type, but this allowed developing a validation approach that did help answer specific questions such as the following.

1. What is the reliability of passive imagery cloud detection in the MIZ and over solid snow-covered regions?

2. How much do undetected clouds bias imagery-derived irradiance, especially at the surface?

3. What is the relative magnitude of irradiance errors caused by undetected clouds, biased cloud properties, incorrect surface albedo parameterization, and water vapor?

This paper sheds some light on these questions using the combined measured broadband and spectral irradiance in the study region, but these results are far from representative for the Arctic as a whole. To gain a statistically based understanding, validation data from multiple experiments will have to be combined. By aggregating data from multiple missions, it should be possible to answer more general questions, which a single case study cannot address.

– Do existing cloud climatologies from spaceborne passive imagery observations accurately reproduce the frequency of low-level optically thin clouds over different surface types?

– Do existing climatologies of surface albedo capture the spatial and temporal variability sufficiently to keep errors in the derived all-sky irradiance and cloud radiative effects to an acceptable level?

It is unclear what "acceptable" would mean for the second question, but our study showed that the actual surface albedo deviates from commonly used climatologies. Throughout the Arctic, inaccurate knowledge of the surface albedo and its variability will lead to an inaccurate estimation of cloud radiative effects and net surface fluxes, even under clear-sky conditions. This is especially important in the visible part of the spectrum where most of the shortwave energy resides and where the albedo of different surface types (ice, fresh and old snow) varies significantly. Of course, knowledge of the near-infrared variability of snow and ice albedo (via grain size) is also important because it affects the accuracy of imagery-derived cloud products.

To capture the spatial and spectral variability of the surface, we developed a data aggregation technique that combines collective measurements in a partially snow-covered environment into one spectral surface albedo dataset that is parameterized by snow fraction ("binary" representation of the radiative surface properties). The dataset we obtained agrees with ground-based measurements for the two extremes (called spectral end-members): snow and thin ice. In our case, ice-free open ocean was radiatively insignificant, and the two end-members were sufficient to represent the surface variability. In more complex, more general cases, more end-members will be required.

In assessing the relative magnitude of different errors (question 3 above), we found that undetected clouds have the most significant impact on the imagery-derived irradiance. In the case studied here, MODIS did not detect clouds below a threshold of 0.5 in optical thickness, even when including partially cloud-covered pixels. For the above cloud case, this led to a low bias of up to $13\,\mathrm{W\,m^{-2}}$ for the upwelling shortwave irradiance. The below cloud case was harder to interpret due to the limited data and the lack of knowledge about the irradiances at cloud top. However, the model–measurement comparison indicated a high bias of at least $86\,\mathrm{W\,m^{-2}}$ in downwelling shortwave irradiance if clouds are not detected, which again suggested undetected clouds as the dominating error source. While the cloud reflectance (and thereby the TOA cloud radiative effect) bias is minimal (above-cloud case), the cloud transmittance bias (and thereby the surface cloud radiative effect) is significant (below-cloud case). This means that (in reality) the surface would receive less shortwave radiation than derived from satellite imagery and melt less rapidly. Of course, low-level clouds have a warming effect in the longwave. There is, in fact, some in-

dication from land observations (Greenland) that the (long-wave) warming effect from thin liquid-bearing clouds dominates (Bennartz et al., 2013). If a large fraction of clouds in the Arctic are not detected (as also reported by Wendisch et al., 2019), this would mean that the surface melt due to clouds would be significantly underestimated. In any case, the performance of passive imagery detection algorithms in the Arctic, along with shortwave and longwave flux biases, needs to be systematically studied in the future.

In addition to the bias from undetected clouds, secondary error sources are (a) surface albedo, (b) water vapor content, and (c) cloud optical properties. By using an SSFR–BBR-derived surface albedo along with atmospheric profiles from aircraft measurements and MODIS COPs in the RTM calculations, they agreed with the measured spectral and broadband shortwave irradiance within the range of uncertainties, except in regions where MODIS did not detect clouds. It should be pointed out that in absence of an operational surface albedo product, the surface albedo uncertainty by far dominates the calculated shortwave irradiance error.

While the radiation calculations at TOA can be constrained through the radiation product from satellite observations (e.g., CERES), the radiation calculations at the surface do not have such constraints. The attribution of the individual error contributions was done based on measurements from the SSFR–BBR, by distinguishing the different physical mechanisms based on their spectral dependence. Under some circumstances, the different errors compensate partially in the broadband irradiance.

Generalizing the findings from airborne studies such as these will only be possible by improving satellite remote sensing along the way, which in turn requires airborne observations for the development and validation of a new generation of cloud retrievals in the Arctic. Such retrievals (e.g., Ehrlich et al., 2017) will need to account for surface and cloud variability and address the issue of undetected thin clouds. A database of spectral albedos, acquired with similar techniques as proposed here, would provide the necessary test bed for developing operational space-based retrievals for surface reflectance as available for the lower latitudes. With lower COT thresholds for cloud detection, spatially and temporally dependent surface albedo, accurate cloud retrievals even for thin clouds, and passive remote sensing will significantly improve our current understanding of cloud radiative effects in the Arctic.

## Appendix A: Diffuse and direct correction for the polar angle response

The polar angle response ("cosine response") needs to be done separately for the direct and the diffuse downwelling radiation. Therefore, these two components first need to be separated, assuming

$$DR(\lambda) = DR_{clear}(\lambda) \cdot f + DR_{cloud}(\lambda)(1 - f),$$

where DR is the diffuse (to total, or global) ratio, $f$ is the clear-sky fraction, and $(1 - f)$ is the fraction of a diffuser (clouds).

We can make the simplification $DR_{cloud}(\lambda) = 1.0$, (i.e., the radiation under clouds does not have a direct component), leading to

$$DR(\lambda) = 1 - f \cdot (1 - DR_{clear}(\lambda)). \tag{A1}$$

The SPN1 measures the broadband diffuse radio, which we denote as $DR_{SPN1}$:

$$DR_{SPN1} = \frac{\int_{\lambda_1}^{\lambda_2} DR(\lambda) \cdot F^{\downarrow}(\lambda) d\lambda}{\int_{\lambda_1}^{\lambda_2} F^{\downarrow}(\lambda) d\lambda}, \tag{A2}$$

where $\lambda_1$ and $\lambda_2$ indicate the wavelength range of SPN1, and $F^{\downarrow}(\lambda)$ is the calculated downwelling (global) spectral irradiance from a RTM (we did not use the SSFR measurements because they only encompass a subrange of the SPN1). Substituting Eq. (A1) into Eq. (A2), we get

$$DR_{SPN1} = 1 - f \cdot \frac{\int_{\lambda_1}^{\lambda_2} F^{\downarrow}(\lambda) \cdot (1 - DR_{clear}(\lambda)) d\lambda}{\int_{\lambda_1}^{\lambda_2} F^{\downarrow}(\lambda) d\lambda}. \tag{A3}$$

We can then determine $f$ from

$$f = \frac{(1 - DR_{SPN1}) \cdot \int_{\lambda_1}^{\lambda_2} F_{clear}^{\downarrow}(\lambda) d\lambda}{\int_{\lambda_1}^{\lambda_2} F_{clear}^{\downarrow}(\lambda) \cdot (1 - DR_{clear}(\lambda)) d\lambda}, \tag{A4}$$

and the diffuse / direct ratio can be calculated by using this value of $f$ in Eq. (A1).

## Appendix B: Azimuth response

The azimuth response of the SSFR zenith light collector was obtained using the data collected during the so-called calibration flight (2 October 2014), where the aircraft flew a circle to collect radiation measurements at different solar azimuth angles. This was done by referencing the SSFR irradiance measurements to the simultaneous BBR data, building on the fact that unlike SSFR, BBR had no discernable azimuthal dependence. The data used to determine the azimuth response had a solar zenith range of [68.24°, 71.49°] with an average of 70.20°, whereas the solar zenith angle range for the above-cloud case (11 September 2014, where the azimuth correction was applied) was [68.46 to 71.89°] with the mean of 68.91°.

Since SSFR only covers part of BBR's bandwidth from 200 to 3600 nm, RTM calculations were used to fill in SSFR spectra beyond its nominal wavelength range of 350–2050 nm. Subsequently, the RTM-extended SSFR irradiance was spectrally integrated (referred to as $F_{SSFR}$). A second-order Fourier series was then applied to fit the azimuthal dependence captured by the ratio $F_{SSFR}/F_{BBR}$, shown in Fig. B1. It shows this ratio as a function of reference azimuth angle, defined as the azimuth angle of the sun with respect to the light collector, for which 0° is defined as the aircraft flying due north. A second-order Fourier series was applied to fit the azimuthal dependence of $F_{SSFR}/F_{BBR}$. It constitutes SSFR's azimuthal response at this solar zenith angle, which was then used to correct SSFR's downwelling irradiance for the conditions encountered for the SSFR data collected during other research flights. The azimuth response obtained in Fig. B1 can be expressed as (with coefficients)

$$\begin{aligned}
\frac{F_{SSFR}}{F_{BBR}} = {} & 0.9460 + 0.0647 \cdot \cos\left(\frac{\phi}{180} \cdot \pi\right) \\
& + 0.0160 \cdot \sin\left(\frac{\phi}{180} \cdot \pi\right) \\
& - 0.0045 \cdot \cos\left(\frac{\phi}{180} \cdot 2\pi\right) \\
& - 0.0015 \cdot \sin\left(\frac{\phi}{180} \cdot 2\pi\right), \tag{B1}
\end{aligned}$$

where $\phi$ is the reference azimuth angle.

## Appendix C: Adaptive thresholding

The threshold value at each pixel location of the image depends on the neighboring pixel intensities $I$. For a pixel located at $(x, y)$, the threshold value $T(x, y)$ is calculated through the following steps.

1. A subdomain of size $d \times d$ is selected with $(x, y)$ at the center of the subdomain.

2. The weighted average $C(x, y)$ is calculated for the subdomain using Gaussian weights (Davies, 2005) $W(x, y)$, $C(x, y) = \sum_{i=0}^{d} \sum_{j=0}^{d} I(i, j) \cdot W(i, j)$.

3. The threshold for the pixel at $(x, y)$ is the difference of the weighted average calculated in the previous step and a constant $C_0$, $T(x, y) = C(x, y) - C_0$.

$d$ and $C_0$ are input parameters that can be adjusted to improve the results. In this study, $d$ is set to 1501 and $C_0$ is set to 0.

## Appendix D: Uncertainty estimation

### D1   SSFR–BBR irradiance product

For the SSFR spectral measurements, the nominal radiometric uncertainty is 5 % (Schmidt et al., 2010). The nominal

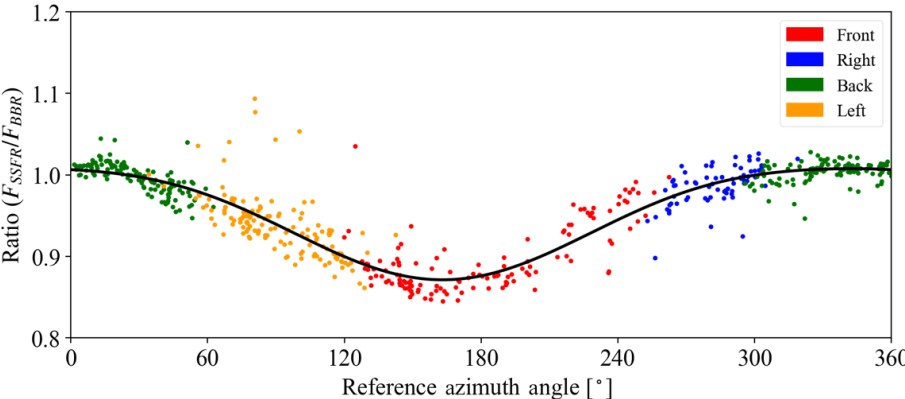

**Figure B1.** Ratio between spectrally integrated SSFR downwelling irradiance and broadband downwelling irradiance from BBR as a function of reference azimuth angle (solar azimuth position with respect to the sensor, 0° pointing north) during 01:00–01:36 UTC on 3 October 2014. The relative positions of the sun with respect to the aircraft are indicated by different colors. The black curve is a fitted function using a second-order Fourier series.

uncertainty of the BBR measurements is 3 % (Smith et al., 2017). As we described in Sect. 2.2, we corrected the azimuthal dependence of the SSFR downwelling irradiance based on the BBR measurements using the method described in Appendix B. After the correction, the SSFR downwelling and upwelling irradiances are still slightly inconsistent with BBR due to an imperfect cosine response comparing to BBR (although they agree with BBR within the range of uncertainty). In addition, the different sun-sensor geometries between the calibration flight (2 October 2014) and the 0911-above-cloud case mean that the azimuthal response as measured during the calibration flight does not necessarily fully apply to the case under study. In order to reference SSFR to BBR and simultaneously estimate the uncertainty of the merged product (SSFR–BBR), we applied a scaling method as shown in Fig. D1. Figure D1a and b show the azimuthally corrected SSFR downwelling and SSFR upwelling irradiance versus BBR. The wide spread of downwelling irradiance indicates that even after applying azimuthal correction for SSFR, some residual uncertainty of the azimuthal response obtained in Appendix B remains in the SSFR measurements. In the upwelling irradiance, the SSFR is more closely related to BBR. Figure D1c and d illustrate how we correct for the remaining biases between SSFR and BBR and estimate the uncertainties of the SSFR–BBR product. Figure D1c and d show the histogram of the ratio of the SSFR and BBR measurements. The ratio histograms indicate a scale factor of 1.006 and 0.946 for the SSFR downwelling and upwelling, with SDs of 0.025 and 0.01 when referencing to BBR. The scale factors of 1.006 and 0.946 are applied as divisors to SSFR downwelling and upwelling irradiance, respectively. The SSFR irradiance after scaling (referred to as SSFR–BBR) versus BBR is shown in Fig. D1a and c in green. After scaling, the SSFR–BBR and BBR achieve a better consistency. The SDs of 0.025 and 0.01 represent the precision for the downwelling and upwelling irradiance

of SSFR–BBR. Thus, we use 2.5 % ($0.025/1.006 \cdot 100$ %) and 1 % ($0.01/0.946 \cdot 100$ %) as the precision estimates for SSFR–BBR downwelling and upwelling, whereas the uncertainty propagates from BBR into the SSFR–BBR product (3 %).

## D2 Snow fraction and surface albedo

When calculating the surface albedo from SSFR–BBR using Eq. (4), we use the precision as determined above because the uncertainty cancels out for the ratio between the upwelling and downwelling irradiance. The uncertainty estimate of 2.7 % for the surface albedo $\alpha$ is then obtained through error propagation using Eq. (4), where

$$\frac{u(\alpha(\lambda))}{\alpha(\lambda)} = \sqrt{\left(\frac{u(F(\lambda)^{\uparrow})}{F(\lambda)^{\uparrow}}\right)^2 + \left(\frac{u(F(\lambda)^{\downarrow})}{F(\lambda)^{\downarrow}}\right)^2}. \quad \text{(D1)}$$

The uncertainty of the snow fraction described in Sect. 2.3 is estimated based on two main sources of error:

1. angle of the field of view (FOV) defining the circular area of the image pixels that were selected for processing;

2. subdomain size $d$ specified in the adaptive thresholding method described in Appendix C.

When the FOV size gets too large, the vignette correction from Eq. (2) leads to a bias stemming from peripheral pixels where the effect is no longer linear. When the FOV size gets too small, we would lose the variation in the snow fraction due to a relatively small area. To use as many pixels as possible while avoiding the inclusion of contaminated pixels due to vignette effect at the corners, we found the best FOV angle to be 140°. In addition, changing the subdomain size $d$ would slightly change the results. Thus, we obtained five sets of snow fraction estimates using a FOV angle of 120, 140,

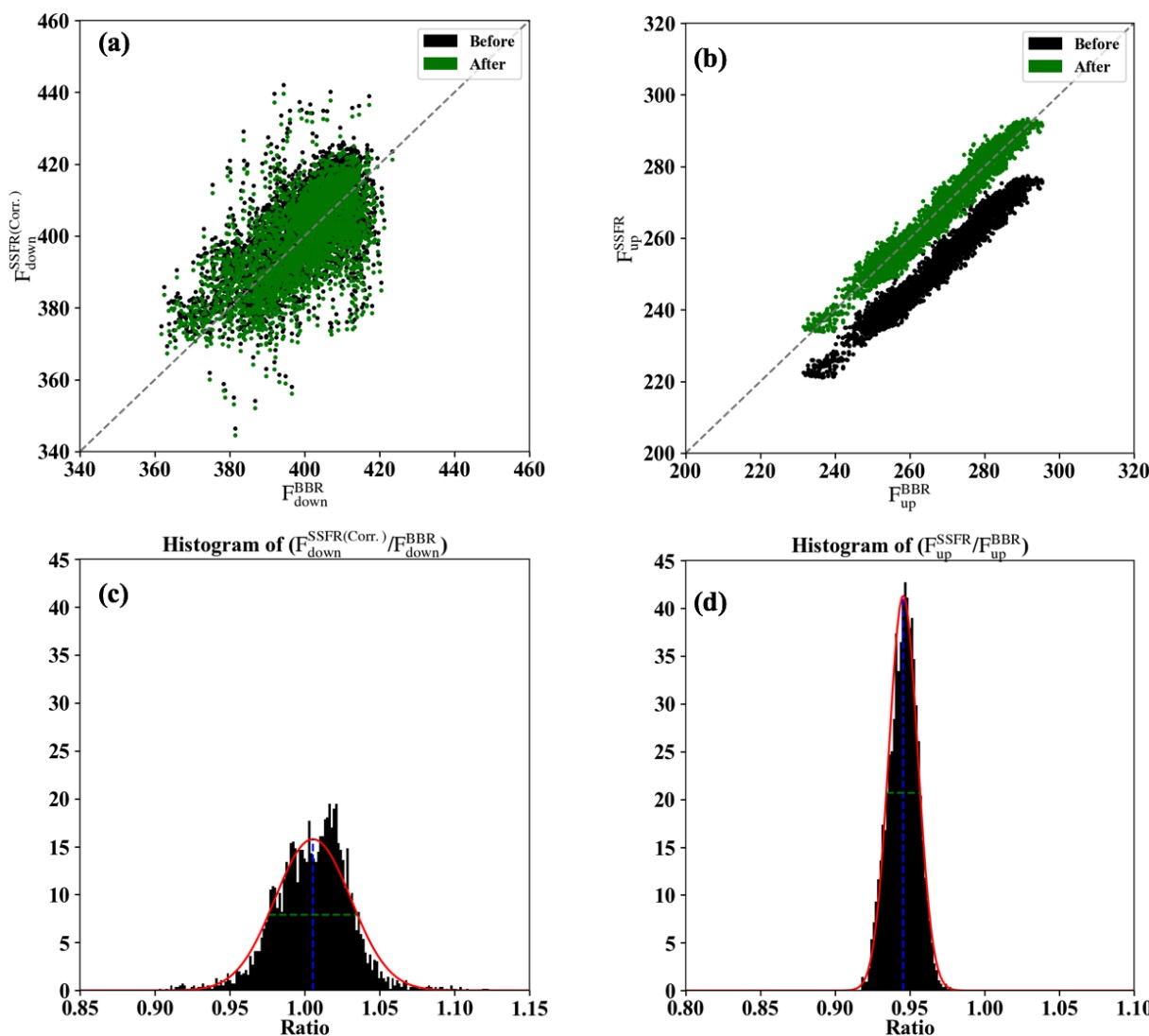

**Figure D1.** SSFR integrated broadband irradiance versus BBR broadband irradiance (**a**: downwelling; **b**: upwelling) and the histograms of the ratio of SSFR integrated broadband irradiance to BBR broadband irradiance (**c** and **d**) for the 0911-above-cloud case. The mean and the full width at half maximum of the Gaussian distribution of the ratio are indicated as blue and green dashed lines in the histogram plots. The SSFR–BBR data (SSFR after applying the scale factor as indicated by the blue dashed line) versus BBR is indicated in green in (**a**) and (**b**).

and 160° and a subdomain size $d$ of 1401, 1501, and 1601. Figure D2 shows the five sets of snow fractions estimated from nadir camera images using before-mentioned FOV angles and subdomain sizes. The SD of the five sets of snow fraction is used as the uncertainties for the snow fraction for each data point.

### D3 Radiative transfer calculations

The uncertainty of the radiative transfer (RT) calculations for the 0911-above-cloud was estimated through the two-stream approximation of the reflectance $R$

$$R = \frac{\tau + \alpha \cdot \left(\frac{2\mu}{1-g}\right)}{\tau + \left(\frac{2\mu}{1-g}\right)}, \tag{D2}$$

TS1 where $\tau$ is the cloud optical thickness, $\alpha$ is the surface albedo, $\mu$ is the cosine of the solar zenith angle, and $g$ is the asymmetry parameter. A value of 0.85 is assumed for $g$. In addition, we assume that the two main sources for the uncertainty are from the cloud optical thickness $\tau$ and surface albedo $\alpha$. The uncertainty of $R$ due to the change of $\tau$ and $\alpha$

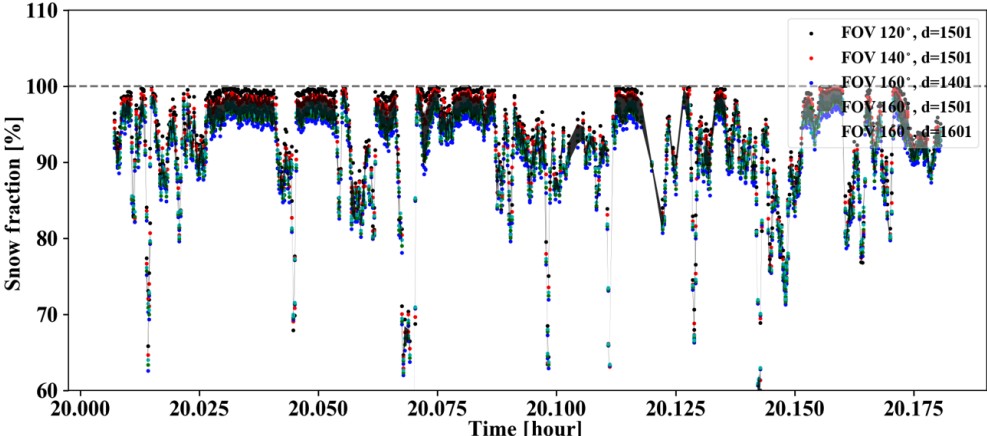

**Figure D2.** Snow fraction estimated using different FOV angles and subdomain sizes in the adaptive thresholding. The SD of the five sets of snow fraction is shaded in black.

is therefore

$$u(R) = \sqrt{\left(\frac{\partial R}{\partial \tau} u(\tau)\right)^2 + \left(\frac{\partial R}{\partial \alpha} u(\alpha)\right)^2}. \tag{D3}$$

This analytical formula allows the calculation of uncertainties without numeric radiative transfer calculations.

## Appendix E: Atmospheric correction

The following steps describe the atmospheric correction applied to the flight level albedo measured by SSFR–BBR.

1. The spectral flight level albedo from SSFR–BBR (referred to as $x_0$) was scaled by 0.6, 0.7, 0.8, 0.9, and 1.0 (referred to as $y_1$, $y_2$, $y_3$, $y_4$, and $y_5$) – each of these are spectra.

2. Five sets of downwelling and upwelling irradiances were obtained from the RTM by changing surface albedo to $y_1$, $y_2$, $y_3$, $y_4$, and $y_5$ while keeping the other model inputs the same.

3. From the five sets of downwelling and upwelling irradiances calculated at flight altitude, we can derive five corresponding flight level albedo values $x_1$, $x_2$, $x_3$, $x_4$, and $x_5$ using Eq. (4).

4. The five pairs of $\{x, y\}$ provide a relationship between surface albedo and flight level albedo (nearly linear), $y = ax + b$.

5. The linear relationship was inverted to infer the surface albedo spectrum from the measurements at flight level $(ax_0 + b)$.

The atmospheric correction corrected less than 0.2 % on flight level albedo at the non-absorbing wavelengths.

## Appendix F: Extending spectral surface albedo

To obtain the spectral surface albedo for a wavelength range from 200 to 3600 nm, several techniques were performed. Using the spectral surface albedo for 0911-above-cloud (Fig. 6) as an example, the following steps were followed:

1. The spectral surface albedo was calculated from Eq. (5), e.g., with SF = 76.4 % (marked in red in Fig. 6).

2. In the gas absorption bands (red area in Fig. 6), the surface albedo was replaced with interpolated values.

3. From 1800 to 1900 nm (yellow area in Fig. 6), a polynomial fit was used for extrapolation, based on the spectral dependence from 1650 to 1800 nm.

4. For the wavelengths shorter than 350 nm and greater than 1900 nm (green area in Fig. 6), a modeled snow albedo (Wiscombe and Warren, 1981) was used, multiplied with a scale factor to match the measurements at the joinder wavelengths.

*Data availability.* MODIS data were provided by the NASA Goddard Space Flight Center's Level-1 and Atmosphere Archive and Distribution System (LAADS) (https://doi.org/10.5067/MODIS/MOD06_L2.061, Platnick et al., 2017a).

*Video supplement.* Synchronized flight videos for ARISE (Arctic Radiation – IceBridge Sea&Ice Experiment) (https://doi.org/10.5281/zenodo.4029241, Chen and Schmidt, 2020). Each video includes (1) C-130 flight track overlay MODIS RGB imagery obtained from NASA Worldview, (2) video imagery from the aircraft forward camera, (3) video imagery from the aircraft nadir camera, (4) time series of irradiance measurements at 500 nm from SSFR if available, (5) spectral irradiance measurements from SSFR if available, and (6) temperature measurements from KT-19 if available.

*Supplement.* The supplement related to this article is available online at: https://doi.org/10.5194/amt-14-1-2021-supplement.

*Author contributions.* HC processed the SSFR data, performed the analysis, and wrote the majority of the paper with input from the other authors. KS collected the SSFR data; helped with the methodology development and data analysis; and helped with developing, writing, and editing the paper. MK and GW helped with the MODIS data processing and interpretation. AB and ER provided the BBR data. MSR advised on the usage of aircraft measurements during ARISE. WS was the PI of the ARISE campaign and provided MODIS false color imagery. PT helped with the validation of Arctic sea ice albedo. SK helped with the interpretation of the CERES product in the Arctic. PP helped with the collection of SSFR data. All the co-authors helped in the reviewing and editing of the paper.

*Competing interests.* The authors declare that they have no conflict of interest.

*Financial support.* This research has been supported by the National Aeronautics and Space Administration (grant nos. NNX12AC11G and NNX14AP72G).

*Review statement.* This paper was edited by Manfred Wendisch and reviewed by three anonymous referees.

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

**Remarks from the typesetter**

TS1  Please give an explanation of why this needs to be changed. We have to ask the handling editor for approval. Thanks.