# Peer review of "The Effect of Low-Level Thin Arctic Clouds on Shortwave Irradiance: Evaluation of Estimates from Spaceborne Passive Imagery with Aircraft Observations"

_Atmospheric Measurement Techniques, 2019_

## Referee Comment (RC1) · Anonymous Referee #1 · 26 Nov 2019

General comments

This paper describes an analysis of solar broadband and spectral irradiance data from an airborne measurement campaign in the Arctic in September 2014. Comparisons were made with radiative transfer model calculations (RTM) that take into account concurrent cloud products from MODIS satellite observations and retrievals of local spectral surface albedos from the aircraft measurements. A significant fraction of the work is concerned with the determination of the spectral ground albedos taking into account measured upwelling and downwelling irradiances, video images from the ground and literature data in spectral ranges where measurements were not feasible. The effort is

justified because under the Arctic conditions, accurate ground albedos are crucial to distinguish cloud radiative effects from the influence of the relatively bright snow- or ice covered surface.

In general, the agreement of measurements and RTM calculations is satisfactory for both broadband and spectral measurement (at two selected wavelengths). However, significant differences were observed during some periods where no clouds were present according to the satellite data. This is taken as a hint towards the presence of undetected, optically thin clouds. However, in my view this interpretation is not as clear as currently presented in the paper and the influence of experimental uncertainties requires more attention. Nevertheless, the paper is well written, structured and documented. It should be published after minor revision.

Specific comments

Page 2, line 10: Clarify "radiative transfer model" instead of "model"

Page 2, line 15: "...about 22% of clouds remained undetected (cloud optical thickness less than 0.5)." This statement is not in agreement with the main text, see page 11, line 36.

Page 3, line 14: " ...every $10^6$ km$^2$ decrease... a 2.5 W m$^{-2}$ increase". The statement is unclear. For which area does this apply?

Page 4, line 36: Give more information on the flight area, altitudes and times e.g. in the text, in a table or a modified Fig. 1.

Page 5, line 16: The broadband instruments were probably not actively aligned. That should be made clear. Were data excluded from the analysis when the aircraft attitude was not horizontal?

Page 5, line 25: "... that keeps the zenith light collector horizontal..." I assume that both collectors were actively aligned. If so, that should be clearly stated because for the determination of net irradiances this is certainly important.

Page 5, line 28: "... that cannot be corrected." Another reason why low sun eleva-
tions are more challenging for the downwelling direct irradiance is that the gradient
d cos(SZA) / (d SZA) increases with SZA.

Page 6, line 9: "... atmospheric effects". You probably mean "collector effects"?

Page 6, line 25: I am convinced that the complicated collector-specific corrections and
calibrations were done thoroughly. But can you estimate the remaining uncertainties
for the upwelling and downwelling measurements, both broadband and spectral? For
example in Appendix B, Fig. 13 the scatter in the ratios indicates uncertainties of the
independently calibrated instruments that will not vanish by applying the azimuth cor-
rection. I assume Fig. 13 shows downwelling irradiances? If so it would be interesting
to see a similar plot for the upwelling (with similar scatter but no azimuth dependence).

Page 6, line 36, Eq. 1: The different weighing implies a conversion to "brightness" or
"luma" rather than a (relative) physical radiance. The source of the three coefficients
should be cited and what they represent.

Page 7, line 15: "...robust estimates". I wonder what robust means. The procedure
is quite complicated and there are obviously several sources of uncertainties: (1) the
coefficients in Eq. 1, (2) the blending technique Eq. 2, (3) the adaptive thresholding
(parameters given in Appendix C) and (4) the (presumably) limited field of view of the
camera compared to the irradiance collector. For example, does it make a significant
difference if 0.333 is inserted in Eq. (1) for R, G, and B? It would be convincing if you
could provide an uncertainty estimate for the snow fraction based on a sensitivity study
taking into account the different aspects (1)-(4).

Page 8, lines 12 and 13: Exchange hygrometer and thermometer in 3) and 4)

Page 8, equation (4): Why not define "$\alpha_{SSFR} =$" here?

Page 9, line 6: I assume the atmospheric correction was comparatively small because
the altitude was below 300 m? You should give the reader a rough idea.

Page 9, line 33: To support this statement you could include the (much lower) albedo of the open ocean in Fig. 5. Moreover, in the introduction you mention a climatological surface albedo for the region by Moody et al., 2007 that is used for the MODIS cloud retrievals. It would be interesting to see how this compares with the locally measured data.

Page 10, line 9: Why 70%? In Fig. 4 a majority of data points is well above 80%.

Page 10, line 21: "cloud optical thickness"

Page 10, line 21: "...agree with measurements within 10%" Does this statement refer to data in Fig. 7b or to other flight periods? In Fig. 7b agreement within 10% under no-cloud conditions is only visible at the very beginning. My impression is that good agreement during periods where no clouds were detected could be reached by increasing the snow fraction to about 80% without significantly affecting the upwelling in the presence of clouds. So the 70% snow fraction assumption and its uncertainty are crucial here and should be discussed.

Page 10, line 23: "... below 0.5" In Fig. 7b it seems that the missing about 30 W m$^{-2}$ upwelling RTM irradiance (above the clear baseline) correspond to a missing COT of about 2 which is well above the threshold of 0.5. Moreover, if the COT were indeed around 2 no ice-structures on the ground would be visible in the photographs (i)-(iv).

Page 10, line 23: The "... continuous variation from leg to leg..." Compared to the COT data the measured peaks in the upwelling irradiance look smoothed out. I wonder if this has to do with the fields of view of the instruments which at 7 km altitude are much greater than that of the satellite (1 km). So even if you fly above a correctly detected cloud gap, the sensors could receive irradiance from surrounding, even distant cloud fields. So, overall wonder if the results in Fig. 7 can be explained without "undetected clouds", i.e. by field of view effects and an underestimated albedo.

Page 11, line 3: "... via the snow fraction" What was the mean snow fraction during

that flight?

Page 11, line 7: "...except...22:22:48" I cannot find that period in the figure.

Page 11, line 11: "... no evidence of any cloud gap". This also hints towards different fields of view of satellite and aircraft. Even if you fly underneath a cloud gap, at estimated solar zenith angles of around 70 deg no direct radiation (and a corresponding increase of irradiance) may be present. It may be detected at other places (also underneath a cloud) but the chances for that are low if the cloud gap is small and cloud thickness is substantial. It would be helpful to estimate the size of the cloud gaps detected by the satellite. Moreover, a statement should be included whether or not the cloud top and bottom heights were correctly retrieved from the satellite data on both days.

Page 11, line 15: Table 1 and 2 contain a main result of this study that is not adequately discussed here and should clearly enter the abstract and conclusions: Under conditions when clouds are detected the numbers in the first three columns agree, at least (I assume) within experimental uncertainties which need to be specified.

Page 11, line 23: Do September 11 and 13 belong to the Arctic dry or wet season?

Page 11, line 36: "Of all pixels along ... 22%" To determine the fraction of undetected clouds as stated in the abstract, green/(cloudy +green) should be calculated.

Page 11, line 37: "... (highlighted in green) are actually cloudy." Again the question arises if during the green periods clouds were undetected by the satellite or radiation was captured unintentionally by the irradiance sensors at some distance from the aircraft.

Page 12, line 19: "... surface albedo ... is biased low by about 9%". The 9% come from the albedo + atmospheric correction which is probably significant at 7 km altitude.

Page 12, line 24: "Simply changing the snow fraction does not improve the agreement..." I am not so sure. Increasing the snow fraction will lift the RTM irradiances for shorter wavelengths more strongly than for the long wavelengths (Fig. 4).

Page 13, line 31: "Undetected thin clouds (COT<0.5) led to a high bias…." I don't think this statement is justified. Looking at Fig. 8 the measured downward irradiances vary around 200 W m$^{-2}$ and COT vary around 6 (where detected) with consistent RTM results. I assume a COT of 0.5 would produce downward irradiances well below the clear ones but also significantly greater than measured. So the measurements are inconsistent with the presence of thin clouds which makes an explanation as given above (page 11, line 11) more likely.

Page 13, line 32: "… above clouds …" should be "…below clouds…" See Fig. 8 and page 11, line 13.

Table 1 and Table 2: Please indicate in the captions place, time and altitude. Specify above or below cloud conditions and the unit of the numbers. Explain what is listed in the last column called "RTM" The precision of the numbers implies an accuracy that is unrealistic. I assume that the numbers in the first three columns agree within experimental uncertainties but that needs to be specified. Please state the average COT, cloud top and bottom heights.

Fig. 1: Specify the maps' latitude and longitude ranges, e.g. in the caption. A costal line can be vaguely recognized showing that the flight area was west of Banks Island but I assume hardly any reader is familiar with the area.

Fig. 2: 80.70% implies a precision that is certainly not justified by the method (see comment in text). Indicate the field of view of the circular areas in (a), the flight altitude, time and location. The y-axis label should read "spectral flux density" or better "spectral irradiance" in accordance with the main text.

Fig. 3: Approximate times and locations should be specified in the caption. The indicated flight levels are confusing here without additional information.

Fig. 4: Given the spectral resolutions, the indicated wavelengths are too precise. 640,

1240 and 1630 nm, as stated in the text, are appropriate.

Fig. 5: Consider including open ocean albedo and data by Moody et al., 2007 (see comment in text).

Fig. 7 and Fig. 8: Indicate the flight altitudes in captions and maybe the cruise speeds so that the size of the cloud gaps can be inferred.

Fig. 13: Please indicate that these are ratios of downwelling irradiances (I assume).

Appendix B, Fig. 13: Were the solar zenith angles during this flight comparable to those during the other flights? The azimuth dependence may change with solar zenith angle for geometrical reasons and because of a varying (wavelength dependent) contribution of direct irradiance.

Appendix C: In order to understand the meaning of the factor d=1501 the images' total pixel dimension should be stated.

Technical corrections

Page 4, line 13: Introduce "CRE" as "cloud radiative effects"

Page 9, line 23: "2014-09-13"

Page 11, line 31: Fig. 7?

Page 11, line 37: Fig. 7?

Fig. 5: Wet and dry season colours are hard to distinguish. Second citation should be Brandt et al.

Fig. 7 and Fig. 8: y-axis: broadband irradiance. Typo "MOODIS" in Fig. 8

Fig. 9 and Fig. 10: y-axis: spectral irradiance

---

## Referee Comment (RC2) · Anonymous Referee #2 · 12 Dec 2019

**1 General remarks**

The manuscript analyses airborne radiation observations and satellite observations of Arctic clouds. A surface albedo parametrization is derived to account for the inhomogeneous Arctic sea ice surface. Spectral and broadband are compared to radiative transfer simulations which are based on satellite observations. The spectral irradiance is analyzed to untangle uncertainties resulting from the surface albedo and the cloud optical properties.

In general, the analysis of airborne observations in remote Arctic areas is of high value

and provides one rare tool to validate satellite observations. Therefore, the study has high potential and is within the scope of AMT. It could have a wide scientific interest and might contribute to improve our understanding of Arctic clouds. However, the manuscript lacks in several major issues and therefore, does not exhaust its full potential. These issues have to be reassessed in detail before publishing the manuscript.

First, the objective of the study is not well presented and outlined. Based on the title and introduction, the readers expectations and the presented analysis may strongly differ. This deficiency might results from a non-existing description of a general approach and methodology how the measurements can and will be used to validate and improve satellite observations. Such a general strategy is an important part of the manuscript in order to promote future application of the methods. Based on the unclear objectives also the conclusions are weak and leave many questions unanswered putting off the reader with promises for future studies. Finally, a throughout uncertainty estimation is missing, which is mandatory if observations are used for validation purposes. I'm sure, there are options to restructure and improve the manuscript in a way that it presents the full potential of the study.

Below, I compiled a list of comments which have to be considered in a revised version of the paper. There might be some contradictory statements which result from my misinterpretation of the text when first reading the manuscript. I am sure the authors will know how to weight in such cases and how to improve the text to avoid misinterpretations by other readers.

**2  Major comments**

**2.1  Unclear objective of the study**

After reading the title and the introduction it is somehow unclear, what the manuscript aims to obtain: irradiance or the cloud radiative effects. The introduction does not match the title. The analysis and methods presented in the study also show not what was promised in the title:

**"Shortwave radiative effect"** was calculated and discussed only briefly. Most of the analysis concentrates on irradiances. The introduction does not give an overview on how cloud shortwave radiative effects are commonly derived. In the analysis CRE is only discussed in two sentences. Neither the method and uncertainties are introduced nor are the values discussed. This does not justifies the title of the manuscript.

**"Imagery-derived irradiance":** To me, this implies, that camera images are used and integrated into an irradiance. Or at least, that the irradiance is directly derived from images. That's not done in the manuscript and also not at all covered by the introduction. What the authors did is a parametrization of the surface albedo based on the sea ice fraction, which was observed from a camera. So I suggest to remove the word "imagery-derived" from the title.

This misleading title leaves the reader searching for the actual objectives of the study. Unfortunately, also the motivation given in begin of the manuscript does not fit to what finally was achieved. E.g.:

**"Validation of CERES-MODIS derived irradiance":** In section 2, the authors state that one objective is to validate CERES-MODIS derived irradiance. This is confusing after reading the title and introduction. CERES irradiance is a different story compared to estimating the CRE. And I also do not see a CERES product in the study. The authors theirself state, that the design of the measurement strategy failed to compare

to the CERES product. MODIS retrieval and own radiative transfer simulations are applied. However, on Page 13 line 36 the authors conclude, that CERES observations are used to constrain the observations. This was not done. To avoid confusion, I suggest to remove CERES form the argumentation. Still, comparing irradiances is not the same as estimating the CRE.

**"longwave radiative effects":** There is one section in the introduction on the longwave effect of water vapor on the surface radiation budget. But the title of the manuscript suggest, that the study is on solar effects only. So longwave radiative effects by water vapor is kind of irrelevant. The manuscript also does not include a study on the radiative effects of the water vapor profiles. Only the pure profiles are discussed.

After all, I had the feeling that the manuscript shows a potpourri of separate analysis, without a clear major goal. This probably is not true, but the manuscript requires a more clear objective. From what I read, the study aims for a closure study, which validates the MODIS cloud product by airborne observations of irradiance. Could this be the major aim? Or do you aim identifying radiative processes in the Arctic atmosphere related to surface-cloud interaction?

I think it is important to clarify the main objective of the manuscript and concentrate on the major aspects needed to achieve these goals. If the aim is a closure study, then I suggest to remove the estimation of cloud radiative effects, which currently is misleading. Or at least shift the calculation of CRE to the end of the study, after the irradiances have been compared. This would allow to extend the validation for CRE based on the uncertainties/conclusions which have been found already before when comparing the irradiances.

**2.2 Methodology needs to be outlined**

Several comparisons of different quantities (albedo, irradiance above, below clouds) are shown in the manuscript. However, it is not always clear what the purpose of each individual step is. The general and also the specific methodology of the analysis should be outlined. In the conclusion the authors write about "developing a validation approach". I don't see a clear validation approach in the study. If there is a strategy, then this needs to be outlined precisely in the begin of the manuscript. Maybe any schematic showing the different comparisons broadband, spectral, flight one, flight two might help.

It is also confusing, how and when the data of both flights is used. It took long until I understood, that the two flights provide different observations (high vs. low flight altitude). I suggest, that the authors clearly report, what is different between both flights. Why two cases are needed and how the observations are mixed/combined in the study?

Similarly, the motivation of section 3.1 was missing and leaves many open questions when reading the section. These questions should be addressed before starting the analysis:

What is the purpose of this analysis and of the parametrization of surface albedo with snow fraction? I only can guess. Wasn't surface albedo measured directly with BBR and SSFR? Where is the need to parameterize surface albedo if albedo is measured anyway?

As the surface albedo properly is an input to the radiative transfer model, section 3.1 should be presented before explaining the radiative transfer simulations.
[Figure]

**2.3   Only limited conclusions**

The conclusion section does contain a lot of "may"s and "if"s. More questions are raised than answered. The authors theirself are hesitant to draw conclusion: "sheds some light on these questions", "the actual surface albedo may deviate from commonly used climatologies", which is more than obvious. Also the limitations of the limited data set for conclusions is acknowledged. Based on these little new results, the entire section, especially the last part of the conclusion read more like an outlook, indicating, that the study did not improve much. It is not mandatory to make big improvements, but I also do not see any method, approach on how to improve all the issues that are summarized in the conclusion section. To improve the manuscript it would be helpful to present and discuss a method or approach of how to perform validation studies based on similar measurements as shown in the study. As mentioned by the authors, there is potential to process more data from the airborne campaign. To do so, a clear approach with step A, B, C,... should be presented in the manuscript.

**2.4   Uncertainty Analysis**

The study aims comparing measured irradiance with simulations. As the data is intended for use in a comparison study, a discussion of the the measurement uncertainties is fundamental. No uncertainty ranges are indicated in the plots.

BBR: What about the accuracy of the data? What is considered in the data processing? Are the BBR instruments actively levelled? The SSFR is levelled. What makes this for a difference comparing SSFR and BBR?

The same holds for SSFR. What are the final uncertainties? How the radiometric calibration contributes to the uncertainties? What is more important, correcting the angular response or tracking changes of the radiometric sensitivity over time?

**P10 L13:** There are several sources of uncertainty in the determination of the surface albedo. How the estimation of the surface albedo affects the uncertainty of the final results/study?

Also extend the discussion of uncertainties by the MODIS cloud retrieval. Only undetected clouds have been considered so far. What about cloud phase, the second thin cloud layer, surface albedo assumed in the retrieval?

**3  List of specific comments**

**P2 L9:** From the abstract it is not clear, why two independent estimates of the surface albedo (from SSFR/BBR and from the camera imagery) are needed?

**P2 L16:** How large is the radiative effect of the non-detected clouds? This is an important value when MODIS misses a significant faction of clouds.

**P3 L14:** The study by Hartmann and Ceppi (2014) does not fit to the topic of cloud radiative effects. Direct radiative effects by sea ice loss has nothing to do with clouds unless you argue, that the expected increased cloud cover over increased area of open water is not able to compensate the reduced reflection of solar radiation by the surface.

**P3 L17:** Explain acronyms CERES-EBAF, 2BFLXHR-LIDAR.

**P4 L2:** This presented state of the art on spectral albedo of Arctic surface types is very pessimistic and does not consider recent publications which cover a lot more data also derived from airborne observations (areal and temporal variability):

Perovich, D. K., Tucker, W. B., and Ligett, K. A.: Aerial observations of the evolution of ice surface conditions during summer, J. Geophys. Res., 107, SHE24-1–SHE24-14, https://doi.org/10.1029/2000JC000449, 2002.

Malinka, A., Zege, E., Heygster, G., and Istomina, L.: Reflective properties of white sea

ice and snow, The Cryosphere, 10, 2541–2557, https://doi.org/10.5194/tc-10-2541-2016, 2016.

Malinka, A., Zege, E., Istomina, L., Heygster, G., Spreen, G., Perovich, D., and Polashenski, C.: Reflective properties of melt ponds on sea ice, The Cryosphere, 12, 1921–1937, https://doi.org/10.5194/tc-12-1921-2018, 2018.

**P4 L29:** Fairbanks is in the center of Alaska. Where did you fly over Arctic sea ice?

**Figure 1:** Add longitude and latitude.

**P6 L22:** Were cloud properties derived from the SSFR measurements? If not, I suggest to remove this statement here.

**P6 L26:** What is the resolution (number of pixel) of the camera? What type of lens is used (distortion-free?)?

**P7 L4:** Instead of using such an interpolation technique, could you determine the vignetting effect by measuring over a white almost lambertain surface? I could imagine, that a snow covered surface could provide this as a first approximation. For your application this should be sufficient. Or use a certified diffuse reflector.

**P7 L5:** "Black" means probably "dark" like the dark signal of a non-illuminated camera sensor? In terms of radiation I would prefer "dark". Black is a color and limited to visible wavelength.

**P7 L5:** How the 2D matrix was determined? Each camera and lens system must have an individual matrix.

**P7 L15:** Can you discuss the retrieved snow fraction of the example and the uncertainties/quality of the method in this section? Only referring to the figure is not sufficient.

**P7 L16, Figure 2:** Figure 2a shows the presence of thin gray ice, which is not detected as sea ice in Figure 2b. This means, that from a physical view, the sea ice fraction is underestimated. Although, optically these areas are less bright, they have a higher

reflectivity and might bias your results.

Can you give an uncertainty estimate, how the sea ice fraction will change with adjusting the threshold between bright and dark pixel?

As the camera provides RGB images, it should also be possible to classify different ice types following the methods describes by Perovich et al., 2002. Did you thought about this?

Perovich, D. K., Tucker, W. B., and Ligett, K. A.: Aerial observations of the evolution of ice surface conditions during summer, J. Geophys. Res., 107, SHE24-1–SHE24-14, https://doi.org/10.1029/2000JC000449, 2002.

**P7 L30:** For altitudes above 6.5 km, a standard atmospheric profile is used. Aren't there any radio soundings available? Barrow? What about dropsonde releases from the aircraft?

**P8 L3:** It would be helpful to include a figure showing a time series or similar plots of the MODIS COPs which are extracted along the flight path. Just to know, what range of COPs have been present and how variable the cloud field was. What about temporal offsets between MODIS and airborne observations?

**P8 L4:** All clouds are assumed to be liquid. Is there any prove for this? In situ observations? The temperature profiles are well below $0\,^{\circ}$C where mixed-phase clouds typically are often present.

**P8 L11:** What quantities are included in the atmosphere profile? I usually understand also temperature and humidity to be part of the atmospheric profile, but there are provided separately.

**P8 L16:** What albedo is assumed here?

**P8 L17:** Specify or provide the slit function in the instrument description.

**P8 L19:** Does MODIS provide cloud base?

**Section 2.5:** The description of the radiative transfer simulations should be separated from this MODIS section. The title of the section does not suggest that it will include the methodology of how the solar irradiance is derived. I suggest to add a separate section "methodology". See general comments.

**P10 L8:** Why the albedo for 11 September needs to be calculated/constructed? I'm lost ... If you have the ice fraction, why there is now surface albedo measurements?

**P10 L9:** I was wondering, why you use "snow fraction" instead of the more common "sea ice fraction" or "cover". Likely because there is dark snow-free sea ice. Can you elaborate the term "snow fraction" more clearly in section. 2.3. This would help the readers to understand immediately, that there is a difference to sea ice fraction and why this is relevant.

**P10 L17:** How the cirrus is considered in the analysis? How strong does it influence the final results? Especially with respect to the proposed CRE of clouds?

**P10 L23. Figure 7b:** There is a large mismatch for areas where no clouds are detected. How strong, the undetected clouds can change the irradiance? I guess, the difference in the "cloud-free" areas is more due to the surface albedo (sea ice fraction) than due to the clouds. $\tau = 0.5$ would not make much difference over bright snow surface.
In optically thick areas, the agreement is better. How strong e.g. 10% uncertainty of the sea ice fraction would influence the results here? Can you rule out any change of the sea ice fraction along the flight track?

**P10 L33-37:** These general details of the cloud conditions that have been present during the two flights is needed much earlier. I also suggest to add a comparison of the differences between both observed cloud cases. Otherwise, it is hard to follow the analysis.

**P10 L40:** So far it was not clear, that for the first case, the albedo was fixed. This

needs to be made clear at the begin of the analysis. What are the differences between both cases?

**P11 L2:** I don't understand, why also for this case the parametrization is used. Isn't the parametrization based on the same data? What are the advantages of this approach?

**P11 L14:** As I understood, the two days are different in a way, that once the aircraft flew above the cloud layer and once below. This means, that the CRE is defined differently for both cases. So how this is accounted here?

**Section 3.2.** What is the benefit of having both BBR and SSFR broadband irradiance here? Why SSFR was integrated to broadband values and compared to BBR? This makes only sense, if e.g. surface albedo is once considered spectrally and once with a fixed broadband value? Or is there any other purpose?

**P11 L18:** Where the "model-measurement biases in the broadband shortwave CRE" are discussed? The different values in Table 1 and 2 have not been explained. How the different CRE are calculated? How CRE is derived based on measurements and how it is derived based on the simulations? If there are estimates based only measurement, you would need cloud free flight sections, which I do not see in the data.

How the cirrus layer is considered in the estimation of the CRE and the radiative transfer simulations? You can not neglect the cirrus.

**P11 L21:** Upward or downward irradiance?

**P11 L25:** I don't understand, why you need a climatological surface albedo, when you measured and parameterized the albedo.

**P11 L28:** Was the surface albedo fixed or varied in the simulations or do I misunderstood this sentence? That a fixed albedo will not represent reality is more than obvious. That's why I don't fully understand the approach to use a fixed albedo here. How is this motivated?

[Figure]

**P11 L29:** Specify. Range of what?

**P11 L31:** Climate models? How you can draw this conclusion? No climate model is applied, analyzed or discussed in the manuscript. Even when a climate model would have been considered, the "underestimation" may only hold for your specific case, where the albedo is assumed to be to high. This must not hold for the climate models. Further, there are climate models available, which use sophisticated snow albedo parametrization accounting for different sea ice types, melt ponds, snow etc. E.g.:

Dorn, W., Rinke, A., Köberle, C., Dethloff, K., and Gerdes, R.: HIRHAM–NAOSIM 2.0: The upgraded version of the coupled regional atmosphere-ocean-sea ice model for Arctic climate studies, Geosci. Model Dev. Discuss., https://doi.org/10.5194/gmd-2018-278, 2018.

**P12 L17:** Please write the equation using the symbols of the quantities which are calculated here, not x, y.

**P12 L19:** The behaviour shown in Fig. 11 can be explained by the change of surface albedo between cloud-free (direct Sun illumination) and cloudy conditions (diffuse illumination). A Similar behavior is reported by Gardner and Sharp (2010). This means, that you albedo is not necessarily wrong. It depends on what you want. A closure study comparing the irradiance in cloudy conditions required the cloudy-sky albedo. For estimating the CRE, the cloud-free albedo needs to be applied in the radiative transfer simulations.

Gardner, A. S., and Sharp, M. J. ( 2010), A review of snow and ice albedo and the development of a new physically based broadband albedo parameterization, J. Geophys. Res., 115, F01009, doi:10.1029/2009JF001444.

**P12 L20:** Figure 11 and the parametrization is shown for broadband quantities. Figure 12 is calculated spectrally resolved but used the broadband parametrization? Shouldn't the parametrization be computed for each wavelength?

**P12 L20:** "Spectrum" of what? Upward downward irradiance or albedo?

**P12 L20:** What is $x$? COT? If yes, then write COT. Also a bracket is missing in line 21.

**P12 L23:** "remarkable agreement". Without uncertainty estimation you can not judge about an agreement.

**P12 L33:** If the major aim of the study is to show how good irradiances or the CRE can be derived from the MODIS cloud product, the most general scenario should be considered as well. The general case is, that you do not have any airborne observations. Which means, you have to rely on the surface albedo product of MODIS. Atmosphere profiles from reanalysis, etc.. This would be the routine/operational approach. Based on that, you may try to improve the approach by exchanging different assumption with the airborne observations, such as measured surface albedo.

**P2 L38:** You did not discuss deeply the uncertainties of the MODIS retrieval. The comparison only shows, that there are differences, but could you use the observations to constrain which COPs have been retrieved wrong and to what degree?

**P13 L1:** The impact of water vapor profiles was not shown in the manuscript.

**P13 L3:** "developing a validation approach": I don't see a clear validation approach in the study. If there is a strategy, then this needs to be outlined precisely in the begin of the manuscript and summarized here.

**P13 L15:** It seems, that the study presented here could not answer any of the questions written in this section. I suggest to start the conclusion section with conclusions and not rise more questions than have been pointed out in the introduction.

**P13 L35:** Define "excellent"! Why you consider the agreement excellent? Any objective measure to judge this? Excellent compared to what? This requires an uncertainty estimation.

**Appendix A:** Can you briefly explain the concept behind these calculations? Is this

done to derive the diffuse fraction based on the present cloud cover? What about broken clouds? Is this covered by the approach?

**P15 Eq 1:** Avoid the large slash.

**P15 Eq 2:** This equation was already given above.

**P15 L24:** How the boundaries of the image are treated if the subdomain is that large?

**Table 1 and 2:** What unit have the numbers? Is this only solar radiative forcing or total (solar + terrestrial)?

**Figure 4:** Can you provide the parametrization equation (linear regression?) and the regression coefficients. This might be useful for other research studies.

**Figure 8:** a) There are data points behind the text. c) Add a legend. And use two labels c) and d) for the left and right panel. In the caption there is a typo in "MOODIS".

**Figure 10a:** The label hides some of the data.

**Figure 10:** Indicate the wavelengths used in Fig. 9.

**Figure 10b:** I suggest to remove the ratio here as the absolute irradiance is almost zero in the range of water vapor absorption. It could be better to show absolute differences instead of a ratio. If the comparison should be linked to the broadband irradiance, the absolute difference will be integrated and result in the difference of the broadband irradiance. Short: 5% difference at 500 nm is more important for broadband than 5% difference at 1600 nm.

**Figure 12:** Why $\tau = 10000$ and not infinity? Please use $\tau$ instead of $x$. Also here: Do you need to include the water vapor bands wavelengths and have a y-axis down to zero?

**Figure 13:** Provide equation and coefficients of the fit. Not necessarily in the figure, but in the text.

---

## Referee Comment (RC3) · Anonymous Referee #3 · 23 Dec 2019

The paper presents a method to validate irradiances derived from satellite observations by aircraft irradiance observations.

The authors calculate irradiances using a radiative transfer model with cloud optical properties derived from MODIS observations as input. These modelled irradiances are compared to airborne observations of broadband irradiances (BBR) and shortwave spectral irradiances (SSFR). From aircraft observations, the spectral surface albedo is also derived, which is besides the cloud optical properties a crucial parameter for the calculation of (spectral) irradiances. On the other hand, if the surface albedo is not well-known, the retrieval of optical properties (particularly optically thin clouds) from

satellite observations is highly uncertain.

The presented results for two case studies show, that the airborne observations are consistent (i.e. BBR and SSFR yield the same broadband flux), thus the radiometric calibration of the two instruments is consistent. However, differences between airborne irradiance observations and satellite-derived irradiances are found. The main differences emerge from pixels, where clouds are not detected in the MODIS images (cloud optical thickness < 0.5). Otherwise the agreement was surprisingly good, the reason might be that different errors compensate during spectral integration.

The paper is generally well written and the results are clearly presented by appropriate figures. However, it should be stated more clearly that the paper presents a methodology rather than a thorough validation (see also comment below). The topic fits well into the scope of AMT, therefore I recommend publication after minor revisions.

General Comments:

- The title is misleading, since the cloud radiative effect of arctic low-level clouds is not discussed much in the paper. Also, the imaginary derived irradiances are not evaluated, because only two case studies are presented. In the conclusions it is stated that a statistical analysis of a large dataset would be required for evaluation but this has not yet been done. An appropriate title could be e.g. "A method to evaluate shortwave irradiances derived from satellite images of low-level arctic clouds with aircraft observations".

- Abstract: "This study suggests that passive imagery cloud detection could be improved through a multi-pixel approach, that would make it more dependable in the Arctic." -> Cloud detection methods are not discussed at all in the paper, thus is sentence should not be in the abstract.

- What is the uncertainty of the airborne observations? Please include error bars in the plots. If this is not possible, discuss the sources of errors in the text and provide a

rough estimate.

Technical corrections:

p4, l11: "radiative calculations" -> "radiative transfer calculations"

p5, l5: "satellite-based radiative transfer calculations" -> "irradiances calculated using an RTE model with input from satellite data ..."

p6 Eq.1: Reference for numbers used to convert RGB to grayscale

p11, l37: Fig. 6b -> Fig 7b

---

## Author Comment (AC1) · 20 Apr 2020

See supplement.

Please also note the supplement to this comment:
https://www.atmos-meas-tech-discuss.net/amt-2019-344/amt-2019-344-AC1-
supplement.pdf

---

## Author Comment (AC2) · 20 Apr 2020

See supplement.

Please also note the supplement to this comment:
https://www.atmos-meas-tech-discuss.net/amt-2019-344/amt-2019-344-AC2-supplement.pdf

---

## Author Comment (AC3) · 20 Apr 2020

The page and line numbers used in the response are referring to the revised manuscript, which is appended to this response. Note that this is not the final revised manuscript, but is provided to show the reviewers where we made changes to address their concerns.

**Anonymous Referee #3**

**The paper presents a method to validate irradiances derived from satellite observations by aircraft irradiance observations.**

**The authors calculate irradiances using a radiative transfer model with cloud optical properties derived from MODIS observations as input. These modelled irradiances are compared to airborne observations of broadband irradiances (BBR) and shortwave spectral irradiances (SSFR). From aircraft observations, the spectral surface albedo is also derived, which is besides the cloud optical properties a crucial parameter for the calculation of (spectral) irradiances. On the other hand, if the surface albedo is not well-known, the retrieval of optical properties (particularly optically thin clouds) from satellite observations is highly uncertain.**

**The presented results for two case studies show, that the airborne observations are consistent (i.e. BBR and SSFR yield the same broadband flux), thus the radiometric calibration of the two instruments is consistent. However, differences between airborne irradiance observations and satellite-derived irradiances are found. The main differences emerge from pixels, where clouds are not detected in the MODIS images (cloud optical thickness < 0.5). Otherwise the agreement was surprisingly good, the reason might be that different errors compensate during spectral integration.**

**The paper is generally well written and the results are clearly presented by appropriate figures. However, it should be stated more clearly that the paper presents a methodology rather than a thorough validation (see also comment below). The topic fits well into the scope of AMT, therefore I recommend publication after minor revisions.**

**R:** Thank you for your comments. We did indeed intend to categorize our paper as a hypothesis (case) study rather than as a systematic validation approach. The objectives of this study are to 1) quantify the discrepancies between observed and satellite imagery (MODIS) derived irradiance for a limited number of cases 2) identify the key error sources of the discrepancies using the broadband and spectral measurements. To make this more clear, we added description for each step of the approach at **Line 5, Page 5** to avoid confusion.

**General Comments:**

- The title is misleading, since the cloud radiative effect of arctic low-level clouds is not discussed much in the paper. Also, the imaginary derived irradiances are not evaluated, because only two case studies are presented. In the conclusions it is stated that a statistical analysis of a large dataset would be required for evaluation but this has not yet been done. An appropriate title could be e.g. "A method to evaluate shortwave irradiances derived from satellite images of low-level arctic clouds with aircraft observations".

R: Thank you for your suggestion; the title was indeed not quite appropriate. We decided to change the title to "The effect of low-level thin Arctic clouds on shortwave irradiance: evaluation of estimates from space-borne passive imagery with aircraft observations". This title expresses that we did evaluate imagery-derived irradiances, albeit only with a limited number of cases (the nature of aircraft observations). A larger data set would be required to confirm/rebut the hypothesis that these types of clouds are ubiquitous in the Arctic.

- Abstract: "This study suggests that passive imagery cloud detection could be improved through a multi-pixel approach, that would make it more dependable in the Arctic." -> Cloud detection methods are not discussed at all in the paper, thus is sentence should not be in the abstract.

R: Thank you for the suggestion. We deleted the sentence in the revised manuscript.

- What is the uncertainty of the airborne observations? Please include error bars in the plots. If this is not possible, discuss the sources of errors in the text and provide a rough estimate.

R: Omitting the uncertainty analysis was an oversight on our part; we only realized when reading the reviewers' comments that it is central to the manuscript, and we made a considerable effort to add a thorough error analysis. We modified the figures by adding in the uncertainties as error bars in the revised manuscript. The uncertainty analysis for SSFR and BBR combined irradiance product (referred to as "SSFR-BBR") is provided in Appendix D (see **Line 12, Page 17**).

Technical corrections:

**p4, l11: "radiative calculations" -> "radiative transfer calculations"**

**R:** Thank you. In the revised manuscript, we decided to remove the discussion of the longwave effect of water vapor from the introduction, including this wording.

**p5, l5: "satellite-based radiative transfer calculations" -> "irradiances calculated using an RTE model with input from satellite data ..."**

**R:** Thank you. We changed the sentence to "Therefore, in this paper, we instead compare aircraft observations directly (pixel by pixel) with calculations based on satellite retrievals along the flight track" (**Line 1, Page 5**).

**p6 Eq.1: Reference for numbers used to convert RGB to grayscale**

**R:** We added the reference as a footnote for the weights in the RGB to Gray conversion equation (**Line 14, Page 7**).

**p11, l37: Fig. 6b -> Fig 7b**

**R:** Thank you for noticing the typo. We corrected the typo (**Line 14, Page, 13**).

[revised manuscript text omitted]

---

## Author Comment (AC4) · 20 Apr 2020

The page and line numbers used in the response are referring to the revised manuscript, which is appended to this response. Note that this is not the final revised manuscript, but is provided to show the reviewers where we made changes to address their concerns.

**Anonymous Referee #1**

**General comments**

**This paper describes an analysis of solar broadband and spectral irradiance data from an airborne measurement campaign in the Arctic in September 2014. Comparisons were made with radiative transfer model calculations (RTM) that take into account concurrent cloud products from MODIS satellite observations and retrievals of local spectral surface albedos from the aircraft measurements. A significant fraction of the work is concerned with the determination of the spectral ground albedos taking into account measured upwelling and downwelling irradiances, video images from the ground and literature data in spectral ranges where measurements were not feasible. The effort is justified because under the Arctic conditions, accurate ground albedos are crucial to distinguish cloud radiative effects from the influence of the relatively bright snow- or ice covered surface.**

**In general, the agreement of measurements and RTM calculations is satisfactory for both broadband and spectral measurement (at two selected wavelengths). However, significant differences were observed during some periods where no clouds were present according to the satellite data. This is taken as a hint towards the presence of undetected, optically thin clouds. However, in my view this interpretation is not as clear as currently presented in the paper and the influence of experimental uncertainties requires more attention. Nevertheless, the paper is well written, structured and documented. It should be published after minor revision.**

R: Thank you very much for your comments, especially for bringing up experimental uncertainties. We didn't appreciate until reading the reviewers' comments how central to the manuscript the uncertainty analysis is (see below).

To make the interpretation of the study more clear and to avoid confusion in the revised manuscript, we 1) modified the aim of this paper in the introduction (see **Line 14, Page 4**); 2) added a detailed description for each step of our approach at the beginning of the Data and Methods section (**see Line 5, Page 5**); 3) removed the discussion of cloud radiative effects that can be derived from the upwelling and downwelling irradiance.

We agree that the uncertainty analysis, which was missing in the original manuscript, was crucial because it can not only affect the conclusions drawn in the paper but can also help to make the conclusions stronger. We made an effort to add the uncertainty analysis for 1) the snow fraction estimated from nadir camera imagery; 2) the SSFR-BBR combined irradiance product; 3) the radiative transfer calculations based on MODIS cloud products to the revised manuscript (see **Appendix D, Line 12, Page 17**). We revised the figures by adding in the uncertainties as error bars as well as the text in the results discussion. Although the figures and the manuscript changed after adding in the uncertainty analysis, the conclusions of the paper did not change.

Specific comments

**Page 2, line 10: Clarify "radiative transfer model" instead of "model"**

R: Thank you. We corrected to "… inter-comparisons of irradiance measurements and radiative transfer calculations" (see **Line 10, Page 2**).

**Page 2, line 15: ". . .about 22% of clouds remained undetected (cloud optical thickness less than 0.5)." This statement is not in agreement with the main text, see page 11, line 36.**

R: Thank you for noticing the typo. We corrected the percentage in the result discussion (**Line 13, Page 13**) to make the statement consistent with the abstract. Also, the reviewer suggested to count the undetected clouds differently. As a result, the percentage number changed, but the numbers are kept consistent throughout the revised manuscript.

**Page 3, line 14: " . . .every $10^6 \, km^2$ decrease. . . a 2.5 W m$^{-2}$ increase". The statement is unclear. For which area does this apply?**

R: To make it more clear, we added the area information "averaged over the region from 75 º N to 90 º N" (**Line 15, Page 3**) to make the statement more clear.

**Page 4, line 36: Give more information on the flight area, altitudes and times e.g. in the text, in a table or a modified Fig. 1.**

R: We modified the Figure 1 by adding the longitude, latitude and altitude information. In addition, we specified the region in the figure caption.

**Page 5, line 16: The broadband instruments were probably not actively aligned. That should be made clear. Were data excluded from the analysis when the aircraft attitude was not horizontal?**

R: The broadband instrument (BBR) was not actively aligned. No, the data is not excluded when the aircraft attitude was not horizontal. A software attitude correction (Long et al., 2010) was applied to the BBR data to account for the change of solar position due to aircraft pitching and rolling. We added clarifications in the revised manuscript (**see Line 16, Page 5**).

**Page 5, line 25: ". . . that keeps the zenith light collector horizontal. . ." I assume that both collectors were actively aligned. If so, that should be clearly stated because for the determination of net irradiances this is certainly important.**

R: No, only the zenith light collector was actively aligned. The nadir light collector was fix mounted at the bottom of the aircraft. We added clarifications in the revised manuscript (see **Line 36, Page 5**).

**Page 5, line 28: ". . . that cannot be corrected." Another reason why low sun elevations are more challenging for the downwelling direct irradiance is that the gradient d cos(SZA) / (d SZA) increases with SZA.**

R: This is an excellent point. We think we have addressed this point in the original manuscript by mentioning "This is particularly important in the Arctic, where low sun elevations lead to large systematic errors for fix-mounted or poorly stabilized sensors (Wendisch et al., 2001)" (see **Line 37, Page 5**).

**Page 6, line 9: ". . . atmospheric effects". You probably mean "collector effects"?**

**R:** The "atmospheric effects" was misleading, but we didn't mean "collector effects". We meant the effect of cirrus. We modified to text to "… atmospheric effects (e.g., cirrus)" (**Line 21, Page 6**) to clarify.

**Page 6, line 25: I am convinced that the complicated collector-specific corrections and calibrations were done thoroughly. But can you estimate the remaining uncertainties for the upwelling and downwelling measurements, both broadband and spectral? For example in Appendix B, Fig. 13 the scatter in the ratios indicates uncertainties of the independently calibrated instruments that will not vanish by applying the azimuth correction. I assume Fig. 13 shows downwelling irradiances? If so it would be interesting to see a similar plot for the upwelling (with similar scatter but no azimuth dependence).**

**R:** Thank you for pointing out the lacking uncertainty estimation – and see above our response. Specifically for this comment: Yes, we made a considerable effort to add a thorough error analysis for the upwelling and downwelling, broadband and spectral irradiance measurements and calculations in the revised manuscript (see **Line 12, Page 17**).

BBR was more accurate (3% accuracy), but SSFR had spectral resolution, as described above. SSFR's azimuthal response was not isotropic (unlike BBR's), and it was corrected with BBR's response during a dedicated calibration flight. After azimuthal correction, the SSFR downwelling irradiance was scaled to BBR using the method described in Appendix D. It is in this sense that SSFR and BBR data were "merged". However, they are not identical. Since the calibration flight and the science flight occurred at different sun-sensor geometries, the broadband irradiance derived from SSFR and measured by BBR may deviate slightly from each other. We show them both to check for consistency between both methods. We need to show SSFR (in addition to just BBR) because we later use spectral irradiances from SSFR only to draw conclusions.

The details can be found in the revised manuscript Appendix D (see **Line 12, Page 17**). Yes, Fig. 13 shows downwelling irradiance. Figure A2 shows a plot of wavelength-integrated SSFR irradiances vs BBR. The linear relationship of upwelling irradiance indicates no azimuth response in upwelling irradiance.

**Page 6, line 36, Eq. 1: The different weighing implies a conversion to "brightness" or "luma" rather than a (relative) physical radiance. The source of the three coefficients should be cited and what they represent.**

**R:** The weighting conversion we used was to convert RGB colors from the video to "brightness". The nadir camera was not radiometrically calibrated, thus it could not provide any physical radiance for this study. Thank you for your suggestion, we added the reference as a footnote for the conversion weights (see **Line 14, Page 7**).

**Page 7, line 15: ". . .robust estimates". I wonder what robust means. The procedure is quite complicated and there are obviously several sources of uncertainties: (1) the coefficients in Eq. 1, (2) the blending technique Eq. 2, (3) the adaptive thresholding (parameters given in Appendix C) and (4) the (presumably) limited field of view of the camera compared to the irradiance collector. For example, does it make a significant difference if 0.333 is inserted in Eq. (1) for R, G, and B? It**

**would be convincing if you could provide an uncertainty estimate for the snow fraction based on a sensitivity study taking into account the different aspects (1)-(4).**

**R:** Thank you for the detailed thoughts on this matter. We used "robust estimates" because the method provides a reasonable snow fraction for the nadir camera images (even for scenes of pure snow) inspected by the human eye. The word "robust" can be subjective so we decided to delete the sentence to avoid confusion. We added an uncertainty analysis for the snow fraction estimated from the nadir camera images. The analysis shows that (1) and (2) were minimal factors compared to (3) and (4). We applied different FOV sizes and different adaptive thresholding areas and used the standard deviation of snow fraction retrieved using different parameter combinations of (3) and (4) to represent the estimate of snow fraction uncertainties. We included text regarding these sources of uncertainty to the revised manuscript (see **Line 34, Page 17**).

**Page 8, lines 12 and 13: Exchange hygrometer and thermometer in 3) and 4) Page 8, equation (4): Why not define "$\alpha_{SSFR} =$" here?**

**R:** Thank you for pointing this out. We exchanged hygrometer and thermometer at **Line 32, Page8** and defined $\alpha_{SSFR}$ in Equation (4) at **Line 24, Page 9**.

**Page 9, line 6: I assume the atmospheric correction was comparatively small because the altitude was below 300 m? You should give the reader a rough idea.**

**R:** Thank you for your suggestion. Yes, the atmospheric correction was comparatively small. The atmospheric correction corrected less than 0.2% on flight level albedo at the non-absorbing wavelengths. We added the magnitude information in the revised manuscript (see **Line 32, Page 18**).

**Page 9, line 33: To support this statement you could include the (much lower) albedo of the open ocean in Fig. 5. Moreover, in the introduction you mention a climatological surface albedo for the region by Moody et al., 2007 that is used for the MODIS cloud retrievals. It would be interesting to see how this compares with the locally measured data.**

**R:** Thank you for your suggestions. The spectral shape of the albedo that we obtained from our parameterization indicates that the surface albedo of the dark endmember (snow fraction of 0) consists of dark ice instead of open ocean, and the nadir camera imagery supports that. As supplementary material, we provide a video for the flight track that we used the aircraft data to do the spectral albedo parameterization (see supplementary material S1 "s1_flight-video_20140913-clear-sky.mp4"). Thus, we decided to not include the open ocean surface albedo.

We agree that the albedo used for the MODIS 1621 cloud retrievals is helpful, so we included it in Fig. 6.

**Page 10, line 9: Why 70%? In Fig. 4 a majority of data points is well above 80%.**

**R:** The 70% was determined by matching the radiative transfer calculations with the upwelling irradiance measured by SSFR-BBR at 1640 nm under clear-sky ($\sim 0.0107$ Wm$^{-2}$nm$^{-1}$). After adding the uncertainty analysis for snow fraction estimated from nadir camera image, the surface albedo parameterization changed. As a result, when we tried to match the clear-sky calculations with the clear-sky measurements, the snow fraction we got from the new surface albedo parameterization was 76.4%.

The data used to come up with the parameterization of the surface albedo in Fig. 4 is only a small portion of the data collected on 2014-09-13 (below-cloud under clear-sky) because only a small portion from this case was collected under clear-sky conditions. The majority of the clear-sky data points in Fig. 4 does indeed have a snow fraction well above 80%, but the conditions for the entire region conducted on 2014-09-11 (a different day/location) are different, and we did not know what was the exact snow fraction was. We arrived at a snow fraction of 76.4% by tweaking the clear-sky calculations until they matched the clear-sky measurements at 1640 nm (we added an explanation how we arrived at this number in the revised manuscript, **Line 38, Page 10**).

**Page 10, line 21: "cloud optical thickness"**

**R:** Thank you for noticing the typo. We corrected to "cloud optical thickness" (see **Line 20, Page 11**).

**Page 10, line 21: ". . .agree with measurements within 10%" Does this statement refer to data in Fig. 7b or to other flight periods? In Fig. 7b agreement within 10% under no-cloud conditions is only visible at the very beginning. My impression is that good agreement during periods where no clouds were detected could be reached by increasing the snow fraction to about 80% without significantly affecting the upwelling in the presence of clouds. So the 70% snow fraction assumption and its uncertainty are crucial here and should be discussed.**

**R:** Yes, the 10% statement was referring to Fig. 7b. As we discussed in the previous response at "Page 10, line 9: ….", the 70% was determined by matching the radiative transfer calculations with the upwelling irradiance measured by SSFR-BBR at 1640 nm under clear-sky (~ 0.0107 $Wm^{-2}nm^{-1}$). We agreed that by adjusting the snow fraction to a higher value would increase the calculation-measurement consistencies in general, however, the nadir camera imageries indicate very thin clouds (wavy patterns) existed. Later, we used the multi-pixel approach (see Fig. 12) and found the surface albedo changed spectrally that the surface albedo parameterization we obtained from 2014-09-13 clear-sky data cannot perfectly represent the surface albedo of 2014-09-11 by adjusting the snow fraction. We added the uncertainty analysis for the snow fraction and associated uncertainty in radiative transfer calculations in the revised manuscript (see **Line12, Page 17**).

**Page 10, line 23: ". . . below 0.5" In Fig. 7b it seems that the missing about 30 W $m^{-2}$ upwelling RTM irradiance (above the clear baseline) correspond to a missing COT of about 2 which is well above the threshold of 0.5. Moreover, if the COT were indeed around 2 no ice-structures on the ground would be visible in the photographs (i)-(iv).**

**R:** Due to the dense collection of the data points, the COT threshold was not clear. We did evaluate the histograms of the irradiance and the 30 $Wm^{-2}$ low bias in the upwelling calculations was associated with the missing COT of about 0.5.

**Page 10, line 23: The ". . . continuous variation from leg to leg. . ." Compared to the COT data the measured peaks in the upwelling irradiance look smoothed out. I wonder if this has to do with the fields of view of the instruments which at 7 km altitude are much greater than that of the satellite (1 km). So even if you fly above a correctly detected cloud gap, the sensors could receive irradiance from surrounding, even distant cloud fields. So, overall wonder if the results in Fig. 7 can be explained without "undetected clouds", i.e. by field of view effects and an underestimated albedo.**

**R:** These are excellent thoughts. Indeed, the FOV size is greater than that of satellite and the clouds from surrounding or distant area can contribute signals in upwelling irradiance, and we now added a discussion regarding this point (see **Line 23, Page 11**). However, our conclusions were drawn based on the

following two points (1) With a 45º Field Of View (FOV), the FOV size is much smaller than the cloud gap; (2) from the time-synched images from the nadir camera, thin clouds were clearly seen beneath the aircraft where MODIS identified as clear-sky. The primary factor that we base our conclusion on is (1). To support our point that the FOV is smaller than the cloud gaps, we added horizontal error bars in Fig. 7b. Overall, the size of these horizontal error bars adds uncertainty to the percentage number of clouds that go undetected by MODIS, but the general statement (that MODIS under-detects low-level thin clouds) is still supported by our data. The nadir imagery (2) further supports makes our statement more plausible because we can actually see clouds. The camera imagery thus does not serve as quantitative estimation of the amount of clouds that are missed, but simply corroborate our findings in a qualitative way.

**Page 11, line 3: ". . . via the snow fraction" What was the mean snow fraction during that flight?**

**R:** The mean snow fraction during that flight (2014-09-13 below cloud case) was 91.6%.

**Page 11, line 7: ". . .except. . .22:22:48" I cannot find that period in the figure.**

**R:** The 22:22:48 was in the format of HH:MM:SS for Fig. 8 (a) and (b).

**Page 11, line 11: ". . . no evidence of any cloud gap". This also hints towards different fields of view of satellite and aircraft. Even if you fly underneath a cloud gap, at estimated solar zenith angles of around 70 deg no direct radiation (and a corresponding increase of irradiance) may be present. It may be detected at other places (also underneath a cloud) but the chances for that are low if the cloud gap is small and cloud thickness is substantial. It would be helpful to estimate the size of the cloud gaps detected by the satellite. Moreover, a statement should be included whether or not the cloud top and bottom heights were correctly retrieved from the satellite data on both days.**

**R:** The reviewer is right; from the below-cloud observations one cannot actually make a strong statement. That is because the satellite is viewing the scene from above (potentially overhead), whereas the change in irradiance as seen from the aircraft is dominated by the direct beam, which can easily be blocked by clouds even if there are gaps (because of the low sun elevation and cloud geometrical thickness). We overlooked this because our impression and understanding of the clouds came from the nadir and forward camera videos that were not included in the manuscript. We attached the flight video we created (synchronized the forward, nadir camera) for this particular flight track (see supplementary material S1 "s1_flight-video_20140913-clear-sky.mp4"). The video confirms that clouds were geometrically thin, and

that there were indeed gaps, or very thin clouds with the blue sky shining through as shown on the left figure. From the MODIS perspective, the scale of these cloud gaps or very thin cloud segments is most likely below the pixel dimension (1km). The MODIS cloud detection algorithm seems to interpret these features as gaps, as shown on the map to the right (figure titled "Cloud Optical Thickness"). These occur right on the flight track we analyzed (thick black line), for example, near -133.5°, 73.1°. The size of the gaps in the MODIS L2 retrievals range from 1km to 3km as shown to the

right, but in reality, the gaps (or very thin clouds) as shown on the picture occur on much smaller spatial scales.

[Figure]

After thinking about the two figures included here, we no longer think that the different sun-sensor geometry explains the fact that cloud gaps show up in the MODIS-based calculations, but not in the measurements. Instead, we now believe that small cloud gaps or thin clouds do occur in the measurement area, but they have different effect on the aircraft observations relative to the satellite observations. **From the below-cloud perspective, the gaps (or very thin clouds) are not large or frequent enough permit the direct beam to be transmitted, leading to an irradiance time series that looks fairly smooth. From the satellite perspective, several sub-grid resolution gaps seem to prevent the detection of a cloud at that pixel.** These undetected pixels in the middle of an otherwise fairly homogeneous cloud field (COT 5-6) can be regarded as an artifact. In that sense, MODIS does "miss" these pixels. We surmise that this happens not because the optical thickness is below some threshold (as in the other case), but because of sub-grid variability of the cloud field. The figure on the left (figure titled "MODIS L1B QKM Radiance (860nm)") shows the radiance field at the finest spatial MODIS resolution (0.25 km). It shows a multi-pixel cloud gap (blue color near the northern end of the thick black line). It does not show any real cloud gaps south of that, unlike the L2 retrieval above, which does show some gaps. We added a brief explanation in the manuscript (see **Line 19, Page 12**).

**Page 11, line 15: Table 1 and 2 contain a main result of this study that is not adequately discussed here and should clearly enter the abstract and conclusions: Under conditions when clouds are detected the numbers in the first three columns agree, at least (I assume) within experimental uncertainties which need to be specified.**

**R:** We acknowledge that we did not adequately discuss the cloud radiative effect shown in Table 1 and 2. In the revised manuscript, we decided to focus on the irradiance comparison. Thus we removed Table 1 and 2. In addition, we added uncertainties. They are provided for both the measurements and calculations and indicated as error bars in the updated figures of the revised manuscript.

**Page 11, line 23: Do September 11 and 13 belong to the Arctic dry or wet season?**

**R:** Based on the definition in Kay and L'Ecuyer's (2013) paper, September 11 and 13 are belong to dry season. In the revised manuscript, we added the comparison for the dry- and wet-season albedo with the surface albedo derived for September 11 in Figure 6.

**Page 11, line 36: "Of all pixels along . . . 22%" To determine the fraction of undetected clouds as stated in the abstract, green/(cloudy +green) should be calculated.**

**R:** Thank you – this is a great idea and more suitable for representing "percentage of undetected clouds". We recalculated the percentage using green/(cloudy+green) and we got 27%. We revised the 22% to 27% using your proposed method throughout the revised manuscript.

**Page 11, line 37: ". . . (highlighted in green) are actually cloudy." Again the question arises if during the green periods clouds were undetected by the satellite or radiation was captured unintentionally by the irradiance sensors at some distance from the aircraft.**

**R:** Thank you for your insightful thoughts. You are correct – the large field of view (FOV) of the aircraft radiometers (when flying high) will capture irradiance by clouds at some distance from the aircraft. To evaluate this, we assumed a FOV of 45° for the radiometer and calculated the FOV size based on the aircraft altitude. The FOV is defined as the cone from within which 50-70% of the radiation originates (of course, radiation from the entire hemisphere is detected, but it is weighed much less for incidence angles much larger than 45°). Roughly speaking, the diameter of the SSFR "pixel" size corresponding to the 45° (>half-power) FOV equals the altitude of the aircraft above the cloud deck. Accounting for the motion of the aircraft, we translated the FOV size (units: meter) to time (units: second) and showed the diameter of the FOV along the flight track as horizontal error bars in Fig. 7b. We added an explanation in the revised manuscript (see **Line 23, Page 11**). The horizontal error bars show that the region of undetected clouds is much larger than the FOV effect. The nadir camera imagery also supports this because the clouds can be clearly seen directly below the aircraft. Thus, our conclusions of ". . . (highlighted in green in Figure 9b) are actually cloudy." is correct, even after considering the FOV effect.

**Page 12, line 19: ". . . surface albedo ... is biased low by about 9%". The 9% come from the albedo + atmospheric correction which is probably significant at 7 km altitude.**

**R:** The atmospheric correction would not affect the percentage (e.g., 9%) because we were comparing the upwelling irradiance from the SSFR-BBR and radiative transfer calculations at flight altitude. The atmospheric effects were considered in the radiative transfer model (RTM). However, the cirrus will contribute to the percentage because we didn't consider cirrus in the RTM.

**Page 12, line 24: "Simply changing the snow fraction does not improve the agreement. . ." I am not so sure. Increasing the snow fraction will lift the RTM irradiances for shorter wavelengths more strongly than for the long wavelengths (Fig. 4).**

**R:** In principle, this is correct. However, from what we saw in Fig. 12, we currently have a low bias in the surface albedo at short wavelengths, and a high bias at long wavelengths (see Fig. 12). Therefore, simply changing (increasing/decreasing) the snow fraction will not improve the agreement for both long and short wavelengths. We added more explanation in the revised manuscript to avoid confusion (see **Line 3, Page 14**).

**Page 13, line 31: "Undetected thin clouds (COT<0.5) led to a high bias. . .." I don't think this statement is justified. Looking at Fig. 8 the measured downward irradiances vary around 200 W $m^{-2}$ and COT vary around 6 (where detected) with consistent RTM results. I assume a COT of 0.5 would produce downward irradiances well below the clear ones but also significantly greater than measured. So the measurements are inconsistent with the presence of thin clouds which makes an explanation as given above (page 11, line 11) more likely.**

**R:** Thank you again for these observations/thoughts. Referring to the response given above (the one with the forward camera imagery + MODIS data), the forward camera indicates that there were indeed some translucent areas where one can see the blue sky through the thin clouds. Our conclusions are given above: "From the below-cloud perspective, the gaps (or very thin clouds) are not large or frequent enough permit the direct beam to be transmitted, leading to an irradiance time series that looks fairly smooth. From the satellite perspective, several sub-grid resolution gaps seem to prevent the detection of a cloud at that pixel." (see more detailed response and figures there).

**Page 13, line 32: ". . . above clouds . . ." should be ". . .below clouds. . ." See Fig. 8 and page 11, line 13.**

R: Thank you for noticing the typo. We corrected ". . . above clouds . . ." to ". . .below clouds. . ." (see **Line 14, Page 15**).

**Table 1 and Table 2: Please indicate in the captions place, time and altitude. Specify above or below cloud conditions and the unit of the numbers. Explain what is listed in the last column called "RTM" The precision of the numbers implies an accuracy that is unrealistic. I assume that the numbers in the first three columns agree within experimental uncertainties but that needs to be specified. Please state the average COT, cloud top and bottom heights.**

R: As we mentioned that the Table 1 and Table 2 are removed in the revised manuscript and focus the paper on irradiance comparisons. The last column of "RTM" indicated the estimated irradiance/CRE of the radiative transfer calculations with clouds detected and with clouds undetected in the original manuscript. We agreed the numbers implied an unrealistic accuracy. Since we obtained the uncertainty estimates for the measurements and calculations, we updated the numbers with uncertainties in the manuscript. The COT information of 11 September was provided in the original manuscript (**Line 21, Page 11**). The COT information of 13 September was added in the revised manuscript (**Line 3, Page 12**). The cloud top and bottom heights were added in the revised manuscript (**Line 9, Page 9**; also indicated in Fig. 3).

**Fig. 1: Specify the maps' latitude and longitude ranges, e.g. in the caption. A costal line can be vaguely recognized showing that the flight area was west of Banks Island but I assume hardly any reader is familiar with the area.**

R: Thank you for the suggestion. We added the longitude and latitude labels in Fig. 1, and also added the information of the location in the caption (see **Page 24**).

**Fig. 2: 80.70% implies a precision that is certainly not justified by the method (see comment in text). Indicate the field of view of the circular areas in (a), the flight altitude, time and location. The y-axis label should read "spectral flux density" or better "spectral irradiance" in accordance with the main text.**

R: Thank you for pointing out the precision and uncertainty issue. To justify the snow fraction number, we added an uncertainty analysis for the snow fraction. The uncertainty of the snow fraction for this particular image frame was 4%. Thus, we used "81%±4%" for the snow fraction estimated from the camera imagery. The diameter of the field of view was about 380m based on the aircraft ground speed and altitude of 134m. The FOV size information and the flight altitude, time, and location were added in the figure caption (see **Page 25**).

Thank you for your suggestion. We changed the "spectral flux" to "spectral irradiance" in the y-axis label.

**Fig. 3: Approximate times and locations should be specified in the caption. The indicated flight levels are confusing here without additional information.**

R: Thank you for the suggestion. On 11 September, MERRA-2 data at 21:00 UTC was averaged over the region of [135W, 130.625W, 72.5N, 74N] to represent the atmospheric profiles. On 13 September, aircraft data from a descending leg (19:31 UTC to 19:50 UTC at 133.8W, 74.1N) was used for the atmospheric profiles. The flight level range indicated in the figures are the flight levels of the aircraft data

used for irradiance comparison for above-cloud case (refer to Fig. 7) and below-cloud case (refer to Fig. 8). We added the information to clarify in the revised manuscript (see **Page 26**).

**Fig. 4: Given the spectral resolutions, the indicated wavelengths are too precise. 640, 1240 and 1630 nm, as stated in the text, are appropriate.**

**R:** Thank you for your suggestion. We revised the wavelengths in the figure legend to 640, 1240, and 1630 nm (see **Page 27**).

**Fig. 5: Consider including open ocean albedo and data by Moody et al., 2007 (see comment in text).**

**R:** Thank you for your suggestion. As we discussed in the previous response, the surface albedo we derived for the dark endmember (snow fraction = 0) was mostly for the surface consisting of dark ice. Therefore, we decided not to include open ocean albedo, but we did include the climatological surface albedo used by the MODIS 1621 cloud retrieval over snow.

**Fig. 7 and Fig. 8: Indicate the flight altitudes in captions and maybe the cruise speeds so that the size of the cloud gaps can be inferred.**

**R:** Thank you for the suggestion. We added the flight altitude and cruise speed information in the figure captions (see **Page 30 and 31**).

**Fig. 13: Please indicate that these are ratios of downwelling irradiances (I assume).**

**R:** Yes, they are ratios of downwelling irradiances. We added clarifications in the figure caption (see **Page 36**). We also changed the name of the figure to Figure A1.

**Appendix B, Fig. 13: Were the solar zenith angles during this flight comparable to those during the other flights? The azimuth dependence may change with solar zenith angle for geometrical reasons and because of a varying (wavelength dependent) contribution of direct irradiance.**

**R:** The solar zenith angles during the calibration flight (2014-10-02) and the flight where the azimuth correction was applied (2014-09-11) were comparable. The data collected during calibration flight (2014-10-02) has solar zenith range of [68.24 º, 71.49 º] with an average of 70.20 º. The solar zenith angle range for the above-cloud case (2014-09-11) is [68.46º to 71.89º] with the mean of 68.91º. We added this information to the revised manuscript to clarify (see **Line 20, Page16**)

**Appendix C: In order to understand the meaning of the factor d=1501 the images' total pixel dimension should be stated.**

**R:** The pixel dimension is 2592 (width) ×1944 (height). The information was added to the text (**Line 5, Page 7**).

Technical corrections

**Page 4, line 13: Introduce "CRE" as "cloud radiative effects"**

**R:** As we mentioned in the general response, we decided to remove the discussion of cloud radiative effects. Thus we changed to "… the challenges for deriving shortwave irradiance …"(see **Line 11, Page 4**).

**Page 9, line 23: "2014-09-13"**

**R:** Thank you for noticing the typo. We corrected to "2014-09-13" (see **Line 15, Page 10**).

**Page 11, line 31: Fig. 7?**

**R:** Thank you for noticing the typo. We corrected to Figure 7 (see **Line 5, Page 13**).

**Page 11, line 37: Fig. 7?**

**R:** Thank you for noticing the typo. We corrected to Figure 7 (see **Line 14, Page 13**).

**Fig. 5: Wet and dry season colours are hard to distinguish. Second citation should be Brandt et al.**

**R:** We moved the wet and dry season albedo from Fig. 5 to Fig. 6 to avoid confusion and corrected the citation to "Brandt et al." in the legend of Figure 5.

**Fig. 7 and Fig. 8: y-axis: broadband irradiance. Typo "MOODIS" in Fig. 8 Fig. 9 and Fig. 10: y-axis: spectral irradiance**

**R:** Thank you for your suggestion and noticing the typo. We changed the y-axis labels from "flux" to "irradiance" for Figure 7, Figure 8, Figure 9, and Figure 10. We corrected the "MOODIS" to "MODIS" .

[revised manuscript text omitted]

---

## Author Comment (AC5) · 20 Apr 2020

The page and line numbers used in the response are referring to the revised manuscript, which is appended to this response. Note that this is not the final revised manuscript, but is provided to show the reviewers where we made changes to address their concerns.

**Anonymous Referee #2**

**1 General remarks**

**The manuscript analyses airborne radiation observations and satellite observations of Arctic clouds. A surface albedo parametrization is derived to account for the inhomogeneous Arctic sea ice surface. Spectral and broadband are compared to radiative transfer simulations which are based on satellite observations. The spectral irradiance is analyzed to untangle uncertainties resulting from the surface albedo and the cloud optical properties.**

**In general, the analysis of airborne observations in remote Arctic areas is of high value and provides one rare tool to validate satellite observations. Therefore, the study has high potential and is within the scope of AMT. It could have a wide scientific interest and might contribute to improve our understanding of Arctic clouds. However, the manuscript lacks in several major issues and therefore, does not exhaust its full potential. These issues have to be reassessed in detail before publishing the manuscript.**

**First, the objective of the study is not well presented and outlined. Based on the title and introduction, the readers expectations and the presented analysis may strongly differ. This deficiency might results from a non-existing description of a general approach and methodology how the measurements can and will be used to validate and improve satellite observations. Such a general strategy is an important part of the manuscript in order to promote future application of the methods. Based on the unclear objectives also the conclusions are weak and leave many questions unanswered putting off the reader with promises for future studies. Finally, a throughout uncertainty estimation is missing, which is mandatory if observations are used for validation purposes. I'm sure, there are options to restructure and improve the manuscript in a way that it presents the full potential of the study.**

**Below, I compiled a list of comments which have to be considered in a revised version of the paper. There might be some contradictory statements which result from my misinterpretation of the text when first reading the manuscript. I am sure the authors will know how to weight in such cases and how to improve the text to avoid misinterpretations by other readers.**

**R:** Thank you very much for your perspective. As we now realize, the manuscript had some serious shortcomings (especially regarding the uncertainty analysis), which we believe we addressed with the revised version. The annotated revised manuscript is attached to this response; page/line numbers refer to that version unless stated otherwise. We do think that we stated the objectives in the original manuscript (p4l16, original manuscript). However, the title may have been confusing, and we changed it (see below). To summarize: The objectives of this study are 1) to quantify the discrepancies between observed and satellite imagery (MODIS) derived irradiance, and 2) to identify the key error sources of the discrepancies using broadband and spectral measurements. In the original manuscript, the general approach was introduced after the objectives (see **Line 5, Page 5**). As the reviewer states, the title may not have properly reflected the objectives, and we therefore changed it to "The effect of low-level thin Arctic

clouds on shortwave irradiance: Evaluation of estimates from space-borne passive imagery with aircraft observations".

It is always difficult to define the scope of a manuscript. In our case, we used the (limited) data from this study to assess to what extent satellite-derived estimates of irradiances under cloudy conditions (i.e., satellite-derived cloud radiative effects) are rooted in reality. To do so, we had to dive into the surface albedo as one of the parameters controlling the changes of irradiances by a cloud, as well as into the other drivers, finding in the end that the most important factor was (in our case) that MODIS simply didn't detect a significant fraction of these clouds. Indeed, we could have done many other things as well (for example, actually calculated net irradiances in the SW and LW to come up with the radiative effect of these clouds), but the emphasis was on the irradiances themselves, not on the derived quantity, the radiative effect. That said, we are not entirely sure what the reviewer meant by asking for the "full potential" of this paper. We would rather not re-write the paper to do something that was not intended in the beginning, nor do we feel that we should go beyond the scope of the paper because it is long as it is. Instead, we decided to clarify the objectives more, as well as emphasize the importance of the conclusions, which we do believe are strong.

We hope that changing the title, as well as clarifying the objectives (see details below) defines the scope better. We disagree that the conclusions are unclear; on the contrary, we actually think that the conclusions are rather succinct. At **Line 29, Page 13** in the original manuscript, the conclusions were clearly stated. Indeed, our study does raise many questions, particularly in the conclusions section. In our understanding, raising new questions based on observations is a justified and commonly accepted approach. We did answer the fundamental questions of this paper, which are (1) How well do satellite-derived shortwave irradiances compare with observations [caveat: we do so with whatever limited measurements we were able to use]; (2) what are the leading causes of any discovered discrepancies? As stated above, we found that the largest error source was the fact that MODIS does not detect the thinnest clouds. This finding begs the question whether this is true in regions of the Arctic other than our limited sample. We do think it is adequate to raise this question in the conclusion.

We did take to heart the reviewer's comment that the text might be easy to mis-interpret by the reader, and we carefully went through the individual comments below. We hope that changes in the manuscript make it more readable, and would like to again thank the reviewer for pointing out these issues.

Thank you also especially for the comments regarding the uncertainty. This was also raised by the other two reviewers, and constituted a significant omission. We spent considerable time and effort for an in-depth uncertainty analysis, especially in the estimates of the snow fraction from the aircraft imagery, on the SSFR-BBR irradiance product, and on the radiative transfer calculations based on MODIS retrievals. The details can be found in Appendix D (**Line 12, Page 17**), and the error bars can be found in many of the figures in the main body of the paper. After doing all of these, the message still stands, but we feel that it has become stronger.

**2 Major comments**

**2.1 Unclear objective of the study**

After reading the title and the introduction it is somehow unclear, what the manuscript aims to obtain: irradiance or the cloud radiative effects. The introduction does not match the title. The analysis and methods presented in the study also show not what was promised in the title: "Shortwave radiative effect" was calculated and discussed only briefly. Most of the analysis concentrates on irradiances. The introduction does not give an overview on how cloud shortwave radiative effects are commonly derived. In the analysis CRE is only discussed in two sentences. Neither the method and uncertainties are introduced nor are the values discussed. This does not justifies the title of the manuscript.

R: The manuscript aims to evaluate the irradiance derived from passive satellite imagery under cloudy conditions in the Arctic. A better understanding and error quantification of irradiances derived from satellite imagery is needed to better understand cloud radiative effects as well. The original title of the manuscript was chosen in that spirit. However, although cloud radiative effects are closely associated with the irradiance, our discussion on the CRE is indeed limited. We therefore agree that the title with CRE was misleading and we changed to "The effect of low-level thin Arctic clouds on shortwave irradiance: Evaluation of estimates from space-borne passive imagery with aircraft observations". After carefully thinking about changing the title as well as the fact that the CRE is not the main focus of this study, we decided to remove the two sentences that discussed the estimated CRE from the two cases, and focused the content solely on the irradiance evaluation.

"Imagery-derived irradiance": To me, this implies, that camera images are used and integrated into an irradiance. Or at least, that the irradiance is directly derived from images. That's not done in the manuscript and also not at all covered by the introduction. What the authors did is a parametrization of the surface albedo based on the sea ice fraction, which was observed from a camera. So I suggest to remove the word "imagery-derived" from the title.

R: What we meant by "imagery-derived irradiance" in the original title was irradiance derived from passive *satellite* imagery. In fact, the use of the satellite imagery (and derived products) is also explained in the paper: It is done by using the optical thickness and effective radius, derived from the radiances observed by MODIS, in radiative transfer calculations to calculate irradiance. These radiative transfer calculations also require the surface albedo, which is derived from a combination of the airborne radiometers (SSFR) and the *aircraft* imagery (camera). However, the aircraft nadir camera images were not used for any irradiance calculations. We only used the nadir camera images for estimating the snow fraction in the surface albedo parameterization. To clarify, we now used the word "space-borne passive imagery" in the title and elsewhere.

This misleading title leaves the reader searching for the actual objectives of the study. Unfortunately, also the motivation given in begin of the manuscript does not fit to what finally was achieved. E.g.:
"Validation of CERES-MODIS derived irradiance": In section 2, the authors state that one objective is to validate CERES-MODIS derived irradiance. This is confusing after reading the title and introduction. CERES irradiance is a different story compared to estimating the CRE. And I also do not see a CERES product in the study. The authors theirself state, that the design of the measurement strategy failed to compare to the CERES product. MODIS retrieval and own radiative transfer simulations are applied. However, on Page 13 line 36 the authors conclude, that CERES observations are used to constrain the observations. This was not done. To avoid confusion,

**I suggest to remove CERES form the argumentation. Still, comparing irradiances is not the same as estimating the CRE.**

**R:** "Validation of CERES-MODIS derived irradiance" was actually not the objectives of this study, nor was this stated in the manuscript. "Validation of CERES-MODIS derived irradiance" was mentioned when we introduced the ARISE field campaign. When talking about ARISE in general, we had to talk about its primary objective, the statistical validation of CERES/MODIS products, because this objective explains the "lawn mower" patterns, which were developed to cover multiple CERES footprints. However, this statistical comparison is a separate paper by other authors, which, however, has not been published yet. An AGU abstract can be found here:
https://ui.adsabs.harvard.edu/abs/2015AGUFM.C41D0753C/abstract
Whereas the CERES validation paper aggregates multiple CERES footprints in a 100 km x 100 km region and compares the aggregated products with the aircraft measurement PDFs in this same region, we here pursue a direct inter-comparison of MODIS-derived irradiances with aircraft observations on the pixel-level (pixel-by-pixel along the flight track). The distinction between the motivation of ARISE as a whole (the statement the reviewer picked up on), and our study is described in the objectives: 1) to quantify the discrepancies between observed and satellite imagery (MODIS) derived irradiance, and 2) to identify the key error sources of the discrepancies using broadband and spectral measurements. These were actually clearly stated at **Line 14, Page 4**.

Regarding this comment: "**The authors theirself state, that the design of the measurement strategy failed to compare to the CERES product.**", we are not sure where this is coming from; we did not say that the measurement strategy of ARISE failed to validate the CERES product. What we did say was that our paper follows a different approach, and that we did not intercompare CERES-derived irradiances with aircraft observations. Instead, we used MODIS-derived irradiances to compare with irradiance measurements at the pixel level. The basic idea is described in the BAMS paper of ARISE (Smith et al. 2017), and we anticipate that a paper about the direct comparison (see AGU abstract) will still be published.

We also did not state that the CERES observations are used to constrain the observations provided in *this* study. The original words from the manuscript were "While the calculations in the above-cloud case can be constrained through the TOA radiation product from satellite observations (CERES)". This was a *general* statement to emphasize that undetected clouds will be even more problematic for the surface energy budget than at TOA because CERES on Aqua and Terra constrains TOA fluxes more directly than those at the surface. To clarify, we rephrased to "While the radiation calculations at TOA can be constrained through the radiation product from satellite observations (e.g., CERES)". After adding clarifications in the text of the manuscript, we decided to keep the CERES related words in the revised manuscript to make the description of the ARISE campaign accurate and complete.

**"longwave radiative effects": There is one section in the introduction on the longwave effect of water vapor on the surface radiation budget. But the title of the manuscript suggest, that the study is on solar effects only. So longwave radiative effects by water vapor is kind of irrelevant. The manuscript also does not include a study on the radiative effects of the water vapor profiles. Only the pure profiles are discussed.**

**R:** It is true that the manuscript is focused on the shortwave radiative effects. We agree that the longwave effect of water vapor has no immediate relevance for this manuscript; we had initially included for context because insufficient knowledge of the water vapor may lead to uncertainties in the (surface) radiation budget rivaling those from insufficient knowledge about clouds. We decided to remove the discussion of the longwave effect of water vapor from the introduction of the revised manuscript.

We also acknowledge that the impact of insufficient knowledge about water vapor on the *shortwave* irradiance was not discussed thoroughly in the original manuscript. We used atmospheric profiles from MERRA-2 and aircraft data in the radiative transfer calculations, and they agreed well with the measurements in the water vapor absorption bands. In the revised manuscript, we checked the sensitivity with respect to water vapor by adding radiative transfer calculations with the climatological water vapor profile (in addition to the MERRA-2/aircraft-derived water vapor profile). The error contribution due to inaccurate water vapor knowledge is shown in Fig. 10 (see **Page 34**).

**After all, I had the feeling that the manuscript shows a potpourri of separate analysis, without a clear major goal. This probably is not true, but the manuscript requires a more clear objective. From what I read, the study aims for a closure study, which validates the MODIS cloud product by airborne observations of irradiance. Could this be the major aim? Or do you aim identifying radiative processes in the Arctic atmosphere related to surface-cloud interaction?**

**R:** As stated in the response to the general comments, we actually think our objectives were clear, but that they may not have been well enough formulated: 1) to quantify the discrepancies between observed and satellite imagery (MODIS) derived irradiance, and 2) to identify the key error sources of the discrepancies using broadband and spectral measurements. To do so, we first evaluate the key parameters that control the changes of irradiances by a cloud. Since the surface albedo, which is one of the before-mentioned key parameters, can be derived from the direct clear-sky measurements of SSFR, we developed a parameterization that utilized the surface context information from nadir camera imagery to account for the surface variability that the surface albedo parameterization can be used for cloudy calculations, which are the focus of this paper. After the surface parameterization, we discussed the model calculations with measurements from broadband and spectral perspectives and we discovered that the major error contribution is from MODIS not seeing the clouds. We added clarification about this on **Line 14, Page 4** in the revised manuscript.

**I think it is important to clarify the main objective of the manuscript and concentrate on the major aspects needed to achieve these goals. If the aim is a closure study, then I suggest to remove the estimation of cloud radiative effects, which currently is misleading. Or at least shift the calculation of CRE to the end of the study, after the irradiances have been compared. This would allow to extend the validation for CRE based on the uncertainties/conclusions which have been found already before when comparing the irradiances.**

**R:** Thanks for your suggestions on several different ways of clarifying and concentrating on the main objectives of the paper. We realized that the CRE aspect of this paper was confusing, and we decided to remove the estimated CRE from the manuscript as well as the title. We also modified the text at the beginning of the Data and Methods section (see **Line 14, Page 4**) to make the objectives more clear. We hope the message will become clear in the revised manuscript.

**2.2 Methodology needs to be outlined**

**Several comparisons of different quantities (albedo, irradiance above, below clouds) are shown in the manuscript. However, it is not always clear what the purpose of each individual step is. The general and also the specific methodology of the analysis should be outlined. In the conclusion the authors write about "developing a validation approach". I don't see a clear validation approach in**

**the study. If there is a strategy, then this needs to be outlined precisely in the begin of the manuscript. Maybe any schematic showing the different comparisons broadband, spectral, flight one, flight two might help.**

R: Thank you for your comments. The primary goal of this study is to evaluate the radiative transfer calculations of irradiance derived from cloud optical properties provided by passive remote sensing in the Arctic, identify the error sources in the radiative transfer calculations and quantify the error contributions. To do so, we had to try our best to make the surface albedo right using the aircraft measurements. That's why we first provided the surface albedo parameterization. After we obtained the most "suitable" surface albedo set, we evaluated the irradiance under cloudy conditions for above-cloud case and below-cloud case. We intended to use above-cloud case as a proxy to evaluate how well satellite-derived irradiances represent the reality, and below-cloud case as to assess the realism of satellite-derived surface budget parameters (specifically, the downwelling and upwelling irradiance). As one can see, the elements of the paper together are prerequisites to understanding the realism of satellite-derive CRE as well, although we are no longer talking about CRE in the paper (as suggested by the reviewer) to avoid confusion. To make it more clear, we added more detailed description about the steps at **Line 5, Page 5**. The "developing a validation approach" was more of an outlook. The study was not about developing a validation approach, it is a hypothesis study. Thanks for your suggestions on outlining the strategy.

**It is also confusing, how and when the data of both flights is used. It took long until I understood, that the two flights provide different observations (high vs. low flight altitude). I suggest, that the authors clearly report, what is different between both flights. Why two cases are needed and how the observations are mixed/combined in the study?**

R: In the original manuscript, we had the flight information at **Line 2, Page 10**. As stated, for 2014-09-11 (also referred to as above-cloud case), the aircraft measurements were taken when aircraft flew high around 6.5km. However, for 2014-09-13 (referred to as below-cloud case), the aircraft measurements were taken when aircraft flew low at around 200m. The ARISE campaign was designed that flights were either above or below a cloud field, but almost never both (because of sampling time limitations). For this reason, we needed to study two distinct flights to evaluate at above-cloud and below-cloud budget parameters (irradiances). We did not mix or combine the observations of the above-cloud case and below-cloud case together because they occurred on different days.

**Similarly, the motivation of section 3.1 was missing and leaves many open questions when reading the section. These questions should be addressed before starting the analysis:**
**What is the purpose of this analysis and of the parametrization of surface albedo with snow fraction? I only can guess. Wasn't surface albedo measured directly with BBR and SSFR? Where is the need to parameterize surface albedo if albedo is measured anyway?**

R: The BBR/SSFR can directly measure the surface albedo, but not under all conditions, specifically, when clouds were present. Only clear-sky conditions could be used to calculate the surface albedo. These are a rare occurrence because the low-level clouds were ubiquitous. The main driver for the variability of the surface albedo was the snow fraction, and therefore we decided to couple the (rare) direct albedo measurements to the camera-derived snow fraction, which is more readily available, even under partially cloudy conditions. In addition, we could then use the snow fraction as a single "tuning" parameter, which ensured that the calculated upwelling irradiance measurements were in agreement with the measurement above cloud. In short, that is why we used the data aggregation technique, which used a small collection of low-level clear-sky aircraft measurements to parameterize the surface albedo. After the

parameterization, one can retrieve surface albedo for 1) the above-cloud case study, where the clouds cover the surface and the surface albedo cannot be directly measured, 2) the below-cloud case study, where the clouds block the sun and the surface albedo cannot be directly measured. Moreover, the albedo parameterization was our simple way to account for the surface variability in the Arctic using a collection of data. We clarified this in the revised manuscript (see **Line 35, Page 10**).

**As the surface albedo properly is an input to the radiative transfer model, section 3.1 should be presented before explaining the radiative transfer simulations.**

**R:** Thank you for your suggestion. However, we preferred to put the surface albedo in the Results section because the spectral surface albedo was crucial for the radiative transfer calculations and can be considered a stand-alone crucial result of the study along with the irradiance validation.

**2.3 Only limited conclusions**

**The conclusion section does contain a lot of "may"s and "if"s. More questions are raised than answered. The authors theirself are hesitant to draw conclusion: "sheds some light on these questions", "the actual surface albedo may deviate from commonly used climatologies", which is more than obvious. Also the limitations of the limited data set for conclusions is acknowledged. Based on these little new results, the entire section, especially the last part of the conclusion read more like an outlook, indicating, that the study did not improve much. It is not mandatory to make big improvements, but I also do not see any method, approach on how to improve all the issues that are summarized in the conclusion section. To improve the manuscript it would be helpful to present and discuss a method or approach of how to perform validation studies based on similar measurements as shown in the study. As mentioned by the authors, there is potential to process more data from the airborne campaign. To do so, a clear approach with step A, B, C,... should be presented in the manuscript.**

**R:** Thank you for your comments on the conclusion. We changed the wordings where contains "may"s and "if"s to avoid the misinterpretation of a weak conclusion. We wanted to point out that we do not aim to provide a validation approach that can be applied widely to other observations. Instead, this paper is more about a hypothesis study. We are not aware that we were hesitant drawing conclusions. Yes, it is indeed more than obvious that the real surface albedo deviates from the true albedo. At the same time, there is currently no operational product of the spatially and temporally varying surface BRDF (albedo), unlike in most other regions of the world, where this product does exist. Since this is such a big problem, we do believe that even such an obvious statement is warranted. However, this was not the most important part of the conclusions, which was focused on clouds:

We discovered the MODIS misses about 27% of the low-level thin clouds in the Arctic. This had been suspected by earlier studies, but is now confirmed with our data. We do not agree with the reviewer that this is a "weak" conclusion. If this under-detection is a common occurrence throughout the Arctic, the surface budget and melt assessments based on satellite observations could be quite different.

In addition, we presented some lessons learned from ARISE. Most importantly, the separate sampling of above- and below-cloud irradiance in separate flights was not ideal because it impedes a true radiative closure study. This is an important message to be considered in future Arctic field campaigns.

**2.4 Uncertainty Analysis**

**The study aims comparing measured irradiance with simulations. As the data is intended for use in a comparison study, a discussion of the measurement uncertainties is fundamental. No uncertainty ranges are indicated in the plots.**

R: Thank you very much for your comments on the uncertainty analysis. As we realized the uncertainty analysis is very important and was missing in the manuscript, so we added it in Appendix D (**Line 12, Page 17**) of revised manuscript. More importantly, we also added the estimated uncertainties of the measurements and calculations in the plots.

**BBR: What about the accuracy of the data? What is considered in the data processing? Are the BBR instruments actively levelled? The SSFR is levelled. What makes this for a difference comparing SSFR and BBR?**

R: The accuracy of the BBR data is 3%. In contrast to SSFR, the BBR instrument was fix-mounted to the skin of the aircraft. We added this information in the revised manuscript (see **Line 16, Page 5**). A software attitude correction (Long et al., 2010 was applied to BBR data to account for the change of solar position due to aircraft pitching and rolling. Neither the BBR zenith nor the nadir light collector were actively aligned. Only the SSFR zenith light collector was actively aligned. After applying attitude correction for both SSFR and BBR, they shouldn't be much different from a radiation measurements perspective while the aircraft was flying fairly smooth at a slightly positive but stable pitch angle. Without the active leveling, the radiation measured by BBR can be contaminated from side when the aircraft pitched or rolled at a large angle. Since the data was filtered out when the aircraft was ascending, descending or turning, BBR and SSFR should be consistent (the degree to which they are is discussed in the Appendix D, see **Line 12, Page 17**).

**The same holds for SSFR. What are the final uncertainties? How the radiometric calibration contributes to the uncertainties? What is more important, correcting the angular response or tracking changes of the radiometric sensitivity over time?**

R: This is now described in the revised manuscript (see **Line 12, Page 17**). SSFR and BBR are merged into a combined irradiance product, which relies on the higher absolute accuracy of the broadband measurements from BBR and the spectral resolution of SSFR.

**P10 L13: There are several sources of uncertainty in the determination of the surface albedo. How the estimation of the surface albedo affects the uncertainty of the final results/study? Also extend the discussion of uncertainties by the MODIS cloud retrieval. Only undetected clouds have been considered so far. What about cloud phase, the second thin cloud layer, surface albedo assumed in the retrieval?**

R: We added an uncertainty analysis for the surface albedo as well as a discussion of how it propagates into irradiance calculations to the revised manuscript (see **Line 12, Page 17**). After we obtained the uncertainties for the SSFR-BBR combined irradiance product (see **Line 12, Page 17**), the uncertainty of the surface albedo was done through error propagation. The details can be found in the revised manuscript (see **Line 12, Page 17**). We updated the figures contains the surface albedo results with error bars.

In addition to the uncertainty estimate for the surface albedo, we also included the uncertainty of the MODIS cloud retrieval. Based on the cloud phase from MODIS cloud products, only 2% of the clouds were identified as ice clouds. In addition, limited in-situ measurements showed the majority of the clouds were liquid. Therefore we considered only liquid clouds in this study. Since we were not aim to provide a closure study but hypothesizing MODIS cannot detect a portion of thin clouds, the second thin cloud layer (cirrus) was not considered in this study. We have added a comment to this effect in the manuscript (see **Line 14, Page 11**). The surface albedo assumed in the cloud retrieval was included in Figure 5 for comparison.

**3 List of specific comments**

**P2 L9: From the abstract it is not clear, why two independent estimates of the surface albedo (from SSFR/BBR and from the camera imagery) are needed?**

**R:** These estimates are not independent. The camera and BBR/SSFR are used jointly as explained in the paper and also above in this response.

**P2 L16: How large is the radiative effect of the non-detected clouds? This is an important value when MODIS misses a significant faction of clouds.**

**R:** The shortwave cloud radiative effect (CRE) of the non-detected clouds is about -40 $Wm^{-2}$. Since we changed the title of the manuscript by removing the shortwave radiative effects, we decided not to include the discussions for shortwave CRE in the manuscript.

**P3 L14: The study by Hartmann and Ceppi (2014) does not fit to the topic of cloud radiative effects. Direct radiative effects by sea ice loss has nothing to do with clouds unless you argue, that the expected increased cloud cover over increased area of open water is not able to compensate the reduced reflection of solar radiation by the surface.**

**R:** We agree that Hartmann and Ceppi's paper does not fit the topic of cloud radiative effects but we decided to keep their paper here for two reasons. First, Hartmann and Ceppi's paper shows the application of satellite remote sensing in the Arctic to study sea ice extent. Since the radiative effects caused by sea ice loss are associated with albedo feedback, the surface albedo is relevant to what we discuss in our paper. Secondly, Hartmann and Ceppi's paper can be used as a good introductory to bring up the following study by Kay and L'Ecuyer (2013), which discussed the cloud radiative effect and associated sea ice loss in the Arctic.

**P3 L17: Explain acronyms CERES-EBAF, 2BFLXHR-LIDAR.**

**R:** The full name of CERES-EBAF is Clouds and Earth's Radiant Energy Systems – Energy Balanced And Filled. The 2B-FLXHR-LIDAR represents level 2B radiative fluxes and heating rates calculated from radiative transfer model by utilizing radar-lidar cloud and aerosol. We made changes to the revised manuscript (see **Line 18, Page 3**) to introduce these acronyms.

**P4 L2: This presented state of the art on spectral albedo of Arctic surface types is very pessimistic and does not consider recent publications which cover a lot more data also derived from airborne observations (areal and temporal variability):**

**Perovich, D. K., Tucker, W. B., and Ligett, K. A.: Aerial observations of the evolution of ice surface conditions during summer, J. Geophys. Res., 107, SHE24-1–SHE24-14, https://doi.org/10.1029/2000JC000449, 2002.**

**Malinka, A., Zege, E., Heygster, G., and Istomina, L.: Reflective properties of white sea ice and snow, The Cryosphere, 10, 2541–2557, https://doi.org/10.5194/tc-10-2541- 2016, 2016.**

**Malinka, A., Zege, E., Istomina, L., Heygster, G., Spreen, G., Perovich, D., and Po- lashenski, C.: Reflective properties of melt ponds on sea ice, The Cryosphere, 12, 1921–1937, https://doi.org/10.5194/tc-12-1921-2018, 2018.**

**R:** Thank you for the providing these references, which we added. However, our pessimistic view on spectral surface albedo of Arctic inhomogeneous surface originally came from a space perspective (see earlier comment on the lack of an operational product). We weren't being very inclusive about the most up-to-date spectral surface albedo studies for different surface types in the Arctic. We did add more discussion at **Line 2, Page 4** for the provided references.

**P4 L29: Fairbanks is in the center of Alaska. Where did you fly over Arctic sea ice? Figure 1: Add longitude and latitude.**

**R:** These two cases were within the following region: longitude west of -136°, longitude east of 130°, latitude south of 72°, latitude north of 74.5°. Thanks for the suggestion. We added the longitude and latitude to Figure 1.

**P6 L22: Were cloud properties derived from the SSFR measurements? If not, I suggest to remove this statement here.**

**R:** No, the cloud properties were not derived from SSFR in this paper. That said, retrieving cloud optical properties from SSFR measurements is a common application of SSFR. Despite that, we removed this sentence from the original text to avoid confusion.

**P6 L26: What is the resolution (number of pixel) of the camera? What type of lens is used (distortion-free?)?**

**R:** The resolution of the image extracted from nadir video camera is 2592*1952. It is a standard, commercially available video camera. We added this information in the revised manuscript (see **Line 2, Page 7**).

**P7 L4: Instead of using such an interpolation technique, could you determine the vignetting effect by measuring over a white almost Lambertian surface? I could imagine, that a snow covered surface could provide this as a first approximation. For your application this should be sufficient. Or use a certified diffuse reflector.**

**R:** Thanks for your suggestion. It is a good idea to determine the vignetting effect my measuring over a white surface. However, by manually going through the nadir camera images, we could not find an all-white image. Fortunately, our sensitivity study showed that the error contribution in snow fraction from the interpolation factor was minor comparing to other factors (see Appendix D). This is because the corner points of the image (most affected by the vignetting effect) are removed from the snow fraction estimation.

**P7 L5: "Black" means probably "dark" like the dark signal of a non-illuminated camera sensor? In terms of radiation I would prefer "dark". Black is a color and limited to visible wavelength.**

**R:** Actually the "black" matrix here means the matrix contains value of 0 (color black). This is an extrapolation technique explained in Haeberli and Voorhies 1994 (see explanation in the manuscript). We applied the described image processing technique to the nadir camera imagery data to estimate the snow fraction. However, the nadir camera imagery did not provide any radiation data (radiance or irradiance).

**P7 L5: How the 2D matrix was determined? Each camera and lens system must have an individual matrix.**

**R:** The 2D matrix was manually set to compensate for the vignetting effect. The nadir camera was the only camera we used to estimate the snow fraction.

**P7 L15: Can you discuss the retrieved snow fraction of the example and the uncertainties/quality of the method in this section? Only referring to the figure is not sufficient.**

**R:** Thank you for your comments. We added the uncertainty discussion of snow fraction estimated from nadir camera image using adaptive thresholding method the in Appendix D.

**P7 L16, Figure 2: Figure 2a shows the presence of thin gray ice, which is not detected as sea ice in Figure 2b. This means, that from a physical view, the sea ice fraction is underestimated. Although, optically these areas are less bright, they have a higher reflectivity and might bias your results.**

**Can you give an uncertainty estimate, how the sea ice fraction will change with adjusting the threshold between bright and dark pixel?**

**R:** Thank you for the comments on Figure 2. We acknowledge the limitations of the adaptive thresholding method and the uncalibrated nadir camera imagery. However, we do believe that we were able to quantify the snow fraction sufficiently well to subsequently parameterize the surface albedo. To address the reviewer's concern, we conducted a sensitivity analysis (added in the Appendix D).

**As the camera provides RGB images, it should also be possible to classify different ice types following the methods describes by Perovich et al., 2002. Did you thought about this?**

**Perovich, D. K., Tucker, W. B., and Ligett, K. A.: Aerial observations of the evolution of ice surface conditions during summer, J. Geophys. Res., 107, SHE24-1–SHE24-14, https://doi.org/10.1029/2000JC000449, 2002.**

**R:** No, we didn't thought about the method by Perovich et al. 2002 for the original manuscript. Perovich et al. used a histogram thresholding method, which manually set threshold for the histograms of image RGB values to identify different surface types. We began our snow fraction estimation from the histogram thresholding method by using threshold value from the bimodal distribution of the image grayness. However, the method became problematic during automation because the bimodal feature was not clear for all the images and the estimated snow fraction was not accurate by the judgment of human eye. That's why we turned to the adaptive thresholding method. Due to the fact that the nadir camera is not radiometrically calibrated and our major goal was to get only an estimate of snow fraction from the image, we kept the albedo work fairly simple but good enough for automation. The Perovich method is very valuable if done with appropriate equipment. We didn't have that equipment, and the purpose of our study was a different one. In future campaigns monitoring the evolution of sea ice albedo, the Perovich method should be used. We think Perovich et al.'s paper is a great reference so we added a brief description at **Line 4, Page 4** in the revised manuscript.

**P7 L30: For altitudes above 6.5 km, a standard atmospheric profile is used. Aren't there any radio soundings available? Barrow? What about dropsonde releases from the aircraft?**

**R:** We did not use the standard atmospheric profile directly. Instead, we used either the atmospheric profile from a) aircraft itself or the atmospheric profile from b) MERRA2 reanalysis data at the closest

location of the average location of the flight for the atmospheric profile below 6.5 km. For the atmospheric profile above 6.5 km, we first rescaled the standard atmospheric profile with the end point of a) or b) and stitched them together to provide atmospheric profile from the surface to the top of the atmosphere. The radio soundings are available at Barrow but it's far away from the area where the aircraft flew, so the reanalysis was preferable. We did not have any dropsondes on the aircraft. For future campaigns, this will be a must-have.

**P8 L3: It would be helpful to include a figure showing a time series or similar plots of the MODIS COPs which are extracted along the flight path. Just to know, what range of COPs have been present and how variable the cloud field was. What about temporal offsets between MODIS and airborne observations?**

**R:** The MODIS COPs time series and time offset between MODIS and airborne observations are shown in Figure 7. The COPs range from 0 to 16 during the above-clouds case flight (2014-09-11). The temporal offsets between MODIS and the airborne observations range from 0 minute to 50 minutes (flight from 21:10 to 22:45 and the MODIS granule was at 22:00).

**P8 L4: All clouds are assumed to be liquid. Is there any prove for this? In situ observations? The temperature profiles are well below $0^\circ$C where mixed-phase clouds typically are often present.**

**R:** The MODIS cloud phase product shows less than 2% of ice clouds present along the flight track. Of course, the phase detection under these conditions is not reliable. Limited in-situ observations were available (forward scattering probe; imaging probes), and they suggested that the clouds primarily consisted of liquid water. We added a brief statement in the revised manuscript to that effect (see **Line 23, Page 8**).

**P8 L11: What quantities are included in the atmosphere profile? I usually understand also temperature and humidity to be part of the atmospheric profile, but there are provided separately.**

**R:** The atmosphere profile includes the temperature and humidity as well as other atmospheric constituents, such as methane, carbon dioxide, ozone etc. We changed the bullet points structure where temperature and humidity were contained in the atmosphere profile (**Line 32, Page 8**).

**P8 L16: What albedo is assumed here?**

**R:** By using method For the 2014-09-11 (above-cloud case), the surface albedo used parameterized albedo with snow fraction of 0.764. In the revised manuscript, we explain how we arrived at that value (see **Line 38, Page 10**). For the 2014-09-13, the surface albedo at each point along the flight track is calculated based on the snow fraction estimated from the nadir camera imagery.

**P8 L17: Specify or provide the slit function in the instrument description.**

**R:** We added the slit function information– "… full width at half maximum (FWHM) of 6 nm for silicon channels and FWHM of 12 nm for InGaAs channels" (see **Line 2, Page 9**).

**P8 L19: Does MODIS provide cloud base?**

**R:** No, MODIS does not provide cloud base.

**Section 2.5: The description of the radiative transfer simulations should be separated from this MODIS section. The title of the section does not suggest that it will include the methodology of how the solar irradiance is derived. I suggest to add a separate section "methodology". See general comments.**

**R:** Thank you for your suggestion. We introduced the method after the introduction of each instrument in the section of Data and Methods. It will not be necessary to add a separate section "methodology". Instead, we added the description of steps at the beginning of the "Data and Methods" to outline the general approach (**Line 5, Page 5**). In addition, we changed the section to "Radiative Transfer Calculations based on MODIS Cloud Products".

**P10 L8: Why the albedo for 11 September needs to be calculated/constructed? I'm lost ... If you have the ice fraction, why there is now surface albedo measurements?**

R: On 11 September, the aircraft took measurements high above the clouds, where the surface albedo cannot be directly obtained from measurements. Instead, we used the data from 13 September, which led to the parameterization of surface albedo via snow fraction as described earlier. We used that parameterization (with a snow fraction of 0.764, see above) for 11 September since the aircraft was flew in a similar area as on 13 September.

**P10 L9: I was wondering, why you use "snow fraction" instead of the more common "sea ice fraction" or "cover". Likely because there is dark snow-free sea ice. Can you elaborate the term "snow fraction" more clearly in section. 2.3. This would help the readers to understand immediately, that there is a difference to sea ice fraction and why this is relevant.**

**R:** The definition of this fraction is as follows: number of bright pixels divided by the total number of pixels in the nadir camera image. Since we don't know whether the bright pixels were snow or ice or both, we assume the bright pixels are snow most of the time. Thus we named it "snow fraction". The dark pixels, on the other hand, might be either open ocean or "young" sea ice. The latter is supported by its spectral shape (see **Line 23, Page10**).

**P10 L17: How the cirrus is considered in the analysis? How strong does it influence the final results? Especially with respect to the proposed CRE of clouds?**

R: We did not include any cirrus in the radiative transfer calculations (see response for "**P11 L18** …"). From Figure 7, the cirrus affected the downwelling irradiance by about 30 Wm$^{-2}$ at most.

**P10 L23. Figure 7b: There is a large mismatch for areas where no clouds are detected. How strong, the undetected clouds can change the irradiance? I guess, the difference in the "cloud-free" areas is more due to the surface albedo (sea ice fraction) than due to the clouds. $\tau = 0.5$ would not make much difference over bright snow surface. In optically thick areas, the agreement is better. How strong e.g. 10% uncertainty of the sea ice fraction would influence the results here? Can you rule out any change of the sea ice fraction along the flight track?**

R: We cannot rule out a change of the snow fraction along the flight track. In fact, this is rather likely. Relevant for our study is that the calculated irradiances jump abruptly near the edges of the clouds as retrieved by MODIS, whereas the measured irradiances vary smoothly, suggesting that the clouds continue beyond the "sharp" edges as detected by MODIS. This is simply due to the optical thickness

detection threshold of MODIS. Our findings are further supported by photos from the camera. Changing the snow fraction by a certain number would change the upwelling irradiance, but it would not remove the jumps in the calculations and bring it into closer agreement with the measurements. In optically thick areas, the measurement-model agreement is better, simply because the surface has less of an impact on the upwelling irradiance.

**P10 L33-37: These general details of the cloud conditions that have been present during the two flights is needed much earlier. I also suggest to add a comparison of the differences between both observed cloud cases. Otherwise, it is hard to follow the analysis.**

R: Thank you for your comments. We changed the name of the cases to "0911-above-cloud" and "0913-below-cloud" to avoid confusion at **Line 30, Page 10**. Since the cloud optical thickness (cloud conditions) was provided for "0911-above-cloud" and "0913-below-cloud" in Figure 7 and Figure 8, we decided to introduce the cloud condition for each case when the figures was discussed. The details for "0911-above-cloud" flight is at **Line 12, Page 11** and the details for "0913-below-cloud" flight is at **Line 2, Page 12**.

**P10 L40: So far it was not clear, that for the first case, the albedo was fixed. This needs to be made clear at the begin of the analysis. What are the differences between both cases?**

R: We added clarifications in the revised manuscript (**Line 35, Page 10**). Although using the same parameterization, the spectral surface albedo was fixed for the above-cloud case. For the below-cloud case, the surface albedo was calculated based on the snow fraction estimated from the nadir camera image.

**P11 L2: I don't understand, why also for this case the parametrization is used. Isn't the parametrization based on the same data? What are the advantages of this approach?**

R: The parametrization did not use the same data. The parametrization used the data collected under clear-sky condition on 2014-09-13. For the irradiance evaluation of the below-cloud case, due to the presence of clouds, the surface albedo cannot be directly measured, but it can be derive via the parameterization as long as the snow fraction can be retrieved from the camera imagery.

**P11 L14: As I understood, the two days are different in a way, that once the aircraft flew above the cloud layer and once below. This means, that the CRE is defined differently for both cases. So how this is accounted here?**

R: Yes, the two days are different. Due to the different cloud fields from two days, the study is limited to draw conclusions about the CRE for different cases (once the below-cloud, and once the above-cloud CRE). It is worth mentioning that this limitation of ARISE was one of the motivations to propose a new field campaigns with flight plans of flying both above and below for the same cloud field. Yes, the CRE was defined differently for both cases. Therefore, we did not draw any conclusions by saying it is a closure study of CRE.

**Section 3.2. What is the benefit of having both BBR and SSFR broadband irradiance here? Why SSFR was integrated to broadband values and compared to BBR? This makes only sense, if e.g. surface albedo is once considered spectrally and once with a fixed broadband value? Or is there any other purpose?**

R: The SSFR and BBR were used to validate the radiative transfer calculations. BBR was more accurate, but SSFR had spectral resolution, as described above. SSFR's azimuthal response was not isotropic

(unlike BBR's), and it was corrected with BBR's response during a dedicated calibration flight. After azimuthal correction, the SSFR downwelling irradiance was scaled to BBR using the method described in Appendix D. It is in this sense that SSFR and BBR data were "merged". However, they are not identical. Instead, a correction factor is applied to SSFR data that depends on the sun-sensor geometry. Since the calibration flight and the science flight occurred at different sun-sensor geometries, the broadband irradiance derived from SSFR and measured by BBR may deviate slightly from each other. We show them both to check for consistency between both methods. We need to show SSFR (in addition to just BBR) because we later use spectral irradiances from SSFR only to draw conclusions.

**P11 L18: Where the "model-measurement biases in the broadband shortwave CRE" are discussed? The different values in Table 1 and 2 have not been explained. How the different CRE are calculated? How CRE is derived based on measurements and how it is derived based on the simulations? If there are estimates based only measurement, you would need cloud free flight sections, which I do not see in the data. How the cirrus layer is considered in the estimation of the CRE and the radiative transfer simulations? You cannot neglect the cirrus.**

R: We no longer show the CRE calculations to avoid confusion (see above). Even Cirrus was indeed present during some of the flight segments, we did indeed not include it in the calculations. That is because MODIS does not distinguish between Cirrus and low-level clouds. MODIS derives the column-integral of optical thickness regardless of its vertical location. Of course, the measured downwelling irradiance is lowered under Cirrus clouds, and as a result the upwelling irradiances will also be lower than under Cirrus-free conditions. However, what mattered for the manuscript were the low-level clouds. We added a statement to this effect in the manuscript (see **Line 14, Page 11**). Of course, we would have needed to account for Cirrus if the purpose of the manuscript were a closure study.

**P11 L21: Upward or downward irradiance?**

**R:** It's the upwelling irradiance. We changed the text to clarify (**Line 32, Page 12**).

**P11 L25: I don't understand, why you need a climatological surface albedo, when you measured and parameterized the albedo.**

**R:** The climatological surface albedo was used to put our study in context with others, and to illustrate how much different surface albedos can affect the radiative transfer calculations.

**P11 L28: Was the surface albedo fixed or varied in the simulations or do I misunderstood this sentence? That a fixed albedo will not represent reality is more than obvious. That's why I don't fully understand the approach to use a fixed albedo here. How is this motivated?**

**R:** The surface albedo was fixed for the 11 September case (referred to as "0911-above-cloud") but varied for the 13 September case (referred to as "0913-below-cloud"). As we discussed in previous response, the surface condition was unknown when the aircraft was flying over clouds. When the aircraft was flying below clouds, we could not directly obtain surface albedo from measurements because the downwelling irradiance was varying with clouds. Using a fixed surface albedo derived from clear-sky measurements was the best we can do for the above cloud case. We tweaked the surface albedo until the clear-sky radiative transfer calculations matched with measurements at 1640 nm. We added this discussion in the revised manuscript for clarification (**Line 35, Page 10**).

**P11 L29: Specify. Range of what?**

**R:** The range here means the irradiance range from previous sentence. After revision, this text was removed.

**P11 L31: Climate models? How you can draw this conclusion? No climate model is applied, analyzed or discussed in the manuscript. Even when a climate model would have been considered, the "underestimation" may only hold for your specific case, where the albedo is assumed to be too high. This must not hold for the climate models. Further, there are climate models available, which use sophisticated snow albedo parametrization accounting for different sea ice types, melt ponds, snow etc. E.g.:**

**Dorn, W., Rinke, A., Köberle, C., Dethloff, K., and Gerdes, R.: HIRHAM–NAOSIM 2.0: The upgraded version of the coupled regional atmosphere-ocean-sea ice model for Arctic climate studies, Geosci. Model Dev. Discuss., https://doi.org/10.5194/gmd- 2018-278, 2018.**

**R:** We meant that based on the specific case study provided in the paper, the irradiance calculated from the climate models that used the biased surface albedo would be underestimated. We removed the text to avoid confusion.

Also, thank you for the reference. Although Dorn et al. (2018) used a sophisticated snow albedo parameterization accounting for different surface types, the surface albedo they used is a broadband surface albedo, no spectral surface albedo was used. In our paper, we strive to deliver two messages about the surface albedo 1) the spectral surface albedo is important, 2) accounting for the surface variability is important.

**P12 L17: Please write the equation using the symbols of the quantities which are calculated here, not x, y.**

**R:** We revised the equation by changing the symbols. In the manuscript, the equation was changed to $r = a - e^{b \cdot COT + c}$ at **Line 36, Page 13**.

**P12 L19: The behavior shown in Fig. 11 can be explained by the change of surface albedo between cloud-free (direct Sun illumination) and cloudy conditions (diffuse illumination). A Similar behavior is reported by Gardner and Sharp (2010). This means, that you albedo is not necessarily wrong. It depends on what you want. A closure study comparing the irradiance in cloudy conditions required the cloudy-sky albedo. For estimating the CRE, the cloud-free albedo needs to be applied in the radiative transfer simulations.**

**Gardner, A. S., and Sharp, M. J. ( 2010), A review of snow and ice albedo and the development of a new physically based broadband albedo parameterization, J. Geophys. Res., 115, F01009, doi:10.1029/2009JF001444.**

**R:** This is a good point. The basis of our paper is essentially the "blue-sky" surface albedo because that is what we were able to measure. The "white-sky" surface albedo (i.e., the surface albedo under cloudy conditions) does indeed differ from the blue-sky albedo, but we could not measure it for this campaign. For future campaigns, we hope that this can be done. What we see in Figure 11 (transition from 0 to nonzero optical depth) could indeed be caused by this difference. We have added a comment to this effect in the manuscript (see **Line 5, Page 14**).

**P12 L20: Figure 11 and the parametrization is shown for broadband quantities. Figure 12 is calculated spectrally resolved but used the broadband parametrization? Shouldn't the parametrization be computed for each wavelength?**

**R:** The Figure 11 shows the entire fitted function - upwelling broadband irradiance ratio as a function of cloud optical thickness. In Figure 12, the parameterization was computed separately for each wavelength first. Then two ratio values are obtained for each wavelength parameterization: one at cloud optical thickness equals to 0, indicating clear-sky; and one at cloud optical thickness of infinity, indicating cloudy.

**P12 L20: "Spectrum" of what? Upward downward irradiance or albedo?**

**R:** The spectrum here means the upwelling irradiance ratio, which is the ratio between radiative transfer calculations of upwelling irradiance and SSFR measured upwelling irradiance. We changed the "spectrum" to "spectrum of ratio" at **Line 39, Page 13** in the revised manuscript to make it more clear.

**P12 L20: What is x? COT? If yes, then write COT. Also a bracket is missing in line 21.**

**R:** Yes, the x is COT. We changed the x to COT in the text and added the missing bracket.

**P12 L23: "remarkable agreement". Without uncertainty estimation you cannot judge about an agreement.**

**R:** Thanks for your comments. We added the uncertainty analysis in the Appendix D and the results still indicate good agreement between radiative transfer calculations and observations within uncertainty. We removed the sentence of "With these qualifications in mind, the agreement between MODIS-derived and measured irradiance is remarkable" to avoid making subjective statements.

**P12 L33: If the major aim of the study is to show how good irradiances or the CRE can be derived from the MODIS cloud product, the most general scenario should be considered as well. The general case is, that you do not have any airborne observations. Which means, you have to rely on the surface albedo product of MODIS. Atmosphere profiles from reanalysis, etc.. This would be the routine/operational approach. Based on that, you may try to improve the approach by exchanging different assumption with the airborne observations, such as measured surface albedo.**

**R:** We agree. When aircraft observations are not available, the dominating error by far as far as the estimated shortwave fluxes go would be that of the fixed (assumed) surface albedo, which makes an operational surface albedo product so important. However, the point of the manuscript was not just about the flux validation. We hypothesize that MODIS might generally under-detect clouds (regardless of the shortwave fluxes, and regardless of the surface albedo).

**P2 L38: You did not discuss deeply the uncertainties of the MODIS retrieval. The comparison only shows, that there are differences, but could you use the observations to constrain which COPs have been retrieved wrong and to what degree?**

**R:** Thank you for your comments. We added the uncertainty analysis for the radiative transfer calculations, which included the uncertainty of MODIS cloud retrievals (in Appendix D, **Line 12, Page 17**).

**P13 L1: The impact of water vapor profiles was not shown in the manuscript.**

**R:** We added the few calculations with different climatology atmospheric profiles and the results are added in Figure 10. We also added the discussions at **Line 21, Page 13**.

**P13 L3: "developing a validation approach": I don't see a clear validation approach in the study. If there is a strategy, then this needs to be outlined precisely in the begin of the manuscript and summarized here.**

R: Thank you. The validation approach was not clearly described. We added the description of the validation/evaluation steps at the beginning of the manuscript (see **Line 5, Page 5**) to avoid confusion.

**P13 L15: It seems, that the study presented here could not answer any of the questions written in this section. I suggest to start the conclusion section with conclusions and not rise more questions than have been pointed out in the introduction.**

**R:** Thank you for your suggestion. We did raise more questions here, but we think that is adequate (see also our response to the general comments). Our questions are meant as impetus to improve the observing system in the Arctic, or at the very least design future aircraft missions differently such that the questions can be answered. Such missions could (among other things) be motivated by the hypothesis developed here in this paper.

**P13 L35: Define "excellent"! Why you consider the agreement excellent? Any objective measure to judge this? Excellent compared to what? This requires an uncertainty estimation.**

**R:** The description as "excellent" agreement was for the comparison between irradiance calculations with SSFR-BBR irradiance when clouds were detected by MODIS. Of course, any of these qualifiers are relative, and all that can be said is that calculations and observations agree within their uncertainties. We added the uncertainty for both the calculations and measurements and changed the "excellent agreement" to "… agreed … within the range of uncertainties" at **Line 19, Page 15**. We do think that the agreement between calculations and observations is remarkable, provided that MODIS detects the clouds. That is an encouraging result for the quality of the cloud retrievals.

**Appendix A: Can you briefly explain the concept behind these calculations? Is this done to derive the diffuse fraction based on the present cloud cover? What about broken clouds? Is this covered by the approach?**

**R:** The concept is to perform diffuse/direct correction spectrally on SSFR measurements based on the broadband measurements from SPN-1 and spectral calculations from radiative transfer model under clear-sky. This approach covers clear-sky ($f = 1$), broken clouds ($0 < f < 1$), and cloudy ($f = 0$).

**P15 Eq 1: Avoid the large slash.**

**R:** We replaced the large slash with horizontal line. Now the equation is

$$f = \frac{(1 - DR_{SPN1}) \cdot \int_{\lambda_1}^{\lambda_2} F_{clear}^{\downarrow}(\lambda) d\lambda}{\int_{\lambda_1}^{\lambda_2} F_{clear}^{\downarrow}(\lambda) \cdot \left(1 - DR_{clear}(\lambda)\right) d\lambda}$$

**P15 Eq 2: This equation was already given above.**

**R:** Thanks for noticing the duplicate equations. We removed the repeated one.

**P15 L24: How the boundaries of the image are treated if the subdomain is that large?**

**R:** The resolution of the image extracted from nadir video camera is 2592*1952. Although the subdomain size was large, the subdomain was still contained within the boundaries of the image.

**Table 1 and 2: What unit have the numbers? Is this only solar radiative forcing or total (solar + terrestrial)?**

**R:** The units is Wm$^{-2}$ were added. As we discussed before, we decided to remove Table 1 and 2.

**Figure 4: Can you provide the parametrization equation (linear regression?) and the regression coefficients. This might be useful for other research studies.**

**R:** As stated in the text, the parameterization was linear regression. There were hundreds of coefficients and we decided not to include the coefficients in the text but make them available in the supplementary materials (see "s2_surface-albedo-coefficients.h5"). We will upload these regression coefficients in a data file and shared it as supplementary material of this paper.

**Figure 8: a) There are data points behind the text. c) Add a legend. And use two labels c) and d) for the left and right panel. In the caption there is a typo in "MOODIS".**

**R:** We relocated the text label to reveal the data points and added the legend. We used two different labels (c) and (d) for the left and right panels. Thanks for noticing the typo. We corrected the typo "MOODIS".

**Figure 10a: The label hides some of the data.**

**R:** We relocated the label of legend to avoid covering the data.

**Figure 10: Indicate the wavelengths used in Fig. 9.**

**R:** We added two vertical lines in Figure 10 to indicate the wavelengths used in Figure 9.

**Figure 10b: I suggest to remove the ratio here as the absolute irradiance is almost zero in the range of water vapor absorption. It could be better to show absolute differences instead of a ratio. If the comparison should be linked to the broadband irradiance, the absolute difference will be integrated and result in the difference of the broadband irradiance. Short: 5% difference at 500 nm is more important for broadband than 5% difference at 1600 nm.**

**R:** Thanks for your suggestion. We agreed that the irradiance difference would work better than ratio when indicating the spectral contribution to the broadband bias. We replaced the ratio plot with the irradiance difference plot.

**Figure 12: Why τ = 10000 and not infinity? Please use τ instead of x. Also here: Do you need to include the water vapor bands wavelengths and have a y-axis down to zero?**

**R:** We initially used a large value of τ, e.g., τ = 10000, to simply the calculation. In the revised manuscript, we changed τ to infinity and used the analytic solution for the results. We also changed x to τ. No, we do not need to include the water vapor bands. We removed the results at the absorption wavelengths and zoomed in the y-axis.

**Figure 13: Provide equation and coefficients of the fit. Not necessarily in the figure, but in the text.**

**R:** Thanks for the suggestion. We added the equation and the coefficients of the fit to the text in the Appendix B (see **Line 31, Page 16**).

[revised manuscript text omitted]

---

## Author Response (AR2)

**Response to Reviewer #1**

**The authors put a lot of work into improving the manuscript! It is now more clear, what the analysis of the airborne measurements aims for and the results are presented more clearly. I also acknowledge the detailed uncertainty analysis. However, understanding now, what is done in the study, I see one major issue which needs to be addressed more carefully.**

Thank you for your comments. To address the comments in this review, we made some changes in the manuscript and highlighted them in red. The revised manuscript is attached in this response.

**Based on the reviews, the authors now mention the difference between albedo in cloud-free (blue-sky) and cloudy (white sky) conditions. It is acknowledged, that the use of measurements in cloud-free conditions might induce some bias in the albedo.**
**On page 14 line 5 of the revised manuscript the authors conclude, that the blue-sky and white-sky albedo do not differ in their measurements. I think, this conclusion is made from a wrong comparison. Figure 12 shows ratios of modelled vs. measured upwelling irradiance for tau=0.**

Please see the response below.

**This means, that the measurements at tau=0 again represent cloud-free conditions, which is similar to the albedo assumed in the simulations. As far as I understand, here you can not make any conclusion on the effect of blue-sky vs. white-sky albedo.**

Thank you very much for your comment; our conclusions were indeed backwards – we had not noticed this at first. The reviewer is correct that we cannot draw any conclusion about the effect of blue-sky and white-sky albedo from Figure 13 and 14 (P36 - P37) in the revised manuscript (originally they are Figure 11 and Figure 12). Figure 13 means two things: (1) On the right hand side where COT > 6: cloud optical thickness from MODIS is accurate once it detects the clouds. (2) On the left hand side where COT < 2: the surface albedo assumed in the RTM is too small. After finding these, we did the same analysis but spectrally to find where spectrally the surface albedo is too small, and the result is shown in Figure 14. Figure 14 confirms again that clouds optical thickness from MODIS are pretty accurate. Figure 14 also shows that we have a problem past 1400 nm. One reasonable explanation is that the MODIS effective radius is wrong but we cannot really draw this conclusion because the wavelengths are also sensitive to snow grain size.

The spectral ratio of the surface (red in Figure 14) can only be used to determine how appropriate (accurate) the input surface albedo is. We removed the discussion about the blue-sky vs white-sky albedo in the manuscript (P14, L5 in the original manuscript). In addition, to evaluate and blue-sky and white-sky correctly, we looked at the blue-sky and white-sky surface albedo extracted from MODIS albedo product (MCD43A3) (see the response below). We did not

discuss this in the manuscript much, but wanted to give the reviewer's comments justice in this response.

**However, I actually see no need to apply a wrong albedo in the simulations. You have airborne measurements below clouds. Here you can derive an albedo for overcast conditions (white-sky) and implement this into the simulations.**
Please see the response below.

**This would also avoid the complex derivation of the cloud-free surface albedo from parametrizing the camera observations. As carefully analysed by the authors, all observations show no cloud gaps, which makes the assumption of a white-sky albedo the first choice. As the differences between both albedo are not minor, I strongly suggest to consider this approach even when the parametrization of the cloud-free albedo might become redundant. Or you put everything into a different context and ask: What surface albedo would be available from a MODIS albedo product when aiming for calculating cloud radiative effects? In that case, I would show the difference between assuming parametrized blue-sky albedo and directly measured white-sky albedo.**

[Figure]

Figure R1: Spectral surface albedo 1) calculated from the parameterization in the manuscript (black); 2) calculated from direct measurements of SSFR-BBR (red) for the below-cloud flight leg on 2014-09-13. In addition, the blue-sky (blue) and white-sky (green) from MODIS surface albedo product (MCD43A3) for the Svalbard region is plotted. The shaded area indicate the minimum to maximum range. The solid lines indicate the mean values and the error bars indicate the standard deviation.

The reviewer is correct that we have airborne measurements below clouds and the surface albedo (white-sky) can be directly calculated from the observations. We had initially thought about this approach. One major concern associated with that approach is that the above-cloud case and below-cloud case are in two different days and at different locations. Even if we can obtain the

white-sky albedo from the below-cloud case (as we did, in fact – see figure above), it would not be suitable for the above-cloud case since the location changed, and along with it the surface type (and albedo). For example, during the below-cloud flight leg, the average snow fraction estimated from the nadir camera imagery was 91.6%. If we applied the white-sky albedo from this leg for the above-cloud case (estimated snow fraction is 76.4%), the surface albedo would be too high. The surface albedo parametrization via the snow fraction is a simple way to get variable surface albedos. However, it is strictly speaking a blue-sky albedo, which is later applied as white-sky albedo. To assess the difference between blue-sky and white-sky albedo versus the difference between various different ways of obtaining the surface albedo, we plotted the surface albedo 1) directly obtained from the below-cloud leg (as proposed by the reviewer); 2) parameterized through the clear-sky observations for the below-cloud leg (as done in our manuscript); 3) and the blue-sky and white-sky albedo from MODIS surface albedo product. Since the MODIS surface albedo is not available in the Beaufort Sea (ARISE location), we used a land location (Svalbard) as proxy data. This is presented in Figure R1. Figure R1 indicates that

1. The differences between white-sky and blue-sky albedo are fairly small when comparing with the measurement uncertainty;
2. In the MODIS data, no difference was found in the spectral shape of the blue-sky and white-sky surface albedo in the shortest wavelengths;
3. The spectral shape of the surface albedo as observed on the below-cloud leg (white-sky albedo) does deviate from that obtained through the parameterization, both at the shortest and at the longest wavelengths. That is (as stated above) because both data sets are associated with different locations, solar zenith angle, etc. Of course, the white vs. blue sky aspect also has a bearing on the differences, but other factors likely dominate.

In summary, the differences between white-sky and blue-sky albedo are minor (#1) compared to other factors.

Although the spectral shape of the surface albedo from parameterization does not agree with a "typical" MODIS surface albedo, and although the parameterization-derived albedo does not agree with the white-sky surface albedo (#3 above), the parameterization-derived it is still our first choice because (1) it can be nudged to match the observations in a clear-sky region (see discussion of "baseline" measurements, P11, L1), (2) because the directly measured "white-sky" albedo is from a different location and time and cannot be "nudged" to match the data, (3) because the parameterization-derived surface albedo has a smaller uncertainty in the near-infrared wavelength range, compared to the white-sky albedo.

**Minor comments:**

**P2, L17: add "effect on the downwelling (upwelling) irradiance". just to avoid misunderstandings with CRF.**

Thank you for your suggestions. Based on this comment and the comment from the other reviewer, we decided to remove the statement in the abstract to avoid confusion.

**P10, L35: slightly misleading: the "0911-above-cloud" case does not account for changes along flight track.**

Thank you for your comments. We added additional clarification - "The SF was determined differently for the two cases." (P10, L36).

**P11, L6: Just an idea: You could test the effect of assuming a constant SF vs. variable SF, when you do the same for the "below-cloud" case (use a fixed SF) and compare to the simulations when changes of surface albedo are considered.**

We have thought about this initially. Unfortunately, for the above-cloud case, the snow fraction (*SF*) cannot be derived from nadir camera imagery like the below-cloud case because the clouds are overcast, and the surface cannot be seen through the clouds by the camera.

**P11, 15: Still the potential effects of the cirrus are not sufficiently discussed or quantified to my point of view. Your comparison may just coincidently match! You compare upward irradiance which for MODIS may assume a higher optical thickness than present (MODIS optical thickness will include both cirrus and low level cloud). At the same time, the measured downward irradiance in flight altitude is reduced but not the simulated downward Irradiance (no cirrus in the RTM). Both would add up for higher upwelling irradiance simulated with MODIS compared to the observations. This needs to be discussed! Describe how the cirrus will affect the irradiances and may be a short simulation will estimate the impact in W m$^{-2}$.**

Thank you for your comments. We think there may still be a misinterpretation about the main intent of the paper. Our study does NOT aim to provide a closure study. However, the reviewer is correct that the irradiances might coincidently match due to the presence of cirrus. Figure 7a in the manuscript indicates that the cirrus can lead up to a 10% (40 Wm$^{-2}$) decrease in the measured downwelling irradiance. However, we specifically excluded the cirrus-affected data segments from our analysis. To make this more clear than in the previous version, we added clarification in the current version (P11, L19). The analysis we provided for the above-cloud case only includes

cirrus-free data. Since we did not attempt a closure study, the conclusion of the manuscript does not change.

**P13, L2: You adjusted SF to match the cloud-free measurements and used 1640 nm wavelength. Using this wavelength, which is very sensitive to snow properties (snow grain size), and may not result in a correct broadband albedo. You mentioned, that at 860 nm the agreement was worse. So ice fraction might differ from you SF estimate. Therefore, I recommend to name this adjusted SF an "effective SF".**

The reviewer is correct. The reflectance at 1640 nm is very sensitive to snow properties (snow grain size). This is further confirmed by the surface albedo plot (Figure R1) attached in the response: The surface albedo differs in the near infrared wavelengths between Svalbard and Beaufort Sea and is most likely caused by the different snow grain sizes (Svalbard MODIS (old snow) vs. Beaufort SSFR (new snow)).

However, the 1640 nm has much less variability than other wavelengths and is therefore more reliable to get the effective snow fraction than other wavelengths, e.g., 860 nm. (This is also why King et al. (2004) picked this wavelength to retrieve clouds over snow surface.) Given that, we agree that "effective" (or "inferred") SF is a better description than "SF" itself. We added clarifications for the SF (P11, L1) to indicate the SF reflects an "effective" rather than the actual snow fraction.

**P14, L23: change into "approach that did help" ... later you show answers to your questions.**

Thank you for your suggestion. We changed from "approach that can help" to "approach that did help" (P14, L32).

**P15, L27-30: Here and in the introduction the uncertainty of available and need for new cloud retrieval over snow surfaces is discussed. Some work into this direction has been done in recent years which should be mentioned in such a discussion:**

**Ehrlich, A., Bierwirth, E., Istomina, L., and Wendisch, M.: Combined retrieval of Arctic liquid water cloud and surface snow properties using airborne spectral solar remote sensing, Atmos. Meas. Tech., 10, 3215–3230, https://doi.org/10.5194/amt-10-3215-2017, 2017.**

Thank you for providing the reference. We are aware of this paper. However, the issue we see is the current lack of spaceborne retrievals. This paper introduces innovative aircraft retrievals, but those have not been applied to spaceborne remote sensing yet. We added something in the

summary, as follows: "Such retrievals (e.g., Ehrlich et al., 2017) will need to account for surface and cloud variability, and address the issue of undetected thin clouds." (P16, L7)

**Rolland, P., and Liou, K. N. ( 2001), Surface variability effects on the remote sensing of thin cirrus optical and microphysical properties, J. Geophys. Res., 106( D19), 22965–22977, doi:10.1029/2001JD900160.**

This is a good paper, however, we do not think this reference fits the scope of our manuscript. This reference introduced a novel method to retrieve cirrus optical properties over various surface while retaining low uncertainties. Since cirrus retrieval is not our focus in the manuscript, we decided not to include the reference.

**Figure 7: To present a consistent data analysis, think about adding also histograms to Figure 7 (upward irradiance above cloud) similar to the comparison of the below-cloud case.**

Thank you for your suggestion. We added histograms similar to below-cloud case for the above-cloud case (see Figure 8, P31). In the newly added histograms, only the cirrus free data is presented.

Histograms of cirrus-free data (included as Figure 8 in the revised manuscript):

[Figure]

**P31, There is something wrong with the figure. No number and caption.**

Sorry for the confusion. We realized this figure took too much space, which separated the figure and figure caption into two different pages. To avoid confusion, we moved the histograms (originally Figure 8c and 8d) into a separate figure (now Figure 10a and 10b, P33).

**Response to Reviewer #2**

**General comments**

**This paper is a substantially revised version of a manuscript already reviewed in November 2019. The points raised by this reviewer were considered adequately in the revised version. In particular an uncertainty analysis was included that was missing in the first version.**

Thank you for your comments. We made some changes in the manuscript to address the comments in this review. The revised manuscript is attached in this response and the changes are highlighted in red.

**The paper should be published after minor corrections. Please see specific comments below.**

**Specific comments**

**Page 2, line 17: "The radiative effect of clouds that were detected…was -40 Wm$^{-2}$ (-39 Wm$^{-2}$)….". I think this statement is a leftover from the first version. These results are not discussed in the manuscript. Instead the effect of the undetected clouds should be quantified here.**

Thank you for your suggestion. The reviewer is correct that the statement is a left over from the first version. Based on this comment and comment from the other reviewer, we decided to remove this statement in the abstract to avoid confusion.

**Page 9, line 31: The flight altitude of 240 m does not fit to that given in Fig. 2 (134 m in accordance with the video information).**

Thank you for noticing the wrong flight altitude. The 240 m was calculated for "0913-below-cloud" and misplaced here. We now corrected the value to 149 m for the "0913-clear-sky" (P9, L31).

**Page 11, line 25: I think there is an inconsistency regarding the FOV definition. With a 45° FOV you wouldn't cover half of the hemispheric irradiance (isotropic), you need 90° FOV and your FOV diameter is 14 km when you fly at 7 km altitude. I assume that this was actually applied because the horizontal error bars in Fig. 7 seem to have the correct ±7 km size.**

Thank you for noticing the inconsistency. What we meant was a 90° FOV with diameter of 14 km when the aircraft was flying at 7 km. The FOV angles we provided in the manuscript were half hemispheric angle. We corrected the 45° to 90° and diameter of 7 km to 14 km when the aircraft was flying at 7 km (P11, Line 30). We also realized the FOV angles we put for video camera were wrong in Appendix D (P18, L24-25), and we revised them. We also revised the text and figure legend for the camera FOV accordingly.

**Page 12, line 14: "… except for the time period before 22:22:48". This was statement was already noted in the first version. Do you mean 22:21:48? In any case the x-scales in Figs. 8 (a) and (b) are possibly wrong. Was there a 1 min break in the time series (22:23 is missing)? If so that should be indicated by an empty period or at least a vertical line.**

Thank you for noticing the typo. It was not until now that we realized the label of the x axis was wrong. We forgot to put seconds in the time stamp, which caused the confusion. We revised the labels of the x axis by adding in the seconds. Now one can find 22:22:48 on the x axis (P32).

**Page 13, line 11: "Of all pixels along the flight leg with a MODIS-COD below the detection threshold of 0.5 (i.e., "clear"), 27% (highlighted in green) are actually cloudy where MODIS cloud detection algorithm identified as clear-sky." This statement is unclear and somehow in contradiction with the Abstract information: "27% of clouds remained undetected". So what was calculated? The fraction of undetected cloud / total cloud periods or the fraction of undetected cloud / total clear (MODIS) periods? Please compare with the statement in the first review and clarify.**

Thank you for your comments. The new percentage (27%) was calculated through the green/(cloudy+green) ratio. We changed the statement to "Among all the cloudy pixels along the flight leg (i.e., pixels with clouds above or below the MODIS detection threshold), 27% (highlighted in green) are actually cloudy even though MODIS identifies them as clear sky." (P13, L21).

**Page 15, lines 14 and 16. There is little quantitative information the reader finds in the Conclusions. So the numbers you give should be accurate and have a clear meaning. The 40 Wm$^{-2}$ given for the above cloud case was stated as 30 Wm$^{-2}$ within the text (page 11, line 34). It refers to time periods where you think thin clouds were present. For the below cloud case the 45 Wm$^{-2}$ is a mean of the whole measurement period during which clouds were occasionally undetected (Fig. 8 (c)). So the numbers are not comparable, as implied in the text.**

Thank you for noticing the inconsistency and thank you for your comments about the incomparable values. In the revised manuscript, we modified the conclusion and provided the

quantitative biases we obtained from our study (P15, L23). To make the biases we stated for above-cloud and below-cloud cases comparable, we added histograms for the above-cloud case (Figure 8 in the revised manuscript) and modified the histograms for the below-cloud case (Figure 10 in the revised manuscript). We arrived at a up to 13 Wm$^{-2}$ bias for the above-cloud case from the histogram. We edited the values and made them consistent throughout the manuscript.

**Page 18, line 17 (and Fig. A3): Please check for consistency with the FOV definition discussed above.**

Thank you for noticing the wrong FOV angles. We confirmed that the angles we put in are only half of what they should be. We corrected the values in the text (P18, L24-25) and in Figure A3.

**Fig. 1, Fig. 2 (caption), Fig. 3 (caption) and Fig. 8: Please use consistent notations of longitudes and latitudes.**

We changed the longitudes and latitudes notations in Figure 2 (caption), 3 (caption), and 8 so they have the same notation style as Figure 1.

**Fig. 2: "The diameter of the field of view was about 380 m" Please check again. Is this the diameter or the radius of the field of view?**

We confirmed that the 380 m was radius instead of the diameter of the field of view. We corrected the diameter to radius in the figure caption (P25).

**Fig. 3 (a) and (b): Typo "Temperature"**

Thank you for noticing the typo. We corrected the label of the upper y axis (Figure 3, P26) to the correct spelling.

**Fig. 6: "…cloud optical thinness of…"?**

Thank you for noticing the typo. The typo was made in Figure 7. We changed it to the correct spelling (P30).

[revised manuscript text omitted]